https://doi.org/10.1038/s42003-022-03851-6　　OPEN
# Engineered endosymbionts that alter mammalian cell surface marker, cytokine and chemokine expression

Cody S. Madsen [1,2,4], Ashley V. Makela[1,2,4], Emily M. Greeson[2,3], Jonathan W. Hardy[2,3] & Christopher H. Contag [1,2,3✉]

Developing modular tools that direct mammalian cell function and activity through controlled delivery of essential regulators would improve methods of guiding tissue regeneration, enhancing cellular-based therapeutics and modulating immune responses. To address this challenge, *Bacillus subtilis* was developed as a chassis organism for engineered endosymbionts (EES) that escape phagosome destruction, reside in the cytoplasm of mammalian cells, and secrete proteins that are transported to the nucleus to impact host cell response and function. Two synthetic operons encoding either the mammalian transcription factors *Stat-1* and *Klf6* or *Klf4* and *Gata-3* were recombined into the genome of *B. subtilis* expressing listeriolysin O (LLO) from *Listeria monocytogenes* and expressed from regulated promoters. Controlled expression of the mammalian proteins from *B. subtilis* LLO in the cytoplasm of J774A.1 macrophage/monocyte cells altered surface marker, cytokine and chemokine expression. Modulation of host cell fates displayed some expected patterns towards anti- or pro-inflammatory phenotypes by each of the distinct transcription factor pairs with further demonstration of complex regulation caused by a combination of the EES interaction and transcription factors. Expressing mammalian transcription factors from engineered intracellular *B. subtilis* as engineered endosymbionts comprises a new tool for directing host cell gene expression for therapeutic and research purposes.

[1] Department of Biomedical Engineering, Michigan State University, East Lansing, MI, USA. [2] Institute for Quantitative Health Science and Engineering, Michigan State University, East Lansing, MI, USA. [3] Department of Microbiology and Molecular Genetics, Michigan State University, East Lansing, MI, USA. [4] These authors contributed equally: Cody S. Madsen, Ashley V. Makela. ✉email: contagch@msu.edu

The endosymbiont theory of the origin of eukaryotic cells postulates that there was a close and long-term biological interaction (symbiosis) between separate single-cell organisms that led to the genesis of organelles[1]. This process can be mimicked in the laboratory, as was demonstrated by engineering *Escherichia coli* to survive within cells of *Saccharomyces cerevisiae*[2]. In addition, bacteria can be engineered to express mammalian transcription factors (TFs) that alter cell fate. Extracellular *Pseudomonas aeruginosa* that deliver TFs via type III secretion into induced pluripotent stem cells (iPSCs) can direct the differentiation of the iPSCs into cardiomyocytes[3]. Together, these studies support the development of non-pathogenic engineered endosymbionts (EES) that persist in the host cell cytoplasm and influence control of mammalian gene expression. The term EES refers to the functional combination of bacteria that remain viable in the cytoplasm of mammalian cells with the engineered production of modulators (proteins, metabolites or nucleic acids) that can redirect host cell biology.

To demonstrate both cytoplasmic persistence of the EES and host cell fate alteration, phagocytic immune cells[4] provide a useful model. Because phagocytic immune cells readily internalize bacteria and demonstrate altered cell fate that is the result of specific TFs, these cells represent a testable system for EES function[4]. The immune response is delicately balanced in mammalian systems, both recognizing self and building tissues and defending against disease and foreign invaders capable of damage. Macrophages are commonly recruited by stimuli in response to inflammation and contribute to progression or suppression of associated pathologies[5], thus representing a significant component of the inflammatory microenvironment[6]. Macrophages may prove to be a key cell for molecular therapies directed at modifying cellular functions, since these cells are present within injured, damaged and malignant tissues and can be modulated to switch phenotypes to alter the disease course[7]. Macrophages are plastic, with the ability to alternate between synthesizing pro-inflammatory or anti-inflammatory signals[8] and their function is influenced by the microenvironment in which they reside. Pro-inflammatory macrophages (M1) act in a fashion to destroy pathogens, and anti-inflammatory (M2) macrophages decrease inflammation, support angiogenesis and promote tissue remodeling and repair[9,10]. This dichotomy is known as macrophage polarization. Polarized inflammation effector states are distinguished by changes in cell surface markers including, cluster of differentiation (CD)86 for M1 or CD206 for M2, or through differential expression of cytokines and chemokines[11].

*Bacillus subtilis* expressing listeriolysin O (LLO) from *Listeria monocytogenes* is an engineered intracellular bacterium[12,13]. LLO lyses the phagocytic vacuole, releasing internalized bacteria into the cytosol. The *hlyA* gene encoding LLO was placed under control of an isopropyl β-D-1-thiogalactopyranoside- (IPTG) inducible promoter and inserted into the genome of *B. subtilis*[12,13]. Since *B. subtilis* expressing LLO can access the cytoplasm and does not have a lipopolysaccharide- (LPS) mediated immune response[14], it was chosen as a chassis organism for the development of a cell fate-controlling EES. Additionally, *B. subtilis* is a non-pathogenic, Gram-positive, soil bacterium that respires as a facultative anaerobe making it capable of replicating in the host cell cytoplasm[15]. This bacterium has been classically used for secreting complex proteins into the surrounding extracellular space through the general secretory (Sec) and twin-arginine translocation (Tat) pathways[16]. *B. subtilis* has been well characterized to the point of full genome annotation and databases (*e.g.* BsubCyc database) have been developed for metabolism analysis and protein production[17]. Several sugar-regulated inducible systems[18] including D-mannose, which has been shown to be actively transported inside of mammalian cells, provide additional techniques to regulate EES gene

expression[19,20]. *B. subtilis* is an ideal chassis organism for development of an EES.

Bacteria have been developed that impact mammalian cell physiology for therapeutic approaches, bacteriotherapy, and advances in this field support the development of EES for cellular control. Bacille Calmette-Guerin (BCG, the *Mycobacterium bovis* strain used as a tuberculosis vaccine) bacteriotherapy has become standard of care for bladder cancer, and other clinical applications for bacteriotherapy are being tested[21–27]. Here, the EES can be used to modulate mammalian cell function by expressing engineered operons that encode mammalian TFs that are delivered to the nuclei of mammalian cells (Fig. 1 and Supplementary Fig. 1). Due to TFs regulating the expression of groups of genes to direct cellular fates[28–31], TFs are being used in ongoing clinical trials in new efforts to alter cellular function for therapies in cancer, wound repair, regeneration and immune modulation[32]. However, delivery mechanisms for TFs are limited and in need of new strategies[32]. Therefore, an EES was designed to express and deliver TFs in two operons for modulating macrophage phenotype towards pro- or anti-inflammatory states. One operon encodes the TFs signal transducer and activator of transcription 1 (STAT-1) and Krüppel-like factor 6 (KLF6) which induce a general response to an inflammatory state in macrophages, and the second encodes Krüppel-like factor 4 (KLF4) and GATA binding protein 3 (GATA-3) which are both characterized to drive an anti-inflammatory response in macrophages[33–37]. Both STAT-1 and KLF4 are upstream regulators that impact several pathways, while KLF6 and GATA-3 are more specific in regulation, which lends to a dual approach towards driving the desired cell fates[33–37]. When expressed from intracellular EES these TFs altered patterns of cell surface markers, cytokine and chemokine expression with some patterns of modulation towards anti- or pro-inflammatory phenotypes, indicating that the EES may be used to direct immune cell function and elucidate mechanisms of macrophage response to intracellular bacteria.

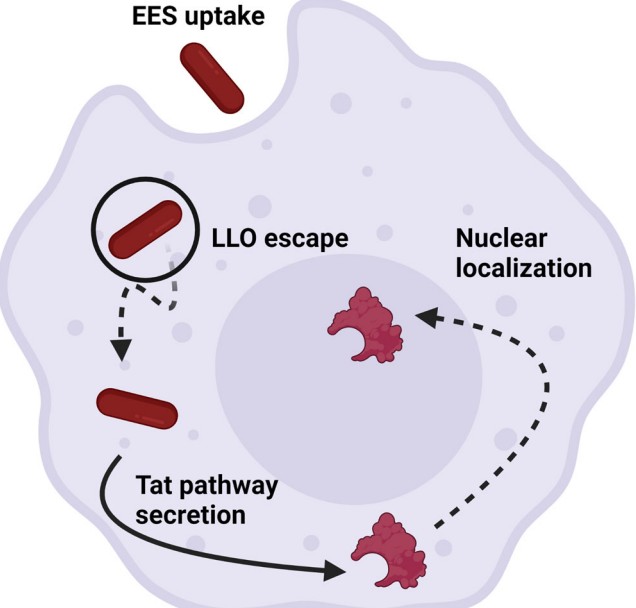

**Fig. 1 EES as a means of controlling gene expression in mammalian host cells.** The EES enter phagocytic mammalian host cells and escape the phagosome using the LLO protein. The EES then secrete a reporter protein or transcription factor into the cytoplasm through the Tat pathway followed by localization to the host cell nuclei. Expression of mammalian transcription factors from the EES were shown to direct macrophage function.

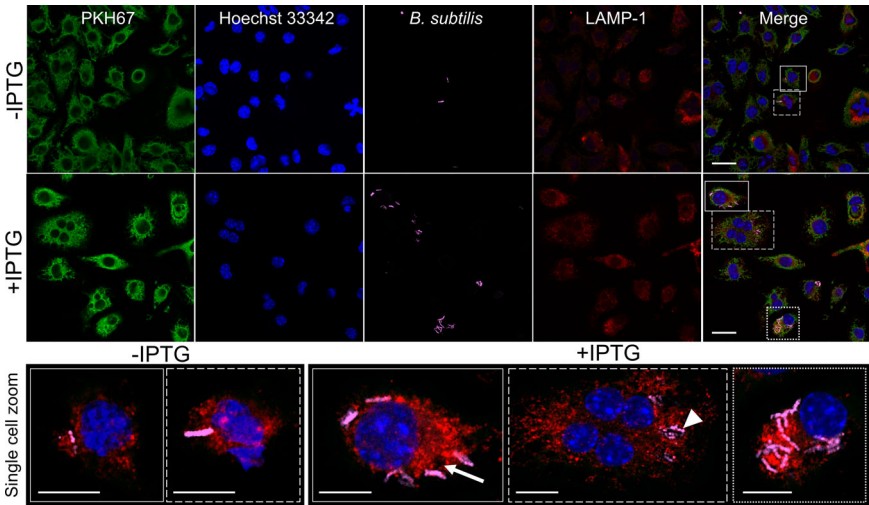

**Fig. 2 Confocal imaging identifies LLO strain phagosomal escape into the cytoplasm of J774A.1 cells.** Confocal imaging was used to identify the PKH67 membrane stain (green; J774A.1 cells), nuclear Hoechst 33342 (blue), *B. subtilis* (magenta) and LAMP-1 (red). The LLO strain was introduced to J774A.1 cells at a multiplicity of infection (MOI) of 25:1 and treated without (-IPTG) or with IPTG (+IPTG). Examples of single cells are displayed as zoomed regions with Hoechst 33342, B. subtilis and LAMP-1 channels merged (below). Without IPTG, there were few *B. subtilis* positive regions (dashed line), mostly consisting of punctate regions of signal (solid line). When LLO was induced with IPTG (+IPTG), there was evidence of *B. subtilis* LLO escape. Empty LAMP-1 structures could be identified (solid line, white arrow) with adjacent *B. subtilis* LLO. There were also *B. subtilis* LLO which had not yet escaped the phagosome (dashed line, white arrowhead) but the majority of identified *B. subtilis*LLO were within the cytoplasm of the cells (dotted line). The z-depth was chosen for each zoomed image and each channel was adjusted to provide a representative image of each scenario. Scale bars = 20 μm (upper); scale bars = 10 μm (lower).

## Results

**B. subtilis LLO escape from phagosomes of J774A.1 cells.** Confocal microscopy confirmed the escape of *B. subtilis* LLO from phagosomes after uptake into J774A.1 cells[12]. To further elucidate the mechanism of escape, *B. subtilis* LLO (magenta) localization was compared to LAMP-1[38] positive structures (phagosomes, red) in J774A.1 cells (green) with and without IPTG induction of LLO expression (+IPTG and -IPTG). The LAMP-1 protein is crucial for phagosomal assembly and therefore will reveal when the LLO strain is contained within the phagosomes and when the phagosomes have been disrupted[39]. When LLO expression was induced by IPTG, many of the LLO strain were intact and present throughout the mammalian cells (Fig. 2, zoom-dotted line; Supplementary Movie 1). Z-stack data analysis identified *B. subtilis* LLO throughout the cytoplasm of J774A.1 cells and not associated with LAMP-1 positive structures (Fig. 2, zoom-solid line; Supplementary Movie 1). In contrast, without IPTG induction, few of the LLO strain were observed and many regions of punctate signal within LAMP-1 positive regions were observed (Fig. 2, zoom-solid line; Supplementary Movie 2). Accordingly, the expression of LLO when induced by IPTG allows *B. subtilis* LLO to access the cytoplasm of the host cells.

**Viability of J774A.1 cells and B. subtilis LLO replication in the host cell cytoplasm.** Host cell viability was assessed by an MTS assay and flow cytometry after delivery of the LLO strain at different multiplicity of infection (MOI) at 2 different time points. At 1 h (h) post-bacterial addition, there was significant change in host cell viability only at the highest MOI (50:1; Fig. 3, left panel). At 4 h post-bacterial addition, both the 25:1 and 50:1 MOI conditions revealed significant losses in host cell viability (Fig. 3, left panel). The same trend was observed using flow cytometry to assess cell viability (Supplementary Fig. 2). From these results, an MOI of 25:1 was chosen for the LLO strain delivering proteins to the nuclei of host cells. To determine the viability of *B. subtilis* LLO (LLO strain) in the cytoplasm, live cell imaging was performed to image the interaction of the fluorescently stained LLO

strain in live J774A.1 cells. With IPTG induction, the LLO strain was observed to replicate in the host cell cytoplasm after phagosomal escape indicating active metabolism and viability (Fig. 3, right panel). An increase in the number of bacteria was visualized over time, after co-incubation with a 10:1 MOI. Zoomed regions demonstrate that each bacterium doubled twice during the two time points from 3 bacteria at 1 h to 12 at 2.5 h in this representative instance (Fig. 3, two right panels). Additionally, uptake of the LLO strain by the host cell was quantified in multiple conditions. The number of cells containing *B. subtilis* LLO with rod morphology and the number of bacteria per cell were determined (Supplementary Table 1), alongside viability assessments using an MTS assay. At an MOI of 10:1, the doubling trend of the LLO strain was confirmed across the population as seen with the representative instance in Fig. 3 (Supplementary Table 1). At each MOI tested, approximately 50% of the added LLO strain entered host cells and at MOIs of 25:1 or 50:1, nearly 100% of the host cells contained the LLO strain (Supplementary Table 1).

**Engineered B. subtilis LLO secretes β-gal with delivery to the nuclei of J774A.1 cells.** Protein secretion from *B. subtilis* LLO was used to further demonstrate bacterial viability within the cytoplasm and as an initial demonstration of protein delivery to the nucleus. *B. subtilis* LLO was engineered to produce and secrete β-galactosidase (β-gal) (strain designated LLO-*lacZ*) through the twin-arginine translocation (Tat) pathway by synthesizing the PhoD signal peptide[40] and the amino acids from 126-132 of the simian virus (SV) 40 nuclear localization signal (NLS)[41]. The production of β-gal by LLO-*lacZ*, with and without a nuclear localization signal (NLS; LLO-*lacZ*-NLS and LLO-*lacZ*-no NLS), was studied with IPTG or mannose control. The mannose-inducible system was amplified from the genome of the *B. subtilis* strain 168[19] to provide a genetic switch specifically for controlling protein delivery to nucleus. The mannose-inducible system was chosen due to the characterized uptake of this sugar in mammalian cells[20]. Localization of the reporter protein to J774A.1

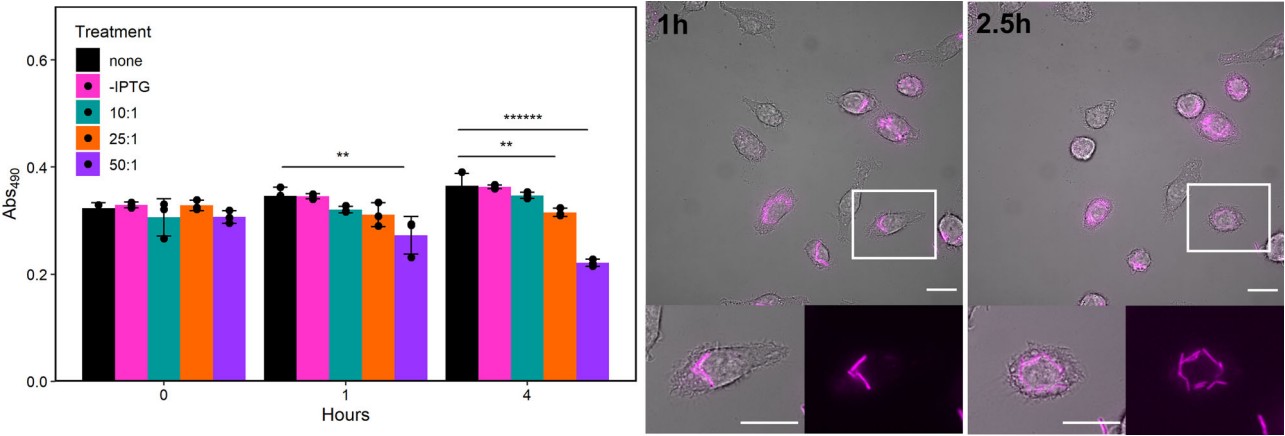

**Fig. 3 Host cell viability and replication of the LLO strain in the cytoplasm of J774A.1 cells.** J774A.1 cell viability at multiple time points after treatment with the LLO strain under various conditions (left, MTS assay). Live cell microscopy revealed the LLO strain replicating in a single host cell by comparing images at 1 h and 2.5 h post-bacterial addition (right). J774A.1 cells were visualized in brightfield, and the LLO strain using fluorescence (magenta); zoomed images reveal the LLO strain replication in the cytoplasm. Plotted data is mean ± SD from $n = 3$ biological replicates; **$p < 0.01$, ******$p < 0.000001$. Scale bars = 20 μm (upper); scale bars = 20 μm (lower).

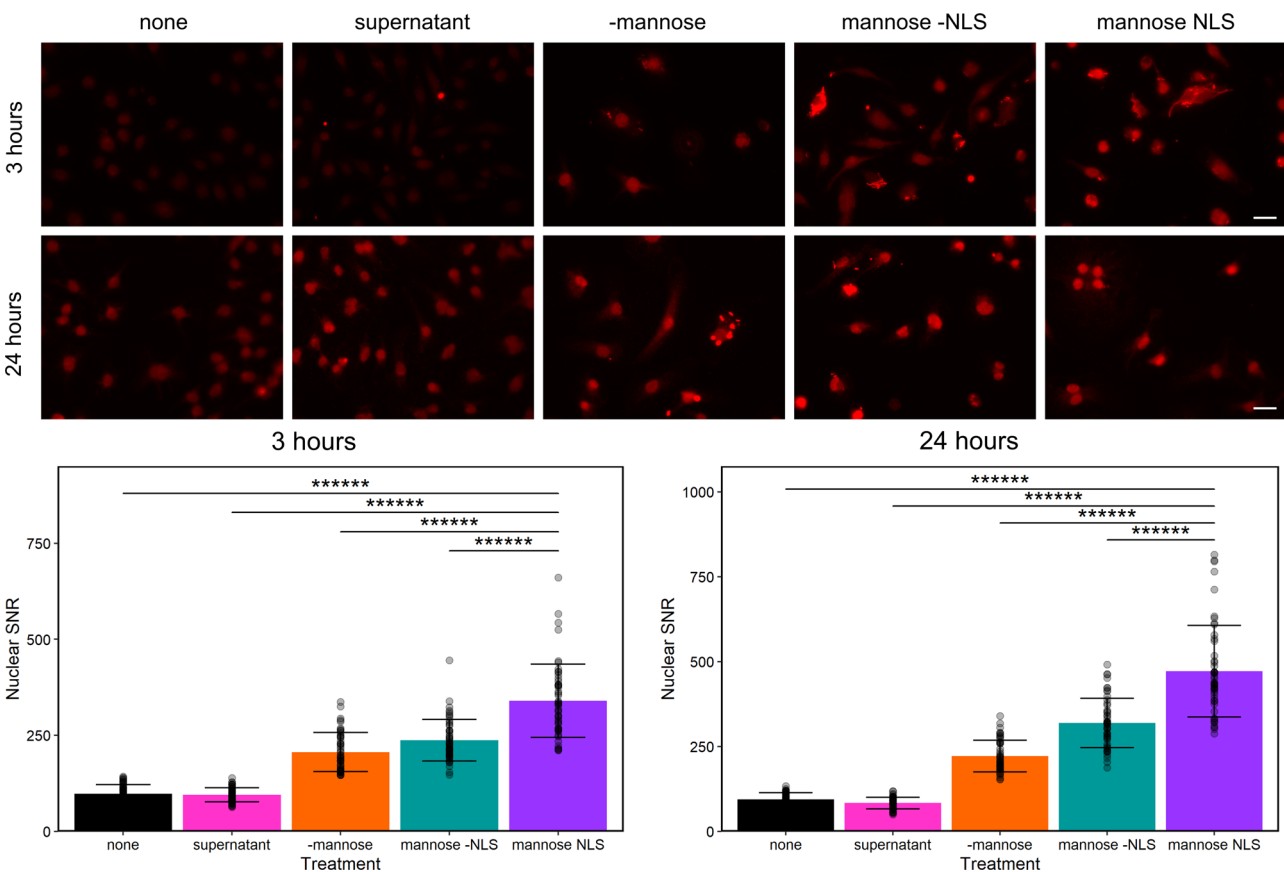

**Fig. 4 Intracellular localization of LLO-lacZ secreted β-gal.** The presence of β-gal in the nuclei of host cells was determined by measuring fluorescent signals of anti-β-gal in the nuclei compared to background signals as a ratio (SNR). Fluorescence microscopy (top) and SNR (bottom) of the following: J774A.1 cells with no LLO strain (none), J774A.1 cells incubated with β-gal collected as supernatant from induced LLO-*lacZ*-NLS (supernatant), J774A.1 cells incubated with uninduced LLO-*lacZ*-NLS (-mannose), J774A.1 cells incubated with induced LLO-*lacZ*-no NLS (mannose -NLS) and J774A.1 cells incubated with induced LLO-*lacZ*-NLS (mannose NLS). Plotted data is mean ± SD from $n = 50$ random individuals from a representative experiment; ******$p < 0.000001$. Scale bars = 20 μm.

nuclei was confirmed after co-incubation with LLO-*lacZ* and end-point fluorescence microscopy (Fig. 4 and Supplementary Fig. 3). After incubation of J774A.1 cells with mannose-induced LLO-*lacZ*-NLS (3 h), the nuclear to background fluorescence signal to noise ratio (SNR; Supplementary Fig. 3) was higher than that observed for untreated J774A.1 cells, cells incubated with β-gal protein from supernatant of LLO-*lacZ*-NLS cultures, non-induced LLO-*lacZ*-NLS and LLO-*lacZ*-no NLS (Fig. 4). At 3 h, the mannose-inducible

system was more efficient than the IPTG-inducible system as indicated by the intensity of fluorescent signals from the nuclei (Supplementary Fig. 4). For this reason and to provide a second control switch specifically for protein delivery, the mannose inducible system was used for the additional studies. Further, the addition of a high concentration of gentamicin (25 μg/mL) at 3 h after induction of β-gal eliminated intracellular bacteria and rescued the J774A.1 cells from overgrowth of the LLO-lacZ, allowing for another 21 h of trafficking of protein to the nucleus (Fig. 4, lower image panels; Supplementary Fig. 3). Nuclear β-gal SNR of cells exposed to mannose-induced LLO-lacZ-NLS for 3 h and then incubated for 21 additional hours (Fig. 4) was higher than the controls of J774A.1 cells only (5.1-fold), cells incubated with the supernatant of induced LLO-lacZ-NLS, no mannose induction (2-fold) and cells exposed to the LLO-lacZ-no NLS strain (1.5-fold; Fig. 4). In summary, the mannose-inducible system in the LLO strain was shown to be controlled inside mammalian cells and to provide regulation of secreted proteins that are directed to the J774A.1 nuclei.

**Engineered *B. subtilis* LLO transcription factor delivery and modulation of J77A.1 cell marker expression.** β-gal delivery by the LLO-lacZ strains demonstrated the possibility of an intracellular EES to functionally deliver proteins capable of transcriptional regulation to the nuclei of host cells. Accordingly, TFs were chosen to be delivered by the EES to alter host cell function. The LLO strain was engineered to secrete two distinct pairs of TFs known to impact macrophage function: STAT-1 and KLF6 (LLO-SK; pro-inflammatory) or KLF4 and GATA-3 (LLO-KG; anti-inflammatory) (Supplementary Fig. 5). Introduction of the TF pairs into the genome of *B. subtilis* LLO minimally impacted bacterial growth rates, did not adversely affect J774A.1 cell viability compared to the LLO strain and did not alter the ability to escape destruction of phagosomes and persist in macrophages (percent cells containing bacteria and distribution of the fluorescence intensity relating to number of bacteria per cell; Supplementary Fig. 6). After staining for protein expression, quantification of fluorescence confirmed production and delivery of STAT-1/KLF6 and KLF4/GATA-3 in cells containing LLO-SK or LLO-KG, respectively, after 3 h of TF delivery (Supplementary Fig. 7). Quantification of nuclear SNR identified increases in nuclear fluorescence in all four TFs after delivery from the LLO-SK or LLO-KG strains with and without addition of D-mannose. D-mannose significantly increased delivery of all four TFs. Additionally, the positive controls for STAT-1/KLF6 (LPS and IFN-γ) and KLF4/GATA-3 (IL-4 and IL-13) produced some increase in nuclear SNR but significantly less than the LLO-SK and LLO-KG strains over the 3 h.

To determine the impact of engineered *B. subtilis* LLO strain TF delivery on J774A.1 modulation, surface marker expression was examined in cells containing various strains (168 strain, LLO strain, LLO-SK and LLO-KG strains in the presence and absence of mannose) and compared to M0 and M1 + /M2 + polarized J774.1 cells at 24 or 48 h post-incubation. As determined by flow cytometry, engineered *B. subtilis* LLO strains expressing TFs exhibited different patterns of J774A.1 surface marker expression when compared to the LLO strain (Fig. 5, Supplementary Fig. 8 and Supplementary Fig. 9). There was no difference in surface marker expression between *B. subtilis* strain 168 and the LLO strain at any time point. At 24 h post-incubation, a significant decrease in CD206 expression was observed in J774A.1 cells containing LLO-SK, with and without the addition of mannose ($p = 0.0048$; not shown on graph), compared to the LLO strain. The same trend was observed at 48 h (Fig. 5a, c). Conversely, the mean fluorescence intensity (MFI) for CD206 staining was

increased by LLO-KG, both with and without mannose ($p = 0.0599$; not shown on graph), at levels comparable to those of the positive control at 24 h (M2+). However, the elevated levels were not sustained as indicated by the 48 h time point. CD86 expression levels were the same in all bacterial conditions at 24 h in comparison to resting cells with no significant differences in comparison to the M1 control (M1+; Fig. 5b, d). At 48 h, the LLO-SK strain significantly increased CD86 MFI in comparison to the LLO strain with or without mannose added. The *B. subtilis* 168 strain showed a significant difference in CD86 MFI in comparison to untreated control but not in comparison to the LLO strain. Differences in CD206 and CD86 expression at 24 and 48 h post treatment were analyzed to reveal temporal changes of each treatment (Supplementary Fig. 8). CD206 expression was further increased in M2 + J774A.1 cells, whereas the LLO-KG with and without mannose were not able to sustain CD206 expression and showed a decrease at 48 h. CD86 expression was increased at 48 h in cell with the *B. subtilis* strain 168 and LLO-SK with, and without, mannose.

**Modulation of J77A.1 cell cytokine and chemokine expression using engineered *B. subtilis* LLO strains.** Functional readouts for macrophage polarization and resulting cell fate change include shifts in cytokine and chemokine expression[7,11,42]. Therefore, to characterize the effect of LLO-SK and LLO-KG on macrophage function, cytokine and chemokines produced by the J774A.1 cells were profiled when treated with the engineered strains relative to the LLO strain. *B. subtilis* strain 168 yielded similar marker expression profiles in comparison to the LLO strain and a ΔhlyA mutant (i.e. no LLO expression) of *Listeria monocytogenes* has been shown to either not change cytokine and chemokine profiles in a macrophage cell line, or to change profiles of these proteins equally to the wild-type *L. monocytogenes* in bone marrow derived macrophages[43,44]. Therefore, the 168 strain was not used in further comparisons, because the lack of LLO was not expected to impact J774A.1 profiles. LLO-SK and LLO-KG were shown to alter J774A.1 cytokine and chemokine expression patterns relative to the LLO strain as well as positive controls (Fig. 6 and Supplementary Fig. 10). Addition of LLO-SK and LLO-KG strains to J774A.1 cells led to differential expression of cytokines compared to the cytokine profile observed when the LLO strain was used, as shown by levels of interleukin (IL)-10, IL-12p40 and tumor necrosis factor alpha (TNF-α) (Fig. 6a–d, Supplementary Fig. 10). LLO-SK downregulated granulocyte colony stimulating factor (G-CSF) relative to both the LLO strain and LLO-KG (Fig. 6d, e). Although cytokine production was generally higher at 24 h compared to 48 h post-bacterial exposure for most cytokines and chemokines, significant differences were observed at both time points from LLO-SK and LLO-KG in comparison to the LLO strain, and the M1 and M2 positive controls.

Some of the selected proteins were not impacted by any condition, including vascular endothelial growth factor (VEGF). Several cytokines were only significantly changed at one of the two time points in certain conditions, including IL-6 and macrophage inflammatory protein-2 (MIP-2/CXCL2) (Fig. S10f–g, o, p). D-mannose alone altered the relative levels of certain cytokines including IL-1α and IL-1β; significant results were observed in treatment conditions when D-mannose was added and caused the same impact on certain cytokines (e.g. Fig. 6f and Supplementary Fig. 10c, d). MIP-1α production at 48 h was the only time point within concentration range of the standards even after dilution. IL-4 only appeared in the M2 + condition in which it was added. IL-13 exhibited the same trend of minimal expression at 24 h among all treatment conditions other than M2 + (Supplementary Fig. 10h, i). All *B. subtilis* LLO strains caused a significant change in most cytokine levels relative to the resting state and positive controls.

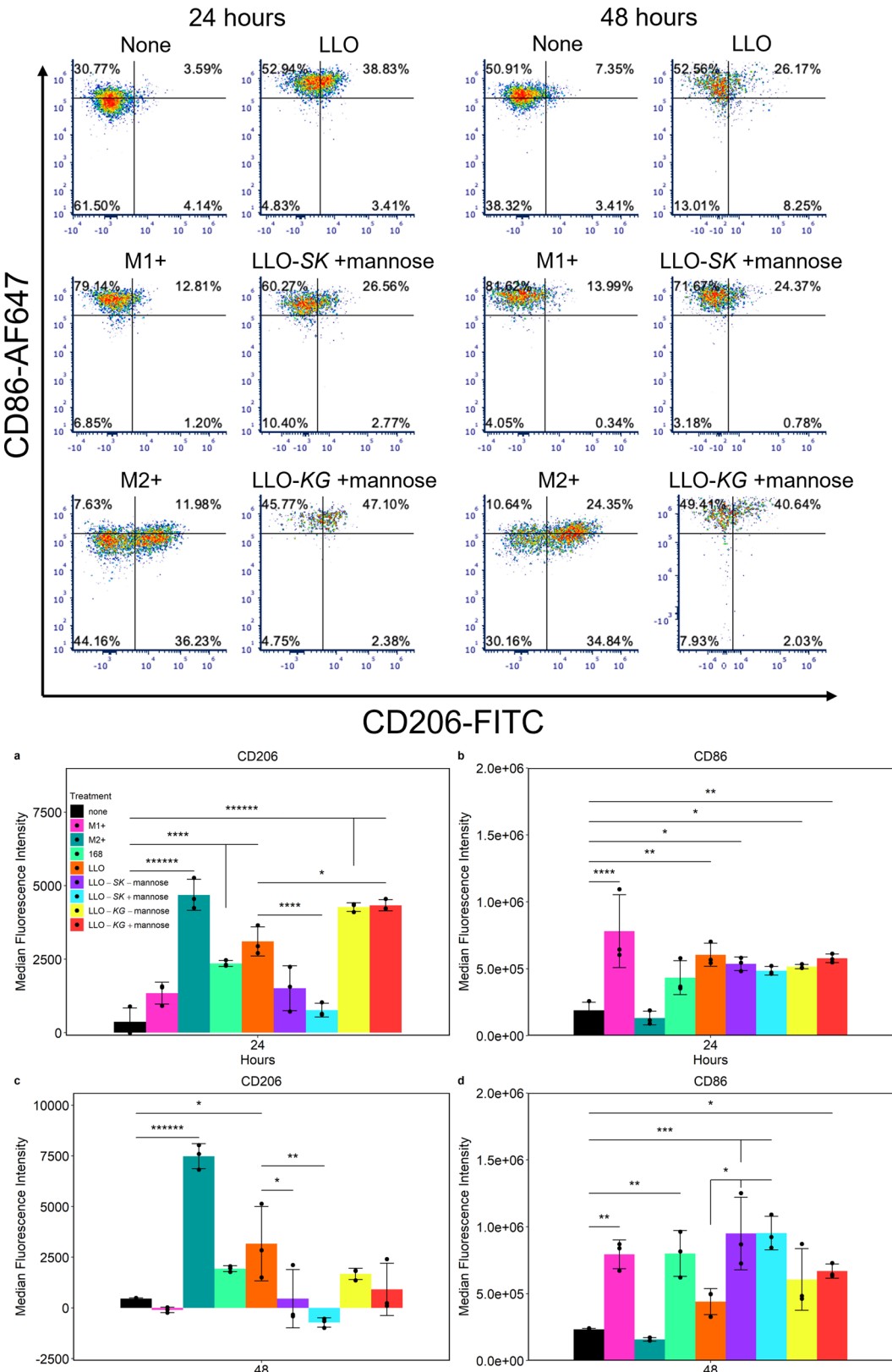

**Fig. 5 Flow cytometry demonstrated TF strain-mediated shifts of J774A.1 cell marker expression.** Flow cytometric analysis revealed changes in CD86 and CD206 induced by expression of transcription factors from the EES (top panel). These data are representative examples, taken as the median value of n = 3. Average CD86 or CD206 median fluorescence intensity (MFI) is compared among all treatments at each time point (lower panel, n = 3); (**a**) CD206 at 24 h, (**b**) CD86 at 24 h, (**a**) CD206 at 48 h, (**b**) CD86 at 48 h. J774A.1 cells were untreated (none), treated with LPS and IFN-γ (M1+), IL-4 and IL-13 (M2+), *B. subtilis* strain 168 (168), LLO strain (LLO), LLO-*SK* with and without mannose (LLO-*SK* -mannose, LLO-*SK* + mannose) and LLO-*KG* with and without mannose (LLO-*KG* -mannose, LLO-*KG* + mannose) at 24 and 48 h post initial treatment. Plotted data is mean ± SD; *p < 0.05, **p < 0.01, ***p < 0.001, *****p < 0.00001.

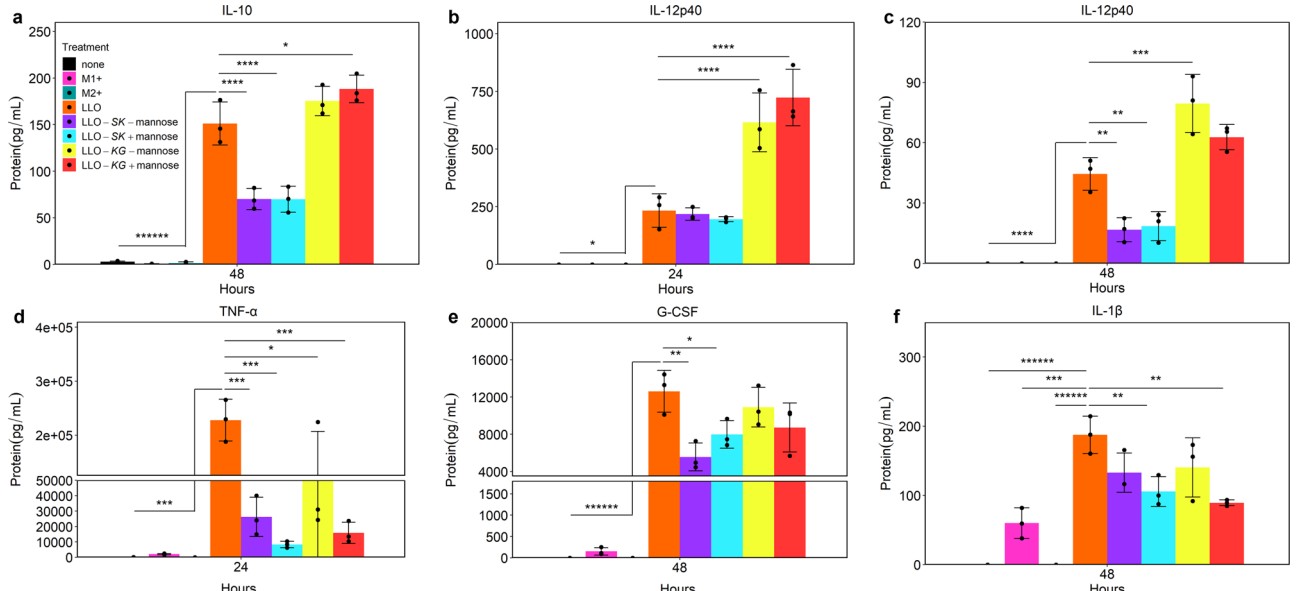

**Fig. 6 TF-expressing strains modulated cytokine and chemokine expression in J774A.1 cells.** Cytokine and chemokine protein concentrations were assayed by Luminex in untreated J774A.1 cells (none), and in cells treated with LPS and IFN-γ (M1+), IL-4 and IL-13 (M2+), LLO strain (LLO), LLO-*SK* with and without mannose (LLO-*SK* -mannose, LLO-*SK* + mannose) and LLO-*KG* with and without mannose (LLO-*KG* -mannose, LLO-*KG* + mannose) at 24 and 48 h-post initial treatment; (**a**) IL-10 at 48 h, (**b**) IL-12p40 at 24 h, (**c**) IL-12p40 at 48 h, (**d**) TNF-alpha at 24 h, (**e**) G-CSF at 48 h, and IL-1beta at 48 h. The most prominent protein changes are shown; 14 of the 17 proteins in the panel revealed gene-specific distinctions between the LLO-*SK* and LLO-*KG* strains. Plotted data is mean ± SD from $n = 3$ biological replicates; *$p < 0.05$, **$p < 0.01$, ***$p < 0.001$, ****$p < 0.0001$, ******$p < 0.000001$. Significance shown is comparing LLO treatment to all other treatments.

## Discussion

For the EES to impact macrophage function, the EES needs to access the cytoplasm by phagosomal escape and remain metabolically active to deliver protein to the host cell cytoplasm, which can then traffic to the nucleus. Induction of LLO expression allowed the LLO strain to maintain its morphology within the cytoplasm of the host cell[38] (Fig. 2, Supplementary Movie 1), indicating escape from phagosome-mediated destruction. After phagosomal escape, the LLO strain was observed to replicate in live J774A.1 cells indicating intracellular viability of the LLO strain (Fig. 3 right panel, Supplementary Table 1). Additionally, the J774A.1 cells were found to contain an average of 11 bacteria (the LLO strain) per cell with 99% of cells containing bacteria indicating effective uptake and persistence (Supplementary Table 1). The LLO strain delivering nuclear targeted β-gal protein to the nucleus was further evidence of bacterial and macrophage viability in the co-cultures (Fig. 4). The presence of viable LLO strain in the host cells did affect the host cell viability; without regulation of the LLO strain replication, the fate of host cells was eventually cell death, due to bacterial proliferation. This was apparent in 10–12% of J774A.1 cells at the final time point (4 h) with 25:1 MOI (Fig. 3). To address this problem, the cultures were treated with a high concentration of gentamicin (25 μg/mL), above the amount used to kill extracellular bacteria (4 μg/mL), for the purpose of eliminating the intracellular bacteria. To further develop an EES chassis, replication would need to be controlled to optimize the interaction between the EES and host cell. Genetic control of specific component of the EES replication machinery, could be used to control replication while still maintaining protein production; an optimized EES could include a genetic switch controlling an essential gene responsible for initiation of replication[45].

During immune responses, STAT-1 is part of the pro-inflammatory response and is a potent modulator that is directly upregulated by exposure to pro-inflammatory cytokines such as IFN-γ. This response was shown at 3 h in this study (Supplementary Fig. 7), while KLF6 has been shown to be upregulated in M1 polarized macrophages[33,34,37]. Therefore,

these two factors were used in LLO-*SK* to compare to IFN-γ and LPS in driving the pro-inflammatory phenotype. The activity of KLF4 is directly stimulated by the signal cascade in cells treated with the cytokine IL-4 and while this stimulation did not produce significant immunofluorescence at 3 h (Supplementary Fig. 7), this TF has been reported to promote M2 polarization. GATA-3 is also highly upregulated in M2 macrophages[35,36], which is the reason these two TF were used in LLO-*KG* to compare to IL-4 and IL-13 in driving the anti-inflammatory phenotype. Delivery of LLO-*SK* and LLO-*KG* strains to J774A.1 cells produced higher levels of these TFs in the nuclei than levels produced by the known signal cascade inducers (IFN-γ and LPS; IL-4 and IL-13). Accordingly, this result demonstrated the potential of *B. subtilis* LLO to impact host cell function within the 3 h timeframe of interaction between the LLO strains and host cells.

In this study macrophage plasticity was modulated using TFs expressed from the EES. Cell surface markers are commonly used to differentiate M1 and M2 polarization[7,11]. Cell surface expression levels of CD86 and CD206 were altered by the bacterial conditions, and the LLO-*SK* and LLO-*KG* were observed to regulate CD206 (Fig. 5) expression as expected based on the known activity of the expressed TFs. At 24 h, CD86 response was more complex as it is expected to be upregulated as a general response to bacteria. However, by 48 h the LLO-*SK* strain had increased expression significantly compared to the LLO strain (Fig. 5 and Supplementary Fig. 8). The *B. subtilis* 168 strain also increased expression between 24 and 48 h but not significantly relative to the LLO strain. The surface protein CD206 has been shown to recognize the surface carbohydrates of pathogens and be triggered by proteases produced by *B. subtilis*[46,47]. This would explain the increase in CD206 expression caused by the *B. subtilis* 168 strain and LLO strain in comparison to the untreated controls. Additionally, while CD86 is regulated directly by inflammatory responses to the engineered *B. subtilis* LLO, the plasticity of the macrophages could cause CD86 expression to change over time[11]. It has been suggested that products of *B. subtilis* such as

sublancin or exopolysaccharide (EPS), or the treatment of macrophages with *B. subtilis* spores can result in either M1 or M2 activation[48]. Even with these complexities, clear changes were observed that indicate that LLO-*SK* and LLO-*KG* impacted surface marker expression in comparison to the LLO strain.

Although changes in cytokine and chemokine expression vary along the spectrum of macrophage polarization, there are well-documented cytokines which are used to broadly identify M1 or M2 macrophages[7,11,49]. These identifying cytokines and chemokines are important when studying disease as they provide information on the function and characteristics of macrophage populations[42,50]. The M1 and M2 polarization classifications represent a trend in immune responses but is complex and contains some mixed signals, likely due to bacterial stimulation which can cause these classifications to be incomplete or oversimplified[49,51]. Complexities in macrophage activation, phenotype and plasticity were encountered in this study. In some instances, the LLO-*SK* and LLO-*KG* strains impacted the host cell as predicted in comparison to the LLO strain alone (Fig. 6 and Supplementary Fig. 10). The upregulation of IL-10 by LLO-*KG* and downregulation by LLO-*SK* was an expected result along with G-CSF being downregulated by LLO-*SK*[35,37,52,53]. The LLO strain increased IL-10 and G-CSF which are expected results because IL-10 and G-CSF have been shown to be produced in response to bacteria[53–55]. However, in some cases there were results that were not anticipated, and the pleiotropic effects of each selected TF need to be considered. The change in IL-12p40 levels is an example of an unexpected result based on known signaling cascades. IL-12p40 is known to be produced during inflammatory phenotypes by nuclear factor kappa light chain enhancer of activated B cells (NF-κB)[56]. Therefore, the increase in production of IL-12p40 by LLO-*KG* is unexpected; however, it is possible GATA-3 could affect IL-12p40 production. The IL-12p40 promoter has a canonical GATA binding site but GATA-3 has not been studied with this promoter[56]. The LLO-*SK* strain showed some downregulation of IL-12p40 at 48 h which could be due to STAT-1 driving alternative cytokine production such as IL-27p28[56,57]. TNF-α downregulation by LLO-*KG* is another expected result[35] but downregulation by LLO-*SK* in comparison to the LLO strain indicates further complexity and potential pleiotropic effects of STAT-1. This could be due to TNF-α being regulated by NF-κB[58–60]. Furthermore, there may have been metabolic impact on cytokine production due to the addition of D-mannose, which impairs glucose metabolism and is known to suppress the succinate-driven HIF1α activation of IL-1β leading to downregulation of IL-1β[61,62]. Accordingly, results indicated that D-mannose had a significant impact on IL-1β and other cytokines and chemokines, which is an important consideration for future EES studies that rely on sugar-inducible systems. Metabolic reprogramming is most likely playing a significant role in response to the chemical inducer of the engineered *B. subtilis* LLO TF operons and possibly the LLO strain alone[62].

Overall, dynamic responses were observed for many of the cytokines and chemokines in response to the LLO strain with the LLO-*SK* and LLO-*KG* strains able to modify expression further in comparison to the LLO strain in some key examples indicating TF specificity. One component of the differential responses were elevated expression of cytokines and chemokines in response to all bacterial strains compared to the positive controls of IFN-γ and LPS (M1+) or IL-4 and IL-13 (M2+). Additionally, the differences in responses to treatment with bacteria compared to the positive controls was larger in the cytokine and chemokine profiles than were observed with the cell surface markers. Enhanced signaling proteins (cytokines and chemokines) response from macrophages, and other professional antigen-presenting cells, compared to cell surface marker expression has

been previously observed[63–65]. Other factors that could contribute to the observed complex response involves *B. subtilis* stimulating Toll-like receptor 2 (TLR2) in contrast to IFN-γ and LPS stimulating TLR4[66]. Additionally, viable bacteria can cause a more dynamic response than just bacterial products alone[67]. Altogether, even with the dynamic responses from the J774A.1 cells in response to complex signals, the LLO-*SK* and LLO-*KG* strains were able to change trends in macrophage surface markers, and cytokine/chemokine profiles in comparison to the LLO strain. This observation is further supported by not observing significant differences in J774A.1 viability, percentage of cells containing fluorescently labeled bacteria and distribution of fluorescence intensity (relating to number of bacteria per cell) when comparing the LLO, LLO-*SK* and LLO-*KG* strains (Supplementary Fig. 6). Future studies should focus on characterizing the impact of these strains in primary macrophages such as bone marrow-derived macrophages to clearly understand application potential. In addition, more TF pairings informed by thorough studies that elucidate the complexity of macrophage response should be constructed and tested to optimize the response from the macrophages for the desired application.

Controlling inflammation could be an important biomedical application of the LLO-*SK* and LLO-*KG* strains. For example, inflammation is an important hallmark of cancer, and the phenotype of tumor-associated macrophages (TAM) is thought to promote tumor growth, metastases and poor outcomes[68]. TAMs are broadly M2 polarized, and IL-10, IL-12p40 and G-CSF all have been shown to play important roles in impacting the tumor microenvironment through regulation of TAMs[68–70] which could be targeted by the LLO-*SK* for cancer bacteriotherapy[26]. Arthritis represents immune polarization where homeostasis is driven to a pro-inflammatory condition[71]. Here, modulating macrophages towards the M2 phenotype could reduce inflammation in joints. Downregulation of TNF-α and upregulation of IL-10 LLO-*KG* shows promise for treating damaging inflammatory conditions such as arthritis[54,71]. Manipulation of immune cells in vivo has been characterized by low efficacy and lack of innate control[72] and the use of EES could circumvent these issues since engineered bacteria can be taken up by phagocytic cells and direct gene expression towards a therapeutic phenotype.

The use of an EES is advantageous when compared to alternative methods of manipulating mammalian cell fates and function. Current methods of manipulating mammalian cell fate include viral vectors, growth factors or signaling molecules[73]. Additionally, chimeric antigen receptor T (CAR-T) cells and CRISPR technologies have proven to be potential futures for some therapeutics and are clinically relevant, but each has some limitations[74]. Viral vectors have been shown to be slow as therapeutics within the immune system, specifically in targeting and modulating macrophages, compared to exogenous cytokines[75]. An alternative method of manipulating cellular fates using prosthetic networks increases the variety of cargo that can be delivered and provides some control with limitations[73]. The EES can build on the precedence of prosthetic networks by having the capability to be constructed to generate complex sets of proteins and molecules once in the cytoplasm of mammalian cells for improved control; continuously supplying TFs, or alternatively providing a method for delivering CRISPR-Cas9[76,77], which could improve directing cell fates.

Studies of pathogens and their virulence factors should inform future development of the EES. The EES will utilize this characterization for defined control within the host cell cytoplasm. This study demonstrated the utility of an EES to alter mammalian cell fates. The use of the EES as a tool to change mammalian cell function may have use in the treatment of diseases by altering the function of mammalian cells. *B. subtilis* serves as an ideal chassis

for the development and optimization of the EES capable of surviving in the cytoplasm and delivering proteins to the nuclei of mammalian cells to alter cellular fates.

## Methods

**B. subtilis LLO constructs.** Constructs were inserted into the genome of *B. subtilis* LLO at the *amyE* locus using a homologous recombination plasmid (pDR111[78], a gift from Dr. Lee Kroos). The pDR111 plasmid was transformed into *B. subtilis* using a natural competence protocol and constructs were selected by spectinomycin then confirmed by PCR amplification out of the genome[79]. *B. subtilis* expressing IPTG-inducible LLO was provided by Dr. Daniel Portnoy. The constructs include the *lacZ*, *Stat-1Klf6* and *Klf4Gata-3* genetic cassettes. *B. subtilis* LLO was designed to secrete β-gal to the nucleus through the twin-arginine translocation (Tat) pathway by synthesizing the PhoD signal peptide[40] and the amino acids from 126-132 of the simian virus (SV) 40 nuclear localization signal (NLS)[41] together and connecting both to *lacZ* when inserting into pDR111 using Gibson assembly after *lacZ* was amplified from the pST5832 plasmid (a gift from Carolyn Bertozzi & Jessica Seeliger, Addgene plasmid #36256). The same construct was engineered without the SV40 signal to confirm specific delivery to the nucleus. The *lacZ* gene and the synthesized PhoD signal peptide plus SV40 NLS were cloned into the NheI restriction site in pDR111 using Gibson assembly. Initially, the β-gal secretion strain was controlled by an IPTG-inducible promoter (Phyper-spank). However, previous studies have shown the IPTG system is limited in controlling protein production so these constructs were engineered to be controlled by the mannose-inducible system amplified from *B. subtilis* ZB307 strain genome[19]. The mannose promoter and regulator were cloned into the pDR111 plasmid to replace the Phyper-spank promoter and LacI regulator using Gibson assembly. Accordingly, the *lacZ* gene with the same design was cloned in the NheI restriction site present in the pDR111 mannose plasmid using Gibson assembly. The *Stat-1* and *Klf6* genes were synthesized by IDT as a custom gene and Gblock, respectively, from the coding sequences obtained from Uniprot. The *Stat-1Klf6* operon was fused by ligation at an introduced EagI restriction site between the genes during cloning into the pDR111 mannose plasmid by restriction cloning in the SalI and NheI restriction sites. The *Klf4* and *Gata-3* genes were synthesized as Gblocks from IDT and fused using the same method as the *Stat-1Klf6* operon. The *Klf4Gata-3* operon was cloned into the pDR111 mannose plasmid using restriction cloning at the SalI and SbfI restriction sites after the SbfI cut site was introduced into the multiple cloning site by inverse PCR then digesting both ends with SbfI and re-ligating the pDR111 mannose plasmid. All constructs were confirmed by restriction digest, sequencing and functionality tests. A list of primers, Gblocks and custom genes are included in Supplementary Data 1.

**Growth conditions for B. subtilis 168, B. subtilis LLO and engineered strains.** *B. subtilis* LLO was grown under the same conditions for all experiments. Each *B. subtilis* LLO construct was grown in Luria-Bertani Miller broth (LB) with the appropriate antibiotic. *B. subtilis* LLO was grown in LB with chloramphenicol (10 μg/mL) and all constructs that were integrated into the *amyE* were grown with spectinomycin (100 μg/mL). The overnight cultures were grown for 16 h at 37 °C and 250 RPM. All constructs were integrated into the genome of *B. subtilis* LLO which allowed for expression of constructs without antibiotics during co-incubation with J774A.1 cells. *B. subtilis* 168 (ATCC-23857, ATCC, Manassas, VA, USA) was grown in LB with no antibiotic selection overnight for 16 h at 37 °C and 250 RPM.

**Delivery protocol for B. subtilis LLO and engineered strains.** The following conditions were utilized to induce *B. subtilis* LLO delivery, unless otherwise described. J774A.1 monocyte/macrophage cells (ATCC-TIB-67, ATCC Manassas, VA, USA) were maintained at 37 °C and 5% $CO_2$ in DMEM (ThermoFisher, MA, USA), supplemented with 10% fetal bovine serum (FBS). Cells were tested negative for mycoplasma using the MycoAlert PLUS Mycoplasma Detection Kit (Lonza, USA). Once cells were confluent, they were seeded onto a plate or 4-well chambered imaging slide and allowed to adhere overnight (described below) when an estimation of total number of cells was made based on confluency. *B. subtilis* LLO was added at an introduced MOI of 25:1 for all experiments besides testing of host cell viability, along with IPTG (500 μM) to induce expression of LLO with or without protein of interest. *B. subtilis* LLO and J774A.1 cells were then co-incubated at 37 °C and 5% $CO_2$ for 1 h J774A.1 cells were then washed three times with PBS and new medium was added containing gentamicin (4 μM) to eliminate any remaining extracellular *B. subtilis* LLO. Co-incubation continued for 3 h at 37 °C and 5% $CO_2$ prior to imaging or preparation for microscopy (described below) (Supplementary Fig. 1).

**Antibody staining for B. subtilis LLO phagosomal escape.** After fixation with 4% paraformaldehyde (PFA), cells were permeabilized using 0.3% Triton X-100 (ThermoFisher) followed by a blocking step containing 0.3% Triton X-100 and 5% normal goat serum (ThermoFisher, cat# 31872,). *B. subtilis* LLO location was determined by incubating a rabbit anti-subtilisin antibody (1:50, Antibodies-online, PA, USA, cat# ABIN958907) at 4 °C overnight followed by a goat anti-rabbit IgG

Dylight 650 (1:4000, Novus, CO, USA, cat# NBP1-76058) secondary antibody at room temperature (RT) for 2 h. Phagosome formation or destruction was shown by incubating an anti-Lamp-1[38] antibody (1:100, AbCam, MA, USA, cat# ab25245) at 4 °C overnight followed by a goat anti-rat IgG Alexa Fluor 555 (1:1000, Invitrogen, CA, USA, cat# A-21434) secondary antibody at RT for 2 h. Nuclei were counter-stained by incubating cells with Hoechst 33342 (1 μg/mL) for 10 min at RT. Membranes of the cells were stained by incubating cells with PKH67 green-fluorescent cell linker kit (10 μM, Sigma MO, USA) for 10 min at room temperature. Slides were then coverslipped using Fluoromount-D mounting media (Southern Biotech, AL, USA). Slides were imaged using confocal microscopy (described below).

**Confocal imaging.** Confocal microscopy was performed using a Nikon A1 CLSM (Nikon, NY, USA) microscope to determine *B. subtilis* LLO escape from the phagosome complex by imaging J774A.1 cells that had been treated with the *B. subtilis* LLO with and without IPTG. Imaging was performed using a 60x oil objective and 1.5x zoom and using filter sets for DAPI (Hoechst 33342), GFP (PKH67), TRITC (Alexa Fluor 555 for LAMP-1) and Cy5 (Dylight 650 for *B. subtilis*). Z-stacks were taken at 0.5 μm steps to confirm location of *B. subtilis* LLO within host cells. Images were analyzed using NIS-Elements AR Software (Nikon) and background noise was reduced by using Nikon denoise.ai algorithm. The three-dimensional volume images and cutaways were produced by the Alpha display mode. The Alpha display mode was also used to generate three-dimensional videos to display z-depth location of the *B. subtilis* LLO during escape from the phagosomes.

**J774A.1 viability by MTS assay and Flow cytometry and uptake rate of engineered B. subtilis LLO strains.** The effect of *B. subtilis* LLO on J774A.1 cell viability was determined after EES delivery. Conditions examined were: multiple time points of interaction between *B. subtilis* LLO and host cells (0, 1, and 4 h), different MOIs (10:1, 25:1, 50:1), no IPTG induction and no treatment, with biological triplicates ($n = 3$) for each time and condition. At each time point, J774A.1 cells were washed once with PBS then MTS reagent (Abcam) was added to cells at a 10-fold dilution with DMEM followed by incubation for 30 min at 37 °C then absorbance was measured at 490 nm. All treatment conditions were compared to J774A.1 cells alone to elucidate any differences in loss of viability of the J774A.1 cells due to the treatment conditions. Flow cytometry was used to analyze cell viability after engineered *B. subtilis* delivery at 10:1, 25:1 and 50:1, compared to no IPTG induction and no treatment at the 4 h time point ($n = 1$) or after the addition of LLO, LLO-SK and LLO-KG (all at 25:1 MOI with IPTG; $n = 3$) compared to no treatment. Furthermore, uptake of *B. subtilis* LLO, LLO-SK and LLO-KG by J774A.1 cells was assessed after staining bacteria with CellTracker Orange CMRA Dye (CTO, Invitrogen, C34564, 2 μM incubated at 37 °C and 250 RPM for 25 min). Following staining, the strains were washed three times before adding to J774A.1 cells at 25:1 MOI for 4 h incubation. Cells were collected, washed once with 1X PBS and incubated with Zombie NIR viability dye (1:750, Biolegend, San Diego, CA, USA; Cat# 423105) in PBS for 20 min, at 4 °C in the dark. Cells were washed twice followed by fixation using 4% PFA and resuspended in 100 μL flow buffer for analysis using the Cytek Aurora Cytometer (Cytek Biosciences, CA, USA). All samples were assessed for percent live cells. Cells which were incubated with CTO bacteria were assessed for percent CTO positive cells (indicating J774A.1 cells containing bacteria) and CTO MFI was used as a relative measure of CTO bacteria per cell. Standard one-way ANOVA with Tukey post-hoc test was used to determine statistically different values.

**Live Cell Imaging.** *B. subtilis* LLO was internalized into J774A.1 cells as described above, using a 96-well black glass-bottom plate (40,000 cells/well; Greiner Bio-One, Austria, cat# 655892). *B. subtilis* LLO was stained with CellTracker Orange CMRA Dye (CTO, Invitrogen, C34564) as described above then added to J774A.1 cells. Live cell imaging was performed on a Leica DMi8 Thunder microscope equipped with a DFC9000 GTC sCMOS camera and LAS-X software (Leica, Wetzlar, Germany). Cells were maintained at 37 °C and 5% $CO_2$ in Fluorobrite medium during the imaging session. Fluorescent images of CTO were acquired using a TRITC filter set. Brightfield and fluorescent images were acquired consecutively, using a 63x oil objective every 1 h beginning at 1 h post co-incubation and continuing until 4 h post-incubation. Z-stacks were taken at all time points at 0.4 μm steps to confirm *B. subtilis* LLO presence within cytoplasm. *B. subtilis* LLO presence in J774A.1 cells was quantified using Fiji (ImageJ) software and cell counter plugin by counting >1.5 μm rods throughout z-depth. An area of 2090 μm by 1254 μm in each well was imaged and used to perform this quantification.

**Engineered B. subtilis LLO β-gal protein secretion.** *B. subtilis* LLO engineered to secrete β-gal (LLO-lacZ) with and without a NLS (LLO-*lacZ*-NLS and LLO-*lacZ*-no NLS) were internalized into J774A.1 cells as described above, using 4-well chambered slides (75,000 cells/well, ThermoFisher, cat #154917). Incubation was also performed using the supernatant from a 16 h mannose-induced LLO-*lacZ*-NLS culture. The second incubation step was 3 h and began after adding gentamicin (4 μg/mL) and with and without D-mannose (1% w/v) for the β-gal secretions strains. During extended studies, the cells were washed gently 3 times with 1X PBS

then a high concentration of gentamicin (25 µg/mL) was added after 3 h and incubation continued for an additional 21 h. The cells were then fixed with 4% PFA for 10 min prior to preparation for microscopy (described below).

**β-gal antibody staining**. After fixation with 4% PFA, cells were permeabilized using 0.3% Triton X-100 (ThermoFisher) followed by a blocking step containing 0.3% Triton X-100 and 5% normal goat serum (ThermoFisher). β-gal secretion to the nucleus was identified using fluorescence imaging after incubation with an anti-β-galactosidase (*E. coli*) antibody-rabbit (1:100, Biorad, CA, USA, cat# AHP1292GA) at 4 °C overnight followed by a goat anti-rabbit IgG Dylight 650 (1:3000, Novus) secondary antibody at RT for 2 h Nuclei were counterstained by incubating cells with Hoechst 33342 (1 µg/mL) for 10 min at room temperature. Slides were then coverslipped using Fluoromount-D mounting media (Southern Biotech, AL, USA). Epi-fluorescent microscopy was performed using a Nikon Eclipse Ci-L microscope equipped with a CoolSNAP DYNO camera for fluorescent imaging and NIS elements BR 5.21.02 software (Nikon). Images were acquired with a 40x phase contrast objective and for fluorescent imaging DAPI and Cy5 filter sets were used. Nuclear SNR was quantified by imaging in random areas in each corner of the well and in the center of the well. At least 5 random images totaling an area of 1085 µm by 825 µm were utilized in drawing regions of interest (ROIs) around nuclei of J774A.1 cells to quantify Cy5 fluorescence by utilizing Hoechst 33342 counterstain to determine nuclei location. Nuclear SNR was calculated by using mean fluorescence intensity of ROI region around nuclei of J774A.1 cells in each of the five random images ($n = 50$ random individuals) divided by standard deviation (SD) of noise in the well. SD of noise in well was determined by drawing 3 ROIs (550 µm$^2$) in background areas of each image then calculating SD of the mean fluorescence from all ROIs drawn in background areas. Statistics was determined using Brown-Forsythe and Welch ANOVA with Dunnett T3 post-hoc test and all treatment conditions were compared.

**Engineered *B. subtilis* LLO modulation of J774A.1 cell protein expression**. The LLO strain, LLO-*SK* and LLO-*KG* were internalized into J774A.1 cells as described above (*B. subtilis* LLO β-gal protein secretion), using a 6-well plate (Corning Costar #3516). TFs were secreted for 3 h then allowed to be trafficked for an additional 21 h as described above (*B. subtilis* LLO β-gal protein secretion). For flow cytometry, Accutase (Sigma, cat# A6964) was used to detach J774A.1 cells for analysis (described below). For Luminex cytokine profiling (Millipore Sigma, MA, USA), the supernatant was removed at both 24 and 48 h and then analysis was performed to quantify cytokines produced (described below). Non-stimulated J774A.1 cells were assumed to be at resting state. J774A.1 cells were polarized with interferon-gamma (IFN-γ) and lipopolysaccharide (LPS) (M1+, 100 ng/mL each) or inter-leukin (IL)-4 and/or IL-13 (M2+, 100 ng/mL each) to be used as positive controls. Furthermore, J774A.1 cells were treated with the *B. subtilis* 168 strain in flow cytometry characterization to compare to LLO strain. The EES with operons were treated with and without mannose as described in *B. subtilis* LLO β-gal secretion and the LLO strain was not as no difference in impact on J774A.1 cells was observed in previous flow experiments. All treatment conditions were performed in biological triplicates ($n = 3$).

**Growth curves**. *B. subtilis* LLO, LLO-*SK* and LLO-*KG* overnight cultures were grown for 16 h at 37 °C and 250 RPM in triplicate ($n = 3$). All cultures were diluted 1:20 then allowed to grow into logarithmic phase for 3 h Subsequently the cultures were normalized to OD$_{580}$ = 0.1 then 100 µL was transferred into columns of a 96 well plate (Falcon, #351172) for a total of 24 replicates ($n = 3$ biological, $n = 8$ technical). Cultures were grown in a PerkinElmer VICTOR Nivo plate reader at 37 °C and 300 RPM with OD$_{580}$ measurements (580 nm was used due to plate reader limitations) taken every 30 min. Measurements were performed until growth rates began to slow in late logarithmic phase in the 100 µL (9 h for all samples). All replicates were plotted to visualize differences in growth rates between strains as mean ± SD.

**Flow cytometry**. After addition of the engineered *B. subtilis* LLO strains and controls, followed by incubation for 24 or 48 h, cells were collected and stained in a 96-well round bottom plate. Treatment addition was staggered so that flow staining and data acquisition could be performed at the same time for 24 and 48 h cells. All staining steps were performed in 100 µL volume at 4 °C in the dark. Samples were first incubated with Zombie NIR viability dye (1:750, Biolegend) for 20 min. Cells were washed once with flow buffer, followed by incubation with TruStain FcX™ PLUS (anti-mouse CD16/32) Antibody (Biolegend, Cat#156603; 0.25 µg/sample) for 10 min. Alexa Fluor® 647 anti-mouse CD86 Antibody (0.125 µg/sample; Biolegend; Cat#105020) and FITC anti-mouse CD206 (MMR) antibody (0.1 µg/sample, Biolegend; Cat#141703) were then added and incubated for 20 min. Cells were washed twice with flow staining buffer and fixed with 4% PFA for 10 min and resuspended in a final volume of 100 µL for flow cytometry analysis using the Cytek Aurora spectral flow cytometer (Cytek). Single stained controls and unstained controls for all conditions were used to assess fluorescent spread and for gating strategies. Flow cytometry data was analyzed with the software FCSExpress (DeNovo Software, CA, USA). A standard one-way ANOVA with Tukey's multiple

comparisons test was used to determine statistically different MFI values amongst all groups within each time point. A two-way ANOVA with Šidák's multiple comparisons test was used to compare the 24 and 48 h data for each surface marker. The data presented herein were obtained using instrumentation in the MSU Flow Cytometry Core Facility. The facility is funded in part through the financial support of Michigan State University's Office of Research & Innovation, College of Osteopathic Medicine, and College of Human Medicine.

**Luminex cytokine profiling assay**. Cell culture supernatant was stored at −20 °C until ready for use. Supernatant was analyzed for CCL2 (MCP-1), CCL3 (MIP-1a), CXCL2/MIP-2, G-CSF, IL-1α, IL-1β, IL-4, IL-6, IL-10, IL-12p40, IL-12p70, IL-13, TNF-α and VEGFα cytokine expression. Cytokine levels of cell supernatants were measured using a MCYTOMAG-70K-17 Mouse Cytokine Magnetic Multiplex Assay (Millipore Sigma) using a Luminex 200 analyzer instrument (Luminex Corp, USA) according to the manufacturer's instructions. Standard one-way ANOVA with Tukey post-hoc test was used to determine statistically different values amongst all treatment groups.

**ICC confirming EES manufacturing and delivering TFs**. Protocol for engineered *B. subtilis* LLO modulation of J774A.1 cell protein expression was used as the template protocol for this experiment. The experiment was performed using a 96-well black glass-bottom plate (40,000 cells/well; Greiner Bio-One). However, J774A.1 cells were fixed at 3 h post mannose addition to reveal engineered *B. subtilis* LLO delivering TFs to host cells using antibodies against the transcription factor of interest. Each transcription factor was stained for with the appropriate antibody individually within the wells. The strains was stained prior to addition to host cells using Invitrogen CellTracker Orange CMRA Dye (Invitrogen) as in live cell imaging. After fixing, cells were permeabilized using 0.3% Triton X-100 (ThermoFisher) followed by a blocking step containing 0.3% Triton X-100 and 5% normal goat serum (ThermoFisher). Cells were then incubated with anti-*Stat-1* (1:50, MyBiosource, CA, USA, cat# MBS125754), rabbit anti-*Klf6* (1:50, MyBiosource, cat# MBS8307089), rabbit anti-*Klf4* (1:20, MyBiosource, cat# MBS2014661) or rabbit anti-*Gata-3* (1:100, MyBiosource, cat# MBS8204267) at 4 °C overnight followed by a goat anti-rabbit IgG Dylight 650 (1:3000, Novus) secondary antibody at RT for 2 h Nuclei were stained by incubating cells with Hoechst 33342 (1 µg/mL) for 10 min at room temperature. Membranes of the cells were stained by incubating cells with PKH67 green-fluorescent cell linker kit (10 µM) for 10 min at room temperature. Imaging was performed using Leica DMi8 Thunder microscope equipped with Leica DFC9000 GTC sCMOS camera and Leica LAS-X software (Leica) with the following light emitting diodes for excitation: Hoechst 33342 (395 nm), PKH67 (475 nm), CellTracker Orange CMRA Dye (555 nm) and Dylight 650 for TFs (635 nm) filters were used and imaged using a 40X objective to confirm LLO-*SK* and LLO-*KG* delivery of TFs. Images were quantified using Fiji (ImageJ). Thresholding was performed on the Hoechst 33342 images to identify the nuclei. Nuclear ROIs were used to quantify fluorescence intensity in the Dylight 650 channel. Nuclear SNR was calculated using mean nuclear fluorescence/standard deviation of the background signal. Background signal was quantified from one data set for each TF. A mean of 158.4 ± 63.6 (SD) cells were analyzed per data set. Outliers were identified using the ROUT methods ($Q = 1\%$) and outliers removed for analysis using a one-way ANOVA and Tukey's multiple comparisons test.

**Statistical analysis and reproducibility**. Statistical analyses were performed using Prism software (9.2.0, GraphPad Inc., La Jolla, CA). Statistical tests are identified for each method. Plotting was performed using R version 4.0.4 with the following packages: ggplot2, dplyr, reshape2, ggsignif, plotrix and ggpubr. Data are expressed as mean ± standard deviation; $p < .05$ was considered a significant finding. All experiments were piloted to optimize the new methods then three independent biological replicates were used unless otherwise specified to examine reproducibility of data. For fluorescent quantification experiments, appropriate representative individuals were quantified using random imaging and quantification methods as to avoid any biases in data acquisition and quantification.

**Reporting summary**. Further information on research design is available in the Nature Research Reporting Summary linked to this article.

## Data availability

All raw data and *B. subtilis* LLO constructs will be made available upon request by the corresponding author. Plasmids used to produce *B. subtilis* LLO constructs are submitted to Addgene with the following IDs: 188396, 188397, 188398, 188399, 188400, and 188401.

## Code availability

All R scripts were written with a general format appropriate for the openly available, established packages mentioned above and can be made available on request.

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

## Acknowledgements

The authors would like to thank Dr. Daniel A. Portnoy for the *B. subtilis* LLO strain, Dr. Jens Schmidt for advice on imaging and Dr. Lee Kroos for *B. subtilis* plasmid pDR111. The authors would also like to thank Dr. Melinda Frame of the Michigan State University Center for Advanced Microscopy for confocal imaging, Dr. Matthew Bernard of the Michigan State University Flow Cytometry Core for flow cytometry aid and Luminex cytokine profiling analysis. The authors would like to acknowledge Chima V. Maduka for his thoughtful and thorough review of the manuscript and providing crucial insights. The authors would like to acknowledge the James and Kathleen Cornelius Endowment for financially supporting this research. Figure 1 and Supplementary Fig. 1 were created using BioRender.com.

## Author contributions

C.S.M. conceptualized *Bacillus subtilis* as a chassis organism, developed all the EES constructs as the platform technology, jointly developed and performed all experiments, jointly developed and analyzed all data to make figures and was one of the primary authors of the manuscript. A.V.M. jointly developed and performed all experiments, jointly developed and analyzed all data to make figures and was one of the primary authors of the manuscript. E.M.G. significantly contributed to the development of the EES platform technology, jointly developed and significantly contributed to the writing and editing of this manuscript. J.W.H. acquired the *B. subtilis* LLO strain from Dr. Daniel A. Portnoy, significantly contributed to the development of the EES platform technology and significantly contributed as an author to this manuscript. C.H.C. conceptualized the initial concept of the EES, supervised the studies, contributed to the experimental design, provided the resources, reviewed data and edited the manuscript.

## Competing interests

The authors declare no competing interests.
