## [Peer Review File · Communications Biology]

Reviewers' comments:

Reviewer #1 (Remarks to the Author):

Madsen and Makela et al report *Bacillus subtilis* strains engineered to secrete transcription factors with the goal of using these strains to control gene expression in mammalian cells. The creation of these transcription factor(TF)-secreting variants and the demonstration that they can deliver transcription factors into the nuclei of macrophages is quite a feat and has exciting potential. The strain development work is interesting, and the experiments are well performed. However, their data for functional readouts of how the TF-secreting strains impact M1/M2 polarization (Fig 5 – mac surface markers and 6 -mac cytokine production) suggest that these strains are not functioning as anticipated. Rather, their data suggest that the EES/LLO chassis (i.e., non-TF secreting) causes a marked innate immune response in macrophages with cytokine responses that dwarf that of polarized M1 and M2 macrophages. This finding means it is difficult to interpret whether the observed changes in macrophage-derived cytokine levels and CD86/CD206 expression between EES and TF-secreting EES actually represent controlled regulation of gene expression or simply changes in the innate response to the EES. As such major revisions are necessary, but overall, with these changes, I believe this paper could make an important contribution to the field.

Major:

Central to the author's manuscript is the idea that gene expression can be modulated by EES and as proof of principle the authors attempt to use their TF-secreting EES to polarize macrophages toward M1 or M2. However, the authors find that EES treatment alone causes macrophages to produce various cytokines (e.g., IL-10, IL-12p40, TNF, G-CSF, IL-1b) at levels, well above their positive controls for macrophage polarization. In some cases (e.g., TNF and G-CSF) cytokine levels are multiple orders of magnitude above the control. The author's discussion of this important finding is relatively cursory, and they should elaborate on what they think is occurring. It seems likely that this is an innate response to EES/LLO infection. Does this occur because the strain is escaping the phagosome? Have the authors compared the cytokine response/immunophenotype (CD86, CD206) of macrophages infected with EES +/- LLO?

The TF-secreting strains reduce or further increase macrophage cytokine production in some cases, however, in general, the changes do not appear to align with the predicted outcome for the EES-SK and EES-KG i.e. polarization toward M1 versus M2. Changes in CD86 and CD206 levels also do not appear to align the cytokine production levels is also suggestive of an alternate explanation. Although I found the author's discussion of their hypothesis relating to EES-SK and EES-KG effects on macrophage polarization to be somewhat unclear and this should be remedied. This is important given that "specificity" is key to how the authors state they would like EES should operate. Specifically, Line 180 – EES-SK or KG - the authors should provide the rationale for why they chose these TF pairs to engineer into EES in the

results i.e. SK because presumably, they hypothesized this would polarize the macs to M1 and KG combination would polarize to M2 and whether or no the data was consistent with that hypothesis. They should also write out the SK and KG transcription factor pairs in full the first time they are used before describing the results of the TF immunofluorescence in Fig S4.

Whether these macrophage responses represent plasticity or a host response also merits some additional investigation. In the experiments shown in Fig 5 and 6, were EES-SK and EES-KG as infectious as the parental strain? Did macrophage survival change in response to the different EES strains? Did both strains reach the same titers as the parental? Related, the authors should report whether there was any metabolic burden (e.g. perform growth curves) for the introduced circuits and TFF production for EES-SK and EES-KG. Answers to these questions would help to resolve whether the authors are observing controlled changes of macrophage polarization as they predict versus simply changes in the innate immune response to EES/LLO infection.

Language throughout this manuscript relating to the ability of the strain to modulate macrophage gene expression and plasticity in a specific and predictive way such as Line 252 “In this study macrophage plasticity was modulated...” should be toned down or supported with further experiments as discussed above.

Throughout the manuscript the phrase “modulation of gene expression changes” or variants thereof is used to refer to the effect of EES delivered transcription factors. However, at no point is gene expression measured. Rather readouts are performed at the protein level. The authors should either measure transcript levels of their target genes or re-phrase accordingly.

Minor:

In general, the Results are quite well written, however, the authors should add some description of each experimental setup. While it is acknowledged that the experimental details are present in the methods, an abbreviated description at the beginning of each Results subsection would aid the reader in understanding what was done without having to flick back and forth between the methods and results. Moreover, describing the rationale within the Results section would help the reader follow the author’s line of reasoning for the experiments being performed.

Line 124: EES term. In the intro, the authors use “EES” broadly to refer to engineered endosymbionts but elsewhere in the manuscript, starting in the Results, it is also used to refer to the engineered *B. subtilis* LLO strain. The authors should distinguish between the two for clarity. I would suggest reserving EES as a general term a new name used for the strain/platform developed in this manuscript.

Line 154 Can the authors discuss whether the anti420 subtilisin antibody stains viable B subtilis exclusively? If it does not, I think a gentamicin protection assay (or similar) is warranted to demonstrate that the intracellular bacteria are indeed viable as the authors claim.

Line 167 "high concentration of gentamicin" – did this effectively kill the intracellular bacteria?

A schematic of the synthetic operons introduced into EES would help the reader better understand these circuits and their regulation. If the figure limit is an issue, perhaps this could accompany figure 1.

Typographical:

Line 405: (Lonza, xx).

Reviewer #2 (Remarks to the Author):

In this study, the authors attempt to control macrophage polarization using engineered bacteria. First, they engineer B. subtilis such that, upon being endocytosed by macrophages, can escape the phagosome and survive in the cytoplasm of the macrophages. Then, they evaluate whether these B. subtilis can express proteins that can be imported into the nucleus of the macrophage. Finally, they attempt to influence macrophage polarization by engineering B. subtilis to express cell fate-determining transcription factors.

While the proposed idea is intriguing, I find that the majority of the data does not support the conclusions the authors want to make. In general, it seems that most of the effects shown are driven by the mere presence of B. subtilis instead of TF expression or addition of the inducer mannose. Moreover, when gene- and mannose-dependent effects are observed, it is sometimes in the opposite direction that would be expected (e.g. Fig 6B and C). While the authors acknowledge some of these irregularities in the discussion section, the rest of the paper is mostly written as if these irregularities did not exist. For example, in line 205 the authors claim that "EES-SK and EES-KG were shown to modulate mammalian gene expression as determined by cytokine profiling (Fig 6)", but don't acknowledge in the same paragraph that most of these cytokines are released in the presence of EES that does not express any TFs. A few other examples:

- Fig 2: It is hard to see colocalization of red and purple spots (or the lack thereof). This is less of a problem in the SI movies but it is still not great. At the very least, I recommend using less similar colors.

- Table S1: IPTG does not seem to make a significant difference in EES/J774A.1 and % infected, therefore I would not conclude that expression of LLO is what is driving EES numbers or replication.

- Figure S4: expression of KLF6, KLF4, and GATA-3 are visible in the negative controls, and thus it is hard to conclude that the EES or addition of manose have any influence in expression of these regulators. Quantification of fluorescence levels may partially alleviate this issue.

- Figure S5: In panel A, are the thresholds chosen to represent some type of transition to the M1 or M2 states? Why do the negative controls start with high levels of CD86? Why does the M1+ positive control also have high CD206? It is hard to asses whether the samples under evaluation are doing anything if the controls are not behaving as expected.

- Figure 6: Why do the M1+ and M2+ positive controls not show any signal?

In summary, it is hard for me to recommend publication of this paper in its current form. I suggest the authors to substantially revise their manuscript and be more forthcoming about the data irregularities.

One last point about line 263: *B. subtilis* 168 absolutely does sporulate, and it is frequently used as a model of sporulation. See, for example, <https://www.ncbi.nlm.nih.gov/pmc/articles/PMC178487/>

Reviewer #3 (Remarks to the Author):

In this manuscript, *Bacillus subtilis* was developed as the chassis organism for engineered endosymbionts (EES) that escape phagosome destruction, reside in the cytoplasm of mammalian cells, and secrete proteins that are transported to the nucleus to control host cell gene expression. Two sets of transcription factors Stat-1-Klf6 and Klf4-Gata-3 were recombined into the genome of the EES and expressed from regulated promoters. Controlled expression of the mammalian proteins from the EES in the cytoplasm of J774A.1 cells directed gene expression and modulated the host cell fates toward anti- or pro- inflammatory phenotypes by each of the distinct transcription factor pairs. Expressing mammalian transcription factors from engineered intracellular *B. subtilis* as engineered endosymbionts provides a new tool for directing cell fate for therapeutic and research purposes. For example, the EES shows the possibility to be applied in treating damaging inflammatory conditions such as arthritis in the future. Overall, this work has good novelty. However, the writing is not easy to understand and needs further modification, such as strengthening the description of the details and the further interpretation of the data.

Major points:

1) How to control the secretion of target proteins by EES? By fusion of TF genes with the Sec and Tat secretion signals or original substrates? The construction strategy of the exogenous protein-delivery strain should be elaborated in the result section.

2) Since the title is “Engineered endosymbionts that direct mammalian cell gene expression”, whether the transcriptional profile (RNA-seq) of the EES-TFs treated J774A.1 cells should be determined to compare with the controls (resting and polarized macrophages) to demonstrate the EES-TFs treatments indeed direct cell fate switch.

Minor points:

1) Line 132, the “Movie S2” in the parentheses should be Movie S1.

2) Line 141, there is no Fig. 3B at all. Do you mean the right panel of Fig. 3? The same issue happens in Fig. 3A at Line 150.

3) The units of Table S1 should be written clearly. How is the “%infected” calculated?

4) Line 159, the full name of β -gal should be given. The “mannose-inducible system” should be introduced briefly at the beginning of the section of “EES secretes β -gal with delivery to the nuclei of J774A.1 cells”.

5) In Fig. 4, the difference between ‘mannose –NLS’ and ‘mannose NLS’ is not obvious in fluorescence microscopy figure. The effect of nuclear localization is not obvious without nucleus staining. With regards to the nuclear SNR, how the authors distinguished the nuclear signal and the cytoplasmic signal?

Reviewers' comments:

The authors would like to thank the reviewers for insightful comments and suggestions for improving the manuscript. We have include direct responses (in blue text) in this document and made notes on where we included the updated text and data analysis in the manuscript files (red text).

Reviewer #1 (Remarks to the Author):

Madsen and Makela et al report *Bacillus subtilis* strains engineered to secrete transcription factors with the goal of using these strains to control gene expression in mammalian cells. The creation of these transcription factor(TF)-secreting variants and the demonstration that they can deliver transcription factors into the nuclei of macrophages is quite a feat and has exciting potential. The strain development work is interesting, and the experiments are well performed. However, their data for functional readouts of how the TF-secreting strains impact M1/M2 polarization (Fig 5 – mac surface markers and 6 -mac cytokine production) suggest that these strains are not functioning as anticipated. Rather, their data suggest that the EES/LLO chassis (i.e., non-TF secreting) causes a marked innate immune response in macrophages with cytokine responses that dwarf that of polarized M1 and M2 macrophages. This finding means it is difficult to interpret whether the observed changes in macrophage-derived cytokine levels and CD86/CD206 expression between EES and TF-secreting EES actually represent controlled regulation of gene expression or simply changes in the innate response to the EES. As such major revisions are necessary, but overall, with these changes, I believe this paper could make an important contribution to the field.

Thank you for the interest in our work and to help with understanding and significance, we have clarified where there is an innate response to the LLO strain and where we observe specific responses associated with TF expression in the engineered strains. We will further address these concerns below through the more detailed comments.

Major:

Central to the author's manuscript is the idea that gene expression can be modulated by EES and as proof of principle the authors attempt to use their TF-secreting EES to polarize macrophages toward M1 or M2. However, the authors find that EES treatment alone causes macrophages to produce various cytokines (e.g., IL-10, IL-12p40, TNF, G-CSF, IL-1b) at levels, well above their positive controls for macrophage polarization. In some cases (e.g., TNF and G-CSF) cytokine levels are multiple orders of magnitude above the control. The author's discussion of this important finding is relatively cursory, and they should elaborate on what they think is occurring. It seems likely that this is an innate response to EES/LLO infection. Does this occur because the strain is escaping the phagosome? Have the authors compared the cytokine response/immunophenotype (CD86, CD206) of macrophages infected with EES +/- LLO?

The LLO strains caused high expression of the cytokines/chemokines compared to the controls and was not observed in macrophage marker expression was an important finding. A few explanations as to why the LLO strain caused cytokine/chemokine production at levels above the controls, include 1. the difference between Toll-like receptor (TLR)2 and TLR4 in determining cytokine response, 2. the impact of live bacteria compared to just single molecule agonists and 3. the patterns of impact on cytokine/chemokines from the positive controls. This is now well

explained in the text. Live bacteria have been shown to cause a higher stimulation of macrophages compared to just single molecules (reference in manuscript response below) and this combined with both the amount of bacteria and TLR2 also being stimulated with the bacteria which can cause certain cytokines and chemokine responses to be much higher than TLR4 inducers (references in manuscript response below) all contribute to the results displayed in this study. We have more clearly and completely addressed these points in the revised manuscript. For example here is the text from the Discussion section, paragraph 5, and parts from the results section: Overall, dynamic responses were observed for many of the cytokines and chemokines in response to the LLO strain with the LLO-SK and LLO-KG strains able to modify expression further in comparison to the LLO strain in some key examples indicating TF specificity. One component of the differential responses were elevated expression of cytokines and chemokines in response to all bacterial strains compared to the positive controls of IFN- γ and LPS (M1+) or IL-4 and IL-13 (M2+). Additionally, the differences in responses to treatment with bacteria compared to the positive controls was larger in the cytokine and chemokine profiles than were observed with the cell surface markers. Enhanced signaling proteins (cytokines and chemokines) response from macrophages, and other professional antigen presenting cells, compared to cell surface marker expression has been previously observed⁶³⁻⁶⁵. Other factors that could contribute to the observed complex response involves *B. subtilis* stimulating Toll-like receptor 2 (TLR2) in contrast to IFN- γ and LPS stimulating TLR4⁶⁶. Additionally, viable bacteria can cause a more dynamic response than just bacterial products alone⁶⁷. Altogether, even with the dynamic responses from the J774A.1 cells in response to complex signals, the LLO-SK and LLO-KG strains were able to change trends in macrophage surface markers, and cytokine/chemokine profiles in comparison to the LLO strain. This observation is further supported by not observing significant differences in J774A.1 viability, percentage of cells containing fluorescently labeled bacteria and distribution of fluorescence intensity (relating to number of bacteria per cell) when comparing the LLO, LLO-SK and LLO-KG strains (Fig. S6). Future studies should focus on characterizing the impact of these strains in primary macrophages such as bone marrow derived macrophages to clearly understand application potential. In addition, more TF pairings informed by thorough studies that elucidate the complexity of macrophage response should be constructed and tested to optimize the response from the macrophages for the desired application.

Previously, *Listeria monocytogenes* was shown to have specific patterns of inducing cytokine/chemokine production based on expression of LLO (references in manuscript response below). In P388D1 cells, a lymphoma monocyte/macrophage line similar to J774A.1, expression of LLO in the WT *L. monocytogenes* strain demonstrated induction of TNF- α , IL-6 and IL- α while the $\Delta hlyA$ strain did not induce these cytokines. Yet, the $\Delta hlyA$ strain did induce expression of IL- β and to the same level as the WT strain. Conversely, in bone marrow derived macrophages, the expression of these cytokines was not dependent upon LLO expression. Ultimately, the phagosome escape does play a role as demonstrated by our study and previous studies which indicates that the LLO strain is the best control for how much the J774A.1 cytokine/chemokine production is being impacted independently of the TFs. Yet, the LLO-SK and LLO-KG strains impacted the expression of the cytokines/chemokines in significant ways as compared to the LLO strain with similar and opposite trends indicating specificity to the TFs for expression differences.

Additionally, CD86/206 expression resulting from the LLO strain impact is less clear in the literature. Accordingly, we performed the flow cytometry again to include the -LLO strain. We observed no significant differences in response to the *B. subtilis* 168 strain (-LLO) compared to

the LLO strain in both surface markers at both time points. The new flow data included an updated version of the staining protocol which improved CD206 staining. We have recently performed additional CD206 optimization and found that CD206 staining was improved when performed without permeabilization (contrary to manufacturers suggestion); this was implemented in the most recent flow cytometry staining and subsequent data.

Results: Fig. 5

Results: Modulation of J77A.1 cell cytokine and chemokine expression using engineered *B. subtilis* LLO strains

B. subtilis strain 168 yielded similar marker expression profiles in comparison to the LLO strain and a $\Delta hlyA$ mutant (i.e. no LLO expression) of *Listeria monocytogenes* has been shown to either not change cytokine and chemokine profiles in a macrophage cell line, or to change profiles of these proteins equally to the wild-type *L. monocytogenes* in bone marrow derived macrophages^{43,44}. Therefore, the 168 strain was not used in further comparisons, because the lack of LLO was not expected to impact J77A.1 profiles.

Discussion: 3rd paragraph

Cell surface expression levels of CD86 and CD206 were altered by the bacterial conditions, and the LLO-SK and LLO-KG were observed to regulate CD206 (Fig. 5) expression as expected based on the known activity of the expressed TFs. At 24 h, CD86 response was more complex as it is expected to be upregulated as a general response to bacteria. However, by 48 h the LLO-SK strain had increased expression significantly compared to the LLO strain (Fig. 5, S8). The *B. subtilis* 168 strain also increased expression between 24 and 48 h but not significantly relative to the LLO strain.

Methods:

After addition of the engineered *B. subtilis* LLO strains and controls, followed by incubation for 24 or 48 h, cells were collected and stained in a 96-well round bottom plate. Treatment addition was staggered so that flow staining and data acquisition could be performed at the same time for 24 and 48 h cells. All staining steps were performed in 100 μ L volume at 4°C in the dark. Samples were first incubated with Zombie NIR viability dye (1:750, Biolegend) for 20 min. Cells were washed once with flow buffer, followed by incubation with TruStain FcX™ PLUS (anti-mouse CD16/32) Antibody (Biolegend, Cat#156603; 0.25 μ g/sample) for 10 min. Alexa Fluor® 647 anti-mouse CD86 Antibody (0.125 μ g/sample; Biolegend; Cat#105020) and FITC anti-mouse CD206 (MMR) antibody (0.1 μ g/sample, Biolegend; Cat#141703) were then added and incubated for 20 min. Cells were washed twice with flow staining buffer and fixed with 4% PFA for 10 min and resuspended in a final volume of 100 μ l for flow cytometry analysis using the Cytex Aurora spectral flow cytometer (Cytex).

The TF-secreting strains reduce or further increase macrophage cytokine production in some cases, however, in general, the changes do not appear to align with the predicted outcome for the EES-SK and EES-KG i.e. polarization toward M1 versus M2. Changes in CD86 and CD206 levels also do not appear to align the cytokine production levels is also suggestive of an alternate explanation. Although I found the author's discussion of their hypothesis relating to EES-SK and EES-KG effects on macrophage polarization to be somewhat unclear and this should be remedied. This is important given that "specificity" is key to how the authors state they

would like EES should operate. Specifically, Line 180 – EES-SK or KG - the authors should provide the rationale for why they chose these TF pairs to engineer into EES in the results i.e. SK because presumably, they hypothesized this would polarize the macs to M1 and KG combination would polarize to M2 and whether or no the data was consistent with that hypothesis. They should also write out the SK and KG transcription factor pairs in full the first time they are used before describing the results of the TF immunofluorescence in Fig S4.

We have expanded and clarified the introduction and results sections to provide clearer rationale and hypothesis of expected results. We included more of the rationale for the chosen transcription factors in the introduction and referenced back to this in results and discussion. We have also written out the full names in the results section. We have also included the discussion on the difference in marker expression versus cytokine/chemokine response in with the previous comment in the discussion section. Here is that updated text:

Introduction:

Therefore, an EES was designed to express and deliver TFs in two operons for modulating macrophage phenotype towards pro- or anti-inflammatory states. One operon encodes the TFs signal transducer and activator of transcription 1 (STAT-1) and Krüppel-like factor 6 (KLF6) which induce a general response to an inflammatory state in macrophages, and the second encodes Krüppel-like factor 4 (KLF4) and GATA binding protein 3 (GATA-3) which are both characterized to drive an anti-inflammatory response in macrophages³³⁻³⁷. Both STAT-1 and KLF4 are upstream regulators that impact several pathways, while KLF6 and GATA-3 are more specific in regulation, which lends to a dual approach towards driving the desired cell fates³³⁻³⁷. When expressed from intracellular EES these TFs altered patterns of cell surface markers, cytokine and chemokine expression with some patterns of modulation towards anti- or pro-inflammatory phenotypes, indicating that the EES may be used to direct immune cell function and elucidate mechanisms of macrophage response to intracellular bacteria.

Results: Modulation of J77A.1 cell cytokine and chemokine expression using engineered *B. subtilis* LLO strains

β -gal delivery by the LLO-*lacZ* strains demonstrated the possibility of an intracellular EES to functionally deliver proteins capable of transcriptional regulation to the nuclei of host cells. Accordingly, TFs were chosen to be delivered by the EES to alter host cell function. The LLO strain was engineered to secrete two distinct pairs of TFs known to impact macrophage function: STAT-1 and KLF6 (LLO-SK; pro-inflammatory) or KLF4 and GATA-3 (LLO-KG; anti-inflammatory) (Fig. S5). Introduction of the TF pairs into the genome of *B. subtilis* LLO minimally impacted bacterial growth rates, did not adversely affect J77A.1 cell viability compared to the LLO strain and did not alter the ability to escape destruction of phagosomes and persist in macrophages (percent cells containing bacteria and distribution of the fluorescence intensity relating to number of bacteria per cell; Fig. S6). After staining for protein expression, quantification of fluorescence confirmed production and delivery of STAT-1/KLF6 and KLF4/GATA-3 in cells containing LLO-SK or LLO-KG, respectively, after 3 h of TF delivery (Fig. S7). Quantification of nuclear SNR identified increases in nuclear fluorescence in all four TFs after delivery from the LLO-SK or LLO-KG strains with and without addition of D-mannose. D-mannose significantly increased delivery of all four TFs. Additionally, the positive controls for

STAT-1/KLF6 (LPS and IFN- γ) and KLF4/GATA-3 (IL-4 and IL-13) produced some increase in nuclear SNR but significantly less than the LLO-SK and LLO-KG strains over the 3 h.

Discussion: 2nd paragraph

During immune responses, STAT-1 is part of the pro-inflammatory response and is a potent modulator that is directly upregulated by exposure to pro-inflammatory cytokines such as IFN- γ . This response was shown at 3 h in this study (Fig. S7), while KLF6 has been shown to be upregulated in M1 polarized macrophages^{33,34,37}. Therefore, these two factors were used in LLO-SK to compare to IFN- γ and LPS in driving the pro-inflammatory phenotype. The activity of KLF4 is directly stimulated by the signal cascade in cells treated with the cytokine IL-4 and while this stimulation did not produce significant immunofluorescence at 3 h (Fig. S7), this TF has been reported to promote M2 polarization. GATA-3 is also highly upregulated in M2 macrophages^{35,36}, which is the reason these two TF were used in LLO-KG to compare to IL-4 and IL-13 in driving the anti-inflammatory phenotype. Delivery of LLO-SK and LLO-KG strains to J774A.1 cells produced higher levels of these TFs in the nuclei than levels produced by the known signal cascade inducers (IFN- γ and LPS; IL-4 and IL-13). Accordingly, this result demonstrated the potential of *B. subtilis* LLO to impact host cell function within the 3 h timeframe of interaction between the LLO strains and host cells.

Discussion paragraphs 3 and 4 also expand on the discussion of the expected versus unexpected patterns more

Discussion: 5th paragraph

Overall, dynamic responses were observed for many of the cytokines and chemokines in response to the LLO strain with the LLO-SK and LLO-KG strains able to modify expression further in comparison to the LLO strain in some key examples indicating TF specificity. One component of the differential responses were elevated expression of cytokines and chemokines in response to all bacterial strains compared to the positive controls of IFN- γ and LPS (M1+) or IL-4 and IL-13 (M2+). Additionally, the differences in responses to treatment with bacteria compared to the positive controls was larger in the cytokine and chemokine profiles than were observed with the cell surface markers. Enhanced signaling proteins (cytokines and chemokines) response from macrophages, and other professional antigen presenting cells, compared to cell surface marker expression has been previously observed⁶³⁻⁶⁵. Other factors that could contribute to the observed complex response involves *B. subtilis* stimulating Toll-like receptor 2 (TLR2) in contrast to IFN- γ and LPS stimulating TLR4⁶⁶. Additionally, viable bacteria can cause a more dynamic response than just bacterial products alone⁶⁷. Altogether, even with the dynamic responses from the J774A.1 cells in response to complex signals, the LLO-SK and LLO-KG strains were able to change trends in macrophage surface markers, and cytokine/chemokine profiles in comparison to the LLO strain. This observation is further supported by not observing significant differences in J774A.1 viability, percentage of cells containing fluorescently labeled bacteria and distribution of fluorescence intensity (relating to number of bacteria per cell) when comparing the LLO, LLO-SK and LLO-KG strains (Fig. S6). Future studies should focus on characterizing the impact of these strains in primary macrophages such as bone marrow derived macrophages to clearly understand application potential. In addition, more TF pairings informed by thorough studies that elucidate the complexity of macrophage response should be constructed and tested to optimize the response from the macrophages for the desired application.

Whether these macrophage responses represent plasticity or a host response also merits some additional investigation. In the experiments shown in Fig 5 and 6, were EES-SK and EES-KG as infectious as the parental strain? Did macrophage survival change in response to the different EES strains? Did both strains reach the same titers as the parental? Related, the authors should report whether there was any metabolic burden (e.g. perform growth curves) for the introduced circuits and TFF production for EES-SK and EES-KG. Answers to these questions would help to resolve whether the authors are observing controlled changes of macrophage polarization as they predict versus simply changes in the innate immune response to EES/LLO infection.

We have performed growth curves, further viability tests on J774A.1 by flow cytometry and characterization of amount of J774A.1 containing the engineered LLO strains and the fluorescence difference throughout the population of J774A.1 cells indicating number of bacteria in cells. Below we describe the results that we found that further support our hypothesis by controlling for these other variables. This was an important suggestion that helped to support the hypothesis and ideas further.

Results: Fig. S6

Results: Engineered *B. subtilis* LLO transcription factor delivery and modulation of J77A.1 cell marker expression

Introduction of the TF pairs into the genome of *B. subtilis* LLO minimally impacted bacterial growth rates, did not adversely affect J774A.1 cell viability compared to the LLO strain and did not alter the ability to escape destruction of phagosomes and persist in macrophages (percent cells containing bacteria and distribution of the fluorescence intensity relating to number of bacteria per cell; Fig. S6).

Discussion: 5th paragraph

Altogether, even with the dynamic responses from the J774A.1 cells in response to complex signals, the LLO-SK and LLO-KG strains were able to change trends in macrophage surface markers, and cytokine/chemokine profiles in comparison to the LLO strain. This observation is further supported by not observing significant differences in J774A.1 viability, percentage of cells containing fluorescently labeled bacteria and distribution of fluorescence intensity (relating to number of bacteria per cell) when comparing the LLO, LLO-SK and LLO-KG strains (Fig. S6). Future studies should focus on characterizing the impact of these strains in primary macrophages such as bone marrow derived macrophages to clearly understand application potential. In addition, more TF pairings informed by thorough studies that elucidate the complexity of macrophage response should be constructed and tested to optimize the response from the macrophages for the desired application.

Language throughout this manuscript relating to the ability of the strain to modulate macrophage gene expression and plasticity in a specific and predictive way such as Line 252 "In this study macrophage plasticity was modulated..." should be toned down or supported with further experiments as discussed above.

We have both toned down the comments on the effect on plasticity and added new data that support our points. We used the added studies and literature to provide more clarity on the study and rephrased many of these areas to indicate impact on what we were predicting but the

complexity that was shown. We focused more on the trends that were seen which did follow some patterns predicted and even those that might have been unexpected could be explained or alluded to based on the TFs. Some examples of these changes are shown below:

Results: Engineered *B. subtilis* LLO transcription factor delivery and modulation of J774.1 cell marker expression

To determine the impact of engineered *B. subtilis* LLO strain TF delivery on J774.1 modulation, surface marker expression was examined in cells containing various strains (168 strain, LLO strain, LLO-SK and LLO-KG strains in the presence and absence of mannose) and compared to M0 and M1+/M2+ polarized J774.1 cells at 24 or 48 h post-incubation. As determined by flow cytometry, engineered *B. subtilis* LLO strains expressing TFs exhibited different patterns of J774.1 surface marker expression when compared to the LLO strain (Fig. 5, S8).

Results: Modulation of J774.1 cell cytokine and chemokine expression using engineered *B. subtilis* LLO strains

LLO-SK and LLO-KG were shown to alter J774.1 cytokine and chemokine expression patterns relative to the LLO strain as well as positive controls (Fig. 6, S10). Addition of LLO-SK and LLO-KG strains to J774.1 cells led to differential expression of cytokines compared to the cytokine profile observed when the LLO strain was used, as shown by levels of interleukin (IL)-10, IL-12p40 and tumor necrosis factor alpha (TNF- α) (Fig. 6A-D, S10). LLO-SK downregulated granulocyte colony stimulating factor (G-CSF) relative to both the LLO strain and LLO-KG (Fig. 6D-E). Although cytokine production was generally higher at 24 h compared to 48 h post bacterial exposure for most cytokines and chemokines, significant differences were observed at both time points from LLO-SK and LLO-KG in comparison to the LLO strain, and the M1 and M2 positive controls.

Discussion: 3rd paragraph

Even with these complexities, clear changes were observed that indicate that LLO-SK and LLO-KG impacted surface marker expression in comparison to the LLO strain.

Discussion paragraph 4 also talks in detail about the patterns of what was expected and seen and what was unexpected and seen

Discussion: 5th paragraph

Altogether, even with the dynamic responses from the J774.1 cells in response to complex signals, the LLO-SK and LLO-KG strains were able to change trends in macrophage surface markers, and cytokine/chemokine profiles in comparison to the LLO strain. This observation is further supported by not observing significant differences in J774.1 viability, percentage of cells containing fluorescently labeled bacteria and distribution of fluorescence intensity (relating to number of bacteria per cell) when comparing the LLO, LLO-SK and LLO-KG strains (Fig. S6).

Throughout the manuscript the phrase “modulation of gene expression changes” or variants thereof is used to refer to the effect of EES delivered transcription factors. However, at no point is gene expression measured. Rather readouts are performed at the protein level. The authors should either measure transcript levels of their target genes or re-phrase accordingly.

While TFs do directly impact the expression of genes at the level of transcription and this was our intent, we agree that we should be more specific and more appropriately discuss the impact on marker expression and cytokine/chemokine production instead of referring to transcriptional regulation. Our measurements were at the level of proteins, and we agree, we should refer to protein levels and not use language that presumes transcriptional regulation. We believe RNA-seq would be ideal for understanding the complex array of transcriptomic changes caused by the EES but we believe these resources are better allocated towards primary macrophages and as a result is outside the scope of the present study. We have altered the title along with results and conclusions to account for this. Some examples are below:

Title:

Engineered endosymbionts that alter mammalian cell surface marker, cytokine and chemokine expression

Introduction:

When expressed from intracellular EES these TFs altered patterns of cell surface markers, cytokine and chemokine expression with some patterns of modulation towards anti- or pro-inflammatory phenotypes, indicating that the EES may be used to direct immune cell function and elucidate mechanisms of macrophage response to intracellular bacteria.

Results: Modulation of J774A.1 cell cytokine and chemokine expression using engineered *B. subtilis* LLO strains

LLO-SK and LLO-KG were shown to alter J774A.1 cytokine and chemokine expression patterns relative to the LLO strain as well as positive controls (Fig. 6, S10). Addition of LLO-SK and LLO-KG strains to J774A.1 cells led to differential expression of cytokines compared to the cytokine profile observed when the LLO strain was used, as shown by levels of interleukin (IL)-10, IL-12p40 and tumor necrosis factor alpha (TNF- α) (Fig. 6A-D, S10).

Discussion: 5th paragraph

Altogether, even with the dynamic responses from the J774A.1 cells in response to complex signals, the LLO-SK and LLO-KG strains were able to change trends in macrophage surface markers, and cytokine/chemokine profiles in comparison to the LLO strain. This observation is further supported by not observing significant differences in J774A.1 viability, percentage of cells containing fluorescently labeled bacteria and distribution of fluorescence intensity (relating to number of bacteria per cell) when comparing the LLO, LLO-SK and LLO-KG strains (Fig. S6). Future studies should focus on characterizing the impact of these strains in primary macrophages such as bone marrow derived macrophages to clearly understand application potential. In addition, more TF pairings informed by thorough studies that elucidate the complexity of macrophage response should be constructed and tested to optimize the response from the macrophages for the desired application.

Minor:

In general, the Results are quite well written, however, the authors should add some description of each experimental setup. While it is acknowledged that the experimental details are present in the methods, an abbreviated description at the beginning of each Results subsection would aid the reader in understanding what was done without having to flick back and forth between

the methods and results. Moreover, describing the rationale within the Results section would help the reader follow the author's line of reasoning for the experiments being performed.

We agree and we have incorporated brief overviews of both rationale and experimental details into the results section for further clarity. Some examples:

Results: Engineered *B. subtilis* LLO secretes β -gal with delivery to the nuclei of J774A.1 cells

B. subtilis LLO was engineered to produce and secrete β -galactosidase (β -gal) (strain designated LLO-*lacZ*) through the twin-arginine translocation (Tat) pathway by synthesizing the PhoD signal peptide⁴⁰ and the amino acids from 126-132 of the simian virus (SV) 40 nuclear localization signal (NLS)⁴¹. The production of β -gal by LLO-*lacZ*, with and without a nuclear localization signal (NLS; LLO-*lacZ*-NLS and LLO-*lacZ*-no NLS), was studied with IPTG or mannose control.

Results: Engineered *B. subtilis* LLO transcription factor delivery and modulation of J77A.1 cell marker expression

β -gal delivery by the LLO-*lacZ* strains demonstrated the possibility of an intracellular EES to functionally deliver proteins capable of transcriptional regulation to the nuclei of host cells. Accordingly, TFs were chosen to be delivered by the EES to alter host cell function. The LLO strain was engineered to secrete two distinct pairs of TFs known to impact macrophage function: STAT-1 and KLF6 (LLO-SK; pro-inflammatory) or KLF4 and GATA-3 (LLO-KG; anti-inflammatory) (Fig. S5).

Line 124: EES term. In the intro, the authors use "EES" broadly to refer to engineered endosymbionts but elsewhere in the manuscript, starting in the Results, it is also used to refer to the engineered *B. subtilis* LLO strain. The authors should distinguish between the two for clarity. I would suggest reserving EES as a general term a new name used for the strain/platform developed in this manuscript.

We agree that this would help with clarity. We have kept the EES term for the broader sense and we have changed the specific strain names to *B. subtilis* LLO (LLO strain), *B. subtilis* LLO *Stat-1Klf6* (LLO-SK) and *B. subtilis* LLO *Klf4Gata-3* (LLO-KG). We also refer to these strains as the engineered *B. subtilis* LLO strains or TF expressing strains when appropriate. This is demonstrated by many of the examples above.

Lie 154 Can the authors discuss whether the anti420 subtilisin antibody stains viable *B. subtilis* exclusively? If it does not, I think a gentamicin protection assay (or similar) is warranted to demonstrate that the intracellular bacteria are indeed viable as the authors claim.

The anti-subtilisin antibody is just specific to the protein so it does not select for viable cells necessarily even though *B. subtilis* would have needed to be viable to make the protein. However, we believe the live cell imaging data in Fig 3 validates the viability within the cytoplasm. We were able to observe *B. subtilis* LLO replicating in real time within the same macrophage over time with many examples of this occurring. We believe this provides a clear demonstration of intracellular viability that gentamicin protection assays suggest and can only be certain of if the antibiotic is applied for long enough and high enough concentration to not allow for extracellular bacteria to grow. Additionally, the protein delivery of β -gal demonstrates

this viability further and the supernatant control indicates that only viable intracellular bacteria contribute to protein localization to the nucleus which is demonstrated again with the TFs (Fig. 4, S6).

Line 167 “high concentration of gentamicin” – did this effectively kill the intracellular bacteria? A schematic of the synthetic operons introduced into EES would help the reader better understand these circuits and their regulation. If the figure limit is an issue, perhaps this could accompany figure 1.

Yes, the high concentration of gentamicin (25 µg/mL) effectively killed the intracellular bacteria and allowed the J774A.1 cells to recover without rods being present at 24 hours (Fig. 4, S3), and the text has been edited to clearly indicate this.

Results: Engineered *B. subtilis* LLO secretes β-gal with delivery to the nuclei of J774A.1 cells

. Further, the addition of a high concentration of gentamicin (25 µg/mL) at 3 h after induction of β-gal eliminated intracellular bacteria and rescued the J774A.1 cells from overgrowth of the LLO-*lacZ*, allowing for another 21 h of trafficking of protein to the nucleus (Fig. 4, lower image panels; Fig. S3).

The schematic has been included in the supplemental figures as Fig. S5.

Typographical:

Line 405: (Lonza, xx).

We have addressed changed this to include country of origin for mycoplasma kit used.

Methods: Delivery protocol for *B. subtilis* LLO and engineered strains

Cells were tested negative for mycoplasma using the MycoAlert PLUS Mycoplasma Detection Kit (Lonza, USA).

Reviewer #2 (Remarks to the Author):

In this study, the authors attempt to control macrophage polarization using engineered bacteria. First, they engineer *B. subtilis* such that, upon being endocytosed by macrophages, can escape the phagosome and survive in the cytoplasm of the macrophages. Then, they evaluate whether these *B. subtilis* can express proteins that can be imported into the nucleus of the macrophage. Finally, they attempt to influence macrophage polarization by engineering *B. subtilis* to express cell fate-determining transcription factors.

While the proposed idea is intriguing, I find that the majority of the data does not support the conclusions the authors want to make. In general, it seems that most of the effects shown are driven by the mere presence of *B. subtilis* instead of TF expression or addition of the inducer mannose. Moreover, when gene- and mannose-dependent effects are observed, it is sometimes in the opposite direction that would be expected (e.g. Fig 6B and C). While the authors acknowledge some of these irregularities in the discussion section, the rest of the paper is mostly written as if these irregularities did not exist. For example, in line 205 the authors claim

that "EES-SK and EES-KG were shown to modulate mammalian gene expression as determined by cytokine profiling (Fig 6)", but don't acknowledge in the same paragraph that most of these cytokines are released in the presence of EES that does not express any TFs. A few other examples:

Thank you for the interest in our work and we clarified our text relative to the observed results and provided more thorough explanations throughout the text with more specifics about the differences that we observed. We have provided more analyses and experiments to control for extraneous variables and accounted for the concerns in flow cytometry by improving the staining method for CD206 (see specifics below and in the revised manuscript). For cytokines and chemokines, all conditions were compared to the LLO strain. The TFs bearing strains did alter expression in comparison to that strain and it is expected that the LLO strain alone would increase many of the cytokines and chemokines which was reported in previous literature when studying *Listeria monocytogenes*. While the macrophage response is complex and is impacted by several factors, we fortunately did see trends from the TF strains. Here are some of the revised sections.

Results: Engineered *B. subtilis* LLO transcription factor delivery and modulation of J774A.1 cell marker expression

Fig. 5, S6

Introduction of the TF pairs into the genome of *B. subtilis* LLO minimally impacted bacterial growth rates, did not adversely affect J774A.1 cell viability compared to the LLO strain and did not alter the ability to escape destruction of phagosomes and persist in macrophages (percent cells containing bacteria and distribution of the fluorescence intensity relating to number of bacteria per cell; Fig. S6). After staining for protein expression, quantification of fluorescence confirmed production and delivery of STAT-1/KLF6 and KLF4/GATA-3 in cells containing LLO-SK or LLO-KG, respectively, after 3 h of TF delivery (Fig. S7). Quantification of nuclear SNR identified increases in nuclear fluorescence in all four TFs after delivery from the LLO-SK or LLO-KG strains with and without addition of D-mannose. D-mannose significantly increased delivery of all four TFs. Additionally, the positive controls for STAT-1/KLF6 (LPS and IFN- γ) and KLF4/GATA-3 (IL-4 and IL-13) produced some increase in nuclear SNR but significantly less than the LLO-SK and LLO-KG strains over the 3 h.

To determine the impact of engineered *B. subtilis* LLO strain TF delivery on J774A.1 modulation, surface marker expression was examined in cells containing various strains (168 strain, LLO strain, LLO-SK and LLO-KG strains in the presence and absence of mannose) and compared to M0 and M1+/M2+ polarized J774A.1 cells at 24 or 48 h post-incubation. As determined by flow cytometry, engineered *B. subtilis* LLO strains expressing TFs exhibited different patterns of J774A.1 surface marker expression when compared to the LLO strain (Fig. 5, S8). There was no difference in surface marker expression between *B. subtilis* strain 168 and the LLO strain at any time point. At 24 h post-incubation, a significant decrease in CD206 expression was observed in J774A.1 cells containing LLO-SK, with and without the addition of mannose ($p = 0.0048$; not shown on plot), compared to the LLO strain. The same trend was observed at 48 h (Fig. 5A, C). Conversely, the mean fluorescence intensity (MFI) for CD206 staining was increased by LLO-KG, both with and without mannose ($p = 0.0599$; not shown on graph), at levels comparable to those of the positive control at 24 h (M2+). However, the elevated levels were not sustained as indicated by the 48 h time point. CD86 expression levels

were the same in all bacterial conditions at 24 h in comparison to resting cells with no significant differences in comparison to the M1 control (M1+; Fig. 5B, D). At 48 h, the LLO-SK strain significantly increased CD86 MFI in comparison to the LLO strain with or without mannose added. The *B. subtilis* 168 strain showed a significant difference in CD86 MFI in comparison to untreated control but not in comparison to the LLO strain. Differences in CD206 and CD86 expression at 24 and 48 h post treatment were analyzed to reveal temporal changes of each treatment (Fig. S8). CD206 expression was further increased in M2+ J774A.1 cells, whereas the LLO-KG with and without mannose were not able to sustain CD206 expression and showed a decrease at 48 h. CD86 expression was increased at 48 h in cell with the *B. subtilis* strain 168 and LLO-SK with, and without, mannose.

Results: Modulation of J77A.1 cell cytokine and chemokine expression using engineered *B. subtilis* LLO strains

Functional readouts for macrophage polarization and resulting cell fate change include shifts in cytokine and chemokine expression^{7,11,42}. Therefore, to characterize the effect of LLO-SK and LLO-KG on macrophage function, cytokine and chemokines produced by the J774A.1 cells were profiled when treated with the engineered strains relative to the LLO strain. *B. subtilis* strain 168 yielded similar marker expression profiles in comparison to the LLO strain and a $\Delta hlyA$ mutant (i.e. no LLO expression) of *Listeria monocytogenes* has been shown to either not change cytokine and chemokine profiles in a macrophage cell line, or to change profiles of these proteins equally to the wild-type *L. monocytogenes* in bone marrow derived macrophages^{43,44}. Therefore, the 168 strain was not used in further comparisons, because the lack of LLO was not expected to impact J774A.1 profiles. LLO-SK and LLO-KG were shown to alter J774A.1 cytokine and chemokine expression patterns relative to the LLO strain as well as positive controls (Fig. 6, S10). Addition of LLO-SK and LLO-KG strains to J774A.1 cells led to differential expression of cytokines compared to the cytokine profile observed when the LLO strain was used, as shown by levels of interleukin (IL)-10, IL-12p40 and tumor necrosis factor alpha (TNF- α) (Fig. 6A-D, S10). LLO-SK downregulated granulocyte colony stimulating factor (G-CSF) relative to both the LLO strain and LLO-KG (Fig. 6D-E). Although cytokine production was generally higher at 24 h compared to 48 h post bacterial exposure for most cytokines and chemokines, significant differences were observed at both time points from LLO-SK and LLO-KG in comparison to the LLO strain, and the M1 and M2 positive controls.

Discussion:

In this study macrophage plasticity was modulated using TFs expressed from the EES. Cell surface markers are commonly used to differentiate M1 and M2 polarization^{7,11}. Cell surface expression levels of CD86 and CD206 were altered by the bacterial conditions, and the LLO-SK and LLO-KG were observed to regulate CD206 (Fig. 5) expression as expected based on the known activity of the expressed TFs. At 24 h, CD86 response was more complex as it is expected to be upregulated as a general response to bacteria. However, by 48 h the LLO-SK strain had increased expression significantly compared to the LLO strain (Fig. 5, S8). The *B. subtilis* 168 strain also increased expression between 24 and 48 h but not significantly relative to the LLO strain. The surface protein CD206 has been shown to recognize the surface carbohydrates of pathogens and be triggered by proteases produced by *B. subtilis*^{46,47}. This would explain the increase in CD206 expression caused by the *B. subtilis* 168 strain and LLO strain in comparison to the untreated controls. Additionally, while CD86 is regulated directly by inflammatory responses to the engineered *B. subtilis* LLO, the plasticity of the macrophages

could cause CD86 expression to change over time¹¹. It has been suggested that products of *B. subtilis* such as sublancin or exopolysaccharide (EPS), or the treatment of macrophages with *B. subtilis* spores can result in either M1 or M2 activation⁴⁸. Even with these complexities, clear changes were observed that indicate that LLO-*SK* and LLO-*KG* impacted surface marker expression in comparison to the LLO strain. Although changes in cytokine and chemokine expression vary along the spectrum of macrophage polarization, there are well documented cytokines which are used to broadly identify M1 or M2 macrophages^{7,11,49}. These identifying cytokines and chemokines are important when studying disease as they provide information on the function and characteristics of macrophage populations^{42,50}. The M1 and M2 polarization classifications represent a trend in immune responses but is complex and contains some mixed signals, likely due to bacterial stimulation which can cause these classifications to be incomplete or oversimplified^{49,51}. Complexities in macrophage activation, phenotype and plasticity were encountered in this study. In some instances, the LLO-*SK* and LLO-*KG* strains impacted the host cell as predicted in comparison to the LLO strain alone (Fig. 6, S10). The upregulation of IL-10 by LLO-*KG* and down regulation by LLO-*SK* was an expected result along with G-CSF being downregulated by LLO-*SK*^{35,37,52,53}. The LLO strain increased IL-10 and G-CSF which are expected results because IL-10 and G-CSF have been shown to be produced in response to bacteria⁵³⁻⁵⁵. However, in some cases there were results that were not anticipated, and the pleiotropic effects of each selected TF need to be considered. The change in IL-12p40 levels is an example of an unexpected result based on known signaling cascades. IL-12p40 is known to be produced during inflammatory phenotypes by nuclear factor kappa light chain enhancer of activated B cells (NF- κ B)⁵⁶. Therefore, the increase in production of IL-12p40 by LLO-*KG* is unexpected; however, it is possible GATA-3 could affect IL-12p40 production. The IL-12p40 promoter has a canonical GATA binding site but GATA-3 has not been studied with this promoter⁵⁶. The LLO-*SK* strain showed some downregulation of IL-12p40 at 48 h which could be due to STAT-1 driving alternative cytokine production such as IL-27p28^{56,57}. TNF- α downregulation by LLO-*KG* is another expected result³⁵ but downregulation by LLO-*SK* in comparison to the LLO strain indicates further complexity and potential pleiotropic effects of STAT-1. This could be due to TNF- α being regulated by NF- κ B⁵⁸⁻⁶⁰. Furthermore, there may have been metabolic impact on cytokine production due to the addition of D-mannose, which impairs glucose metabolism and is known to suppress the succinate-driven HIF1 α activation of IL-1 β leading to downregulation of IL-1 β ^{61,62}. Accordingly, results indicated that D-mannose had a significant impact on IL-1 β and other cytokines and chemokines, which is an important consideration for future EES studies that rely on sugar-inducible systems. Metabolic reprogramming is most likely playing a significant role in response to the chemical inducer of the engineered *B. subtilis* LLO TF operons and possibly the LLO strain alone⁶².

Overall, dynamic responses were observed for many of the cytokines and chemokines in response to the LLO strain with the LLO-*SK* and LLO-*KG* strains able to modify expression further in comparison to the LLO strain in some key examples indicating TF specificity. One component of the differential responses were elevated expression of cytokines and chemokines in response to all bacterial strains compared to the positive controls of IFN- γ and LPS (M1+) or IL-4 and IL-13 (M2+). Additionally, the differences in responses to treatment with bacteria compared to the positive controls was larger in the cytokine and chemokine profiles than were observed with the cell surface markers. Enhanced signaling proteins (cytokines and chemokines) response from macrophages, and other professional antigen presenting cells, compared to cell surface marker expression has been previously observed⁶³⁻⁶⁵. Other factors

that could contribute to the observed complex response involves *B. subtilis* stimulating Toll-like receptor 2 (TLR2) in contrast to IFN- γ and LPS stimulating TLR4⁶⁶. Additionally, viable bacteria can cause a more dynamic response than just bacterial products alone⁶⁷. Altogether, even with the dynamic responses from the J774A.1 cells in response to complex signals, the LLO-SK and LLO-KG strains were able to change trends in macrophage surface markers, and cytokine/chemokine profiles in comparison to the LLO strain. This observation is further supported by not observing significant differences in J774A.1 viability, percentage of cells containing fluorescently labeled bacteria and distribution of fluorescence intensity (relating to number of bacteria per cell) when comparing the LLO, LLO-SK and LLO-KG strains (Fig. S6). Future studies should focus on characterizing the impact of these strains in primary macrophages such as bone marrow derived macrophages to clearly understand application potential. In addition, more TF pairings informed by thorough studies that elucidate the complexity of macrophage response should be constructed and tested to optimize the response from the macrophages for the desired application.

- Fig 2: It is hard to see colocalization of red and purple spots (or the lack thereof). This is less of a problem in the SI movies but it is still not great. At the very least, I recommend using less similar colors.

We have improved Fig. 2 to better present the confocal image data. We were informed by experts in the imaging core that pseudo coloring Cy5 to magenta is becoming a new standard when other orange to red wavelengths are being used. We decided to enhance Fig. 2 by withdrawing the membrane stain in some panels to give better viewing of the LAMP-1 and *B. subtilis* staining. We also slightly adjusted the z-depth projection of *B. subtilis* to reveal better visualization of rods in relation to phagosomal pockets.

Results: *B. subtilis* LLO escape from phagosomes of J774A.1 cells

Fig. 2

- Table S1: IPTG does not seem to make a significant difference in EES/J774A.1 and % infected, therefore I would not conclude that expression of LLO is what is driving EES numbers or replication.

We have adjusted the table to make this more clear. The % infected cells is more accurately the amount of J774A.1 cells containing EES which is calculated by determining fluorescent EES throughout the z-depth of each cell over the total number of cells imaged. The -IPTG condition is the only condition where bacteria per J774A.1 decreased between 1 h and 2 h post bacterial addition due to destruction in the phagosome. Fig. 2 (4 h after bacterial addition) shows minimal rods in -IPTG condition and more punctate signal along with the SI movies. This result was also shown during the *B. subtilis* LLO developmental studies, and is described in the revised manuscript—here is an excerpt:

Results: *B. subtilis* LLO escape from phagosomes of J774A.1 cells

Confocal microscopy confirmed the escape of *B. subtilis* LLO from phagosomes after uptake into J774A.1 cells¹². To further elucidate the mechanism of escape, *B. subtilis* LLO (magenta) localization was compared to LAMP-1³⁸ positive structures (phagosomes, red) in J774A.1 cells (green) with and without IPTG induction of LLO expression (+IPTG and -IPTG). The LAMP-1 protein is crucial for phagosomal assembly and therefore will reveal when the LLO strain is

contained within the phagosomes and when the phagosomes have been disrupted³⁹. When LLO expression was induced by IPTG, many of the LLO strain were intact and present throughout the mammalian cells (Fig. 2, zoom-dotted line; Movie S1). Z-stack data analysis identified *B. subtilis* LLO throughout the cytoplasm of J774A.1 cells and not associated with LAMP-1 positive structures (Fig. 2, zoom-solid line; Movie S1). In contrast, without IPTG induction, few of the LLO strain were observed and many regions of punctate signal within LAMP-1 positive regions were observed (Fig. 2, zoom-solid line; Movie S2). Accordingly, the expression of LLO when induced by IPTG allows *B. subtilis* LLO to access the cytoplasm of the host cells.

- Figure S4: expression of KLF6, KLF4, and GATA-3 are visible in the negative controls, and thus it is hard to conclude that the EES or addition of manose have any influence in expression of these regulators. Quantification of fluorescence levels may partially alleviate this issue.

We have quantified the fluorescence to address this concern and edited the manuscript accordingly. These TFs are native to the host cells so some expression is expected and even serves as a control for the antibodies. Additionally, only in the LLO-SK and LLO-KG conditions with and without mannose are the bacteria also fluorescent indicating expression of the proteins—this is more clearly stated in the revised manuscript. The mannose system is known to affect transcription even without the presence of mannose which is why the -mannose condition also demonstrates signal which follows Fig. 4. The revised results and discussion section describing these observations is excerpted below.

Results: Engineered *B. subtilis* LLO transcription factor delivery and modulation of J774A.1 cell marker expression

After staining for protein expression, quantification of fluorescence confirmed production and delivery of STAT-1/KLF6 and KLF4/GATA-3 in cells containing LLO-SK or LLO-KG, respectively, after 3 h of TF delivery (Fig. S7). Quantification of nuclear SNR identified increases in nuclear fluorescence in all four TFs after delivery from the LLO-SK or LLO-KG strains with and without addition of D-mannose. D-mannose significantly increased delivery of all four TFs. Additionally, the positive controls for STAT-1/KLF6 (LPS and IFN- γ) and KLF4/GATA-3 (IL-4 and IL-13) produced some increase in nuclear SNR but significantly less than the LLO-SK and LLO-KG strains over the 3 h.

Discussion: 2nd paragraph

During immune responses, STAT-1 is part of the pro-inflammatory response and is a potent modulator that is directly upregulated by exposure to pro-inflammatory cytokines such as IFN- γ . This response was shown at 3 h in this study (Fig. S7), while KLF6 has been shown to be upregulated in M1 polarized macrophages^{33,34,37}. Therefore, these two factors were used in LLO-SK to compare to IFN- γ and LPS in driving the pro-inflammatory phenotype. The activity of KLF4 is directly stimulated by the signal cascade in cells treated with the cytokine IL-4 and while this stimulation did not produce significant immunofluorescence at 3 h (Fig. S7), this TF has been reported to promote M2 polarization. GATA-3 is also highly upregulated in M2 macrophages^{35,36}, which is the reason these two TF were used in LLO-KG to compare to IL-4 and IL-13 in driving the anti-inflammatory phenotype. Delivery of LLO-SK and LLO-KG strains to J774A.1 cells produced higher levels of these TFs in the nuclei than levels produced by the known signal cascade inducers (IFN- γ and LPS; IL-4 and IL-13). Accordingly, this result

demonstrated the potential of *B. subtilis* LLO to impact host cell function within the 3 h timeframe of interaction between the LLO strains and host cells.

- Figure S5: In panel A, are the thresholds chosen to represent some type of transition to the M1 or M2 states? Why do the negative controls start with high levels of CD86? Why does the M1+ positive control also have high CD206? It is hard to assess whether the samples under evaluation are doing anything if the controls are not behaving as expected.

We repeated the flow cytometry with an improved staining procedure to resolve the concern as mentioned in the first comment. We tested CD206 with/without permeabilization (manufacturer suggested to permeabilize cells) and we found that CD206 was much improved without permeabilization. The gating/thresholds in the previous data and new data were chosen based on FMO/single stained controls. We observed with this new staining procedure that the positive controls performed as expected and we also saw the TF expressing strains showing trends similar to those expected (Fig. 5). This is now included in the revised manuscript and revised sections include:

Results: Engineered *B. subtilis* LLO transcription factor delivery and modulation of J774.1 cell marker expression

To determine the impact of engineered *B. subtilis* LLO strain TF delivery on J774A.1 modulation, surface marker expression was examined in cells containing various strains (168 strain, LLO strain, LLO-SK and LLO-KG strains in the presence and absence of mannose) and compared to M0 and M1+/M2+ polarized J774.1 cells at 24 or 48 h post-incubation. As determined by flow cytometry, engineered *B. subtilis* LLO strains expressing TFs exhibited different patterns of J774A.1 surface marker expression when compared to the LLO strain (Fig. 5, S8). There was no difference in surface marker expression between *B. subtilis* strain 168 and the LLO strain at any time point. At 24 h post-incubation, a significant decrease in CD206 expression was observed in J774A.1 cells containing LLO-SK, with and without the addition of mannose ($p = 0.0048$; not shown on plot), compared to the LLO strain. The same trend was observed at 48 h (Fig. 5A, C). Conversely, the mean fluorescence intensity (MFI) for CD206 staining was increased by LLO-KG, both with and without mannose ($p = 0.0599$; not shown on graph), at levels comparable to those of the positive control at 24 h (M2+). However, the elevated levels were not sustained as indicated by the 48 h time point. CD86 expression levels were the same in all bacterial conditions at 24 h in comparison to resting cells with no significant differences in comparison to the M1 control (M1+; Fig. 5B, D). At 48 h, the LLO-SK strain significantly increased CD86 MFI in comparison to the LLO strain with or without mannose added. The *B. subtilis* 168 strain showed a significant difference in CD86 MFI in comparison to untreated control but not in comparison to the LLO strain. Differences in CD206 and CD86 expression at 24 and 48 h post treatment were analyzed to reveal temporal changes of each treatment (Fig. S8). CD206 expression was further increased in M2+ J774A.1 cells, whereas the LLO-KG with and without mannose were not able to sustain CD206 expression and showed a decrease at 48 h. CD86 expression was increased at 48 h in cell with the *B. subtilis* strain 168 and LLO-SK with, and without, mannose.

Discussion: 3rd paragraph

In this study macrophage plasticity was modulated using TFs expressed from the EES. Cell surface markers are commonly used to differentiate M1 and M2 polarization^{7,11}. Cell surface

expression levels of CD86 and CD206 were altered by the bacterial conditions, and the LLO-SK and LLO-KG were observed to regulate CD206 (Fig. 5) expression as expected based on the known activity of the expressed TFs. At 24 h, CD86 response was more complex as it is expected to be upregulated as a general response to bacteria. However, by 48 h the LLO-SK strain had increased expression significantly compared to the LLO strain (Fig. 5, S8). The *B. subtilis* 168 strain also increased expression between 24 and 48 h but not significantly relative to the LLO strain. The surface protein CD206 has been shown to recognize the surface carbohydrates of pathogens and be triggered by proteases produced by *B. subtilis*^{46,47}. This would explain the increase in CD206 expression caused by the *B. subtilis* 168 strain and LLO strain in comparison to the untreated controls. Additionally, while CD86 is regulated directly by inflammatory responses to the engineered *B. subtilis* LLO, the plasticity of the macrophages could cause CD86 expression to change over time¹¹. It has been suggested that products of *B. subtilis* such as sublacin or exopolysaccharide (EPS), or the treatment of macrophages with *B. subtilis* spores can result in either M1 or M2 activation⁴⁸. Even with these complexities, clear changes were observed that indicate that LLO-SK and LLO-KG impacted surface marker expression in comparison to the LLO strain.

Methods:

After addition of the engineered *B. subtilis* LLO strains and controls, followed by incubation for 24 or 48 h, cells were collected and stained in a 96-well round bottom plate. Treatment addition was staggered so that flow staining and data acquisition could be performed at the same time for 24 and 48 h cells. All staining steps were performed in 100 μ L volume at 4°C in the dark. Samples were first incubated with Zombie NIR viability dye (1:750, Biolegend) for 20 min. Cells were washed once with flow buffer, followed by incubation with TruStain FcX™ PLUS (anti-mouse CD16/32) Antibody (Biolegend, Cat#156603; 0.25 μ g/sample) for 10 min. Alexa Fluor® 647 anti-mouse CD86 Antibody (0.125 μ g/sample; Biolegend; Cat#105020) and FITC anti-mouse CD206 (MMR) antibody (0.1 μ g/sample, Biolegend; Cat#141703) were then added and incubated for 20 min. Cells were washed twice with flow staining buffer and fixed with 4% PFA for 10 min and resuspended in a final volume of 100 μ l for flow cytometry analysis using the Cytex Aurora spectral flow cytometer (Cytex).

- Figure 6: Why do the M1+ and M2+ positive controls not show any signal?

We have included some panels in Fig. 6 with broken axis to clarify that the controls do causes changes and even magnitudes higher than background but just not to the extent of the LLO strain and TF expressing strains. Fig. S10 also shows more clarity on the impact from the positive controls. We have also noted that the bacteria producing a substantial response in comparison to the positive controls is an explainable result that has been seen previously.

Discussion: 5th paragraph

Overall, dynamic responses were observed for many of the cytokines and chemokines in response to the LLO strain with the LLO-SK and LLO-KG strains able to modify expression further in comparison to the LLO strain in some key examples indicating TF specificity. One component of the differential responses were elevated expression of cytokines and chemokines in response to all bacterial strains compared to the positive controls of IFN- γ and LPS (M1+) or IL-4 and IL-13 (M2+). Additionally, the differences in responses to treatment with bacteria compared to the positive controls was larger in the cytokine and chemokine profiles than were

observed with the cell surface markers. Enhanced signaling proteins (cytokines and chemokines) response from macrophages, and other professional antigen presenting cells, compared to cell surface marker expression has been previously observed^{63–65}. Other factors that could contribute to the observed complex response involves *B. subtilis* stimulating Toll-like receptor 2 (TLR2) in contrast to IFN- γ and LPS stimulating TLR4⁶⁶. Additionally, viable bacteria can cause a more dynamic response than just bacterial products alone⁶⁷. Altogether, even with the dynamic responses from the J774A.1 cells in response to complex signals, the LLO-SK and LLO-KG strains were able to change trends in macrophage surface markers, and cytokine/chemokine profiles in comparison to the LLO strain. This observation is further supported by not observing significant differences in J774A.1 viability, percentage of cells containing fluorescently labeled bacteria and distribution of fluorescence intensity (relating to number of bacteria per cell) when comparing the LLO, LLO-SK and LLO-KG strains (Fig. S6). Future studies should focus on characterizing the impact of these strains in primary macrophages such as bone marrow derived macrophages to clearly understand application potential. In addition, more TF pairings informed by thorough studies that elucidate the complexity of macrophage response should be constructed and tested to optimize the response from the macrophages for the desired application.

In summary, it is hard for me to recommend publication of this paper in its current form. I suggest the authors to substantially revise their manuscript and be more forthcoming about the data irregularities.

We have substantially revised the manuscript with new experimental data, revised text and new information. The new experiments have provided more clarity on our previous observations and conclusions. We have made sure to indicate that we do see some expected results which is promising for the approach and provide a better understanding of macrophage responses to bacteria, and the effect of TFs. We anticipate that the EES can be a mechanism to understanding macrophage responses, but much of this is beyond the scope of the present study.

One last point about line 263: *B. subtilis* 168 absolutely does sporulate, and it is frequently used as a model of sporulation. See, for example, <https://www.ncbi.nlm.nih.gov/pmc/articles/PMC178487/>

We have updated this section to acknowledge that the LLO strain and other engineered *B. subtilis* strains could be producing spores if the strains are under enough stress to do so (<https://www.ncbi.nlm.nih.gov/pmc/articles/PMC2819312/#R1>) in these conditions (this was an important point for clarification). We were still able to see trends in marker expression even with these other factors that could be involved.

Discussion: 3rd paragraph

It has been suggested that products of *B. subtilis* such as subblancin or exopolysaccharide (EPS), or the treatment of macrophages with *B. subtilis* spores can result in either M1 or M2 activation⁴⁸. Even with these complexities, clear changes were observed that indicate that LLO-SK and LLO-KG impacted surface marker expression in comparison to the LLO strain.

Reviewer #3 (Remarks to the Author):

In this manuscript, *Bacillus subtilis* was developed as the chassis organism for engineered endosymbionts (EES) that escape phagosome destruction, reside in the cytoplasm of mammalian cells, and secrete proteins that are transported to the nucleus to control host cell gene expression. Two sets of transcription factors Stat-1-Klf6 and Klf4-Gata-3 were recombined into the genome of the EES and expressed from regulated promoters. Controlled expression of the mammalian proteins from the EES in the cytoplasm of J774A.1 cells directed gene expression and modulated the host cell fates toward anti- or pro- inflammatory phenotypes by each of the distinct transcription factor pairs. Expressing mammalian transcription factors from engineered intracellular *B. subtilis* as engineered endosymbionts provides a new tool for directing cell fate for therapeutic and research purposes. For example, the EES shows the possibility to be applied in treating damaging inflammatory conditions such as arthritis in the future. Overall, this work has good novelty. However, the writing is not easy to understand and needs further modification, such as strengthening the description of the details and the further interpretation of the data.

Thank you for the interest in our work. We have extended our discussion of the data and provided more details. We have also done further analysis and experiments that provide more clarity and resolution. We have also expanded details in results section to hopefully increase the clarity and alleviate ambiguity from introduction through results to discussion. Some examples:

Introduction:

Therefore, an EES was designed to express and deliver TFs in two operons for modulating macrophage phenotype towards pro- or anti-inflammatory states. One operon encodes the TFs signal transducer and activator of transcription 1 (STAT-1) and Krüppel-like factor 6 (KLF6) which induce a general response to an inflammatory state in macrophages, and the second encodes Krüppel-like factor 4 (KLF4) and GATA binding protein 3 (GATA-3) which are both characterized to drive an anti-inflammatory response in macrophages³³⁻³⁷. Both STAT-1 and KLF4 are upstream regulators that impact several pathways, while KLF6 and GATA-3 are more specific in regulation, which lends to a dual approach towards driving the desired cell fates³³⁻³⁷. When expressed from intracellular EES these TFs altered patterns of cell surface markers, cytokine and chemokine expression with some patterns of modulation towards anti- or pro-inflammatory phenotypes, indicating that the EES may be used to direct immune cell function and elucidate mechanisms of macrophage response to intracellular bacteria.

Results: Engineered *B. subtilis* LLO secretes β -gal with delivery to the nuclei of J774A.1 cells

Protein secretion from was used to further demonstrate bacterial viability within the cytoplasm and as an initial demonstration of protein delivery to the nucleus. *B. subtilis* LLO was engineered to produce and secrete β -galactosidase (β -gal) (strain designated LLO-*lacZ*) through the twin-arginine translocation (Tat) pathway by synthesizing the PhoD signal peptide⁴⁰ and the amino acids from 126-132 of the simian virus (SV) 40 nuclear localization signal (NLS)⁴¹. Results: Engineered *B. subtilis* LLO transcription factor delivery and modulation of J77A.1 cell marker expression

β -gal delivery by the LLO-*lacZ* strains demonstrated the possibility of an intracellular EES to functionally deliver proteins capable of transcriptional regulation to the nuclei of host cells.

Accordingly, TFs were chosen to be delivered by the EES to alter host cell function. The LLO strain was engineered to secrete two distinct pairs of TFs known to impact macrophage function: STAT-1 and KLF6 (LLO-SK; pro-inflammatory) or KLF4 and GATA-3 (LLO-KG; anti-inflammatory) (Fig. S5). Introduction of the TF pairs into the genome of *B. subtilis* LLO minimally impacted bacterial growth rates, did not adversely affect J774A.1 cell viability compared to the LLO strain and did not alter the ability to escape destruction of phagosomes and persist in macrophages (percent cells containing bacteria and distribution of the fluorescence intensity relating to number of bacteria per cell; Fig. S6). After staining for protein expression, quantification of fluorescence confirmed production and delivery of STAT-1/KLF6 and KLF4/GATA-3 in cells containing LLO-SK or LLO-KG, respectively, after 3 h of TF delivery (Fig. S7). Quantification of nuclear SNR identified increases in nuclear fluorescence in all four TFs after delivery from the LLO-SK or LLO-KG strains with and without addition of D-mannose. D-mannose significantly increased delivery of all four TFs. Additionally, the positive controls for STAT-1/KLF6 (LPS and IFN- γ) and KLF4/GATA-3 (IL-4 and IL-13) produced some increase in nuclear SNR but significantly less than the LLO-SK and LLO-KG strains over the 3 h.

To determine the impact of engineered *B. subtilis* LLO strain TF delivery on J774A.1 modulation, surface marker expression was examined in cells containing various strains (168 strain, LLO strain, LLO-SK and LLO-KG strains in the presence and absence of mannose) and compared to M0 and M1+/M2+ polarized J774.1 cells at 24 or 48 h post-incubation. As determined by flow cytometry, engineered *B. subtilis* LLO strains expressing TFs exhibited different patterns of J774A.1 surface marker expression when compared to the LLO strain (Fig. 5, S8).

Discussion:

During immune responses, STAT-1 is part of the pro-inflammatory response and is a potent modulator that is directly upregulated by exposure to pro-inflammatory cytokines such as IFN- γ . This response was shown at 3 h in this study (Fig. S7), while KLF6 has been shown to be upregulated in M1 polarized macrophages^{33,34,37}. Therefore, these two factors were used in LLO-SK to compare to IFN- γ and LPS in driving the pro-inflammatory phenotype. The activity of KLF4 is directly stimulated by the signal cascade in cells treated with the cytokine IL-4 and while this stimulation did not produce significant immunofluorescence at 3 h (Fig. S7), this TF has been reported to promote M2 polarization. GATA-3 is also highly upregulated in M2 macrophages^{35,36}, which is the reason these two TF were used in LLO-KG to compare to IL-4 and IL-13 in driving the anti-inflammatory phenotype. Delivery of LLO-SK and LLO-KG strains to J774A.1 cells produced higher levels of these TFs in the nuclei than levels produced by the known signal cascade inducers (IFN- γ and LPS; IL-4 and IL-13). Accordingly, this result demonstrated the potential of *B. subtilis* LLO to impact host cell function within the 3 h timeframe of interaction between the LLO strains and host cells.

In this study macrophage plasticity was modulated using TFs expressed from the EES. Cell surface markers are commonly used to differentiate M1 and M2 polarization^{7,11}. Cell surface expression levels of CD86 and CD206 were altered by the bacterial conditions, and the LLO-SK and LLO-KG were observed to regulate CD206 (Fig. 5) expression as expected based on the known activity of the expressed TFs. At 24 h, CD86 response was more complex as it is expected to be upregulated as a general response to bacteria. However, by 48 h the LLO-SK strain had increased expression significantly compared to the LLO strain (Fig. 5, S8). The *B. subtilis* 168 strain also increased expression between 24 and 48 h but not significantly relative

to the LLO strain. The surface protein CD206 has been shown to recognize the surface carbohydrates of pathogens and be triggered by proteases produced by *B. subtilis*^{46,47}. This would explain the increase in CD206 expression caused by the *B. subtilis* 168 strain and LLO strain in comparison to the untreated controls. Additionally, while CD86 is regulated directly by inflammatory responses to the engineered *B. subtilis* LLO, the plasticity of the macrophages could cause CD86 expression to change over time¹¹. It has been suggested that products of *B. subtilis* such as sublancin or exopolysaccharide (EPS), or the treatment of macrophages with *B. subtilis* spores can result in either M1 or M2 activation⁴⁸. Even with these complexities, clear changes were observed that indicate that LLO-SK and LLO-KG impacted surface marker expression in comparison to the LLO strain.

Although changes in cytokine and chemokine expression vary along the spectrum of macrophage polarization, there are well documented cytokines which are used to broadly identify M1 or M2 macrophages^{7,11,49}. These identifying cytokines and chemokines are important when studying disease as they provide information on the function and characteristics of macrophage populations^{42,50}. The M1 and M2 polarization classifications represent a trend in immune responses but is complex and contains some mixed signals, likely due to bacterial stimulation which can cause these classifications to be incomplete or oversimplified^{49,51}. Complexities in macrophage activation, phenotype and plasticity were encountered in this study. In some instances, the LLO-SK and LLO-KG strains impacted the host cell as predicted in comparison to the LLO strain alone (Fig. 6, S10). The upregulation of IL-10 by LLO-KG and down regulation by LLO-SK was an expected result along with G-CSF being downregulated by LLO-SK^{35,37,52,53}. The LLO strain increased IL-10 and G-CSF which are expected results because IL-10 and G-CSF have been shown to be produced in response to bacteria⁵³⁻⁵⁵. However, in some cases there were results that were not anticipated, and the pleiotropic effects of each selected TF need to be considered. The change in IL-12p40 levels is an example of an unexpected result based on known signaling cascades. IL-12p40 is known to be produced during inflammatory phenotypes by nuclear factor kappa light chain enhancer of activated B cells (NF- κ B)⁵⁶. Therefore, the increase in production of IL-12p40 by LLO-KG is unexpected; however, it is possible GATA-3 could affect IL-12p40 production. The IL-12p40 promoter has a canonical GATA binding site but GATA-3 has not been studied with this promoter⁵⁶. The LLO-SK strain showed some downregulation of IL-12p40 at 48 h which could be due to STAT-1 driving alternative cytokine production such as IL-27p28^{56,57}. TNF- α downregulation by LLO-KG is another expected result³⁵ but downregulation by LLO-SK in comparison to the LLO strain indicates further complexity and potential pleiotropic effects of STAT-1. This could be due to TNF- α being regulated by NF- κ B⁵⁸⁻⁶⁰. Furthermore, there may have been metabolic impact on cytokine production due to the addition of D-mannose, which impairs glucose metabolism and is known to suppress the succinate-driven HIF1 α activation of IL-1 β leading to downregulation of IL-1 β ^{61,62}. Accordingly, results indicated that D-mannose had a significant impact on IL-1 β and other cytokines and chemokines, which is an important consideration for future EES studies that rely on sugar-inducible systems. Metabolic reprogramming is most likely playing a significant role in response to the chemical inducer of the engineered *B. subtilis* LLO TF operons and possibly the LLO strain alone⁶².

Overall, dynamic responses were observed for many of the cytokines and chemokines in response to the LLO strain with the LLO-SK and LLO-KG strains able to modify expression further in comparison to the LLO strain in some key examples indicating TF specificity. One component of the differential responses were elevated expression of cytokines and chemokines

in response to all bacterial strains compared to the positive controls of IFN- γ and LPS (M1+) or IL-4 and IL-13 (M2+). Additionally, the differences in responses to treatment with bacteria compared to the positive controls was larger in the cytokine and chemokine profiles than were observed with the cell surface markers. Enhanced signaling proteins (cytokines and chemokines) response from macrophages, and other professional antigen presenting cells, compared to cell surface marker expression has been previously observed^{63–65}. Other factors that could contribute to the observed complex response involves *B. subtilis* stimulating Toll-like receptor 2 (TLR2) in contrast to IFN- γ and LPS stimulating TLR4⁶⁶. Additionally, viable bacteria can cause a more dynamic response than just bacterial products alone⁶⁷. Altogether, even with the dynamic responses from the J774A.1 cells in response to complex signals, the LLO-SK and LLO-KG strains were able to change trends in macrophage surface markers, and cytokine/chemokine profiles in comparison to the LLO strain. This observation is further supported by not observing significant differences in J774A.1 viability, percentage of cells containing fluorescently labeled bacteria and distribution of fluorescence intensity (relating to number of bacteria per cell) when comparing the LLO, LLO-SK and LLO-KG strains (Fig. S6). Future studies should focus on characterizing the impact of these strains in primary macrophages such as bone marrow derived macrophages to clearly understand application potential. In addition, more TF pairings informed by thorough studies that elucidate the complexity of macrophage response should be constructed and tested to optimize the response from the macrophages for the desired application.

Major points:

1) How to control the secretion of target proteins by EES? By fusion of TF genes with the Sec and Tat secretion signals or original substrates? The construction strategy of the exogenous protein-delivery strain should be elaborated in the result section.

We have included an explanation in the results section as well as Fig. S5 as a schematic of the operon construction. The PhoD secretion peptide is fused to the protein of interest then is cleaved by SPase I when the fully folded protein is translocated across the membrane leaving just the protein of interest secreted. The same construction strategy was used for both β -gal and the TFs except the intrinsic NLS to the TFs was used instead of a viral NLS as for β gal.

Results: Engineered *B. subtilis* LLO secretes β -gal with delivery to the nuclei of J774A.1 cells

Protein secretion from was used to further demonstrate bacterial viability within the cytoplasm and as an initial demonstration of protein delivery to the nucleus. *B. subtilis* LLO was engineered to produce and secrete β -galactosidase (β -gal) (strain designated LLO-*lacZ*) through the twin-arginine translocation (Tat) pathway by synthesizing the PhoD signal peptide⁴⁰ and the amino acids from 126-132 of the simian virus (SV) 40 nuclear localization signal (NLS)⁴¹.

Results: Engineered *B. subtilis* LLO transcription factor delivery and modulation of J77A.1 cell marker expression

β -gal delivery by the LLO-*lacZ* strains demonstrated the possibility of an intracellular EES to functionally deliver proteins capable of transcriptional regulation to the nuclei of host cells. Accordingly, TFs were chosen to be delivered by the EES to alter host cell function. The LLO strain was engineered to secrete two distinct pairs of TFs known to impact macrophage function: STAT-1 and KLF6 (LLO-SK; pro-inflammatory) or KLF4 and GATA-3 (LLO-KG; anti-inflammatory) (Fig. S5).

2) Since the title is “Engineered endosymbionts that direct mammalian cell gene expression”, whether the transcriptional profile (RNA-seq) of the EES-TFs treated J774A.1 cells should be determined to compare with the controls (resting and polarized macrophages) to demonstrate the EES-TFs treatments indeed direct cell fate switch.

We agree that RNA-seq would be ideal for understanding the complex array of transcriptomic changes caused by the EES but we believe these resources are better allocated towards primary macrophages and as a result is outside the scope of the study. We agree with the implication to change the emphasis of the paper to be more specific in regards to impacting marker expression and cytokine/chemokine production. We believe that measuring these protein outputs is consistent with understanding functional changes of the J774A.1 cells after treatment with the LLO strain and TF expressing strains. We have adjusted the title and discussion throughout the manuscript to be more accurate with what was measured. Some examples of changes:

Title:

Engineered endosymbionts that alter mammalian cell surface marker, cytokine and chemokine expression

Introduction:

When expressed from intracellular EES these TFs altered patterns of cell surface markers, cytokine and chemokine expression with some patterns of modulation towards anti- or pro-inflammatory phenotypes, indicating that the EES may be used to direct immune cell function and elucidate mechanisms of macrophage response to intracellular bacteria.

Results: Modulation of J77A.1 cell cytokine and chemokine expression using engineered *B. subtilis* LLO strains

LLO-SK and LLO-KG were shown to alter J774A.1 cytokine and chemokine expression patterns relative to the LLO strain as well as positive controls (Fig. 6, S10). Addition of LLO-SK and LLO-KG strains to J774A.1 cells led to differential expression of cytokines compared to the cytokine profile observed when the LLO strain was used, as shown by levels of interleukin (IL)-10, IL-12p40 and tumor necrosis factor alpha (TNF- α) (Fig. 6A-D, S10).

Discussion: 3rd paragraph

In this study macrophage plasticity was modulated using TFs expressed from the EES. Cell surface markers are commonly used to differentiate M1 and M2 polarization^{7,11}.

Discussion: 4th paragraph

Although changes in cytokine and chemokine expression vary along the spectrum of macrophage polarization, there are well documented cytokines which are used to broadly identify M1 or M2 macrophages^{7,11,49}. These identifying cytokines and chemokines are important when studying disease as they provide information on the function and characteristics of macrophage populations^{42,50}.

Discussion: 5th paragraph

Altogether, even with the dynamic responses from the J774A.1 cells in response to complex signals, the LLO-SK and LLO-KG strains were able to change trends in macrophage surface markers, and cytokine/chemokine profiles in comparison to the LLO strain. This observation is further supported by not observing significant differences in J774A.1 viability, percentage of cells containing fluorescently labeled bacteria and distribution of fluorescence intensity (relating to number of bacteria per cell) when comparing the LLO, LLO-SK and LLO-KG strains (Fig. S6). Future studies should focus on characterizing the impact of these strains in primary macrophages such as bone marrow derived macrophages to clearly understand application potential. In addition, more TF pairings informed by thorough studies that elucidate the complexity of macrophage response should be constructed and tested to optimize the response from the macrophages for the desired application.

Minor points:

1) Line 132, the “Movie S2” in the parentheses should be Movie S1.

This is an important find and we have corrected this error.

2) Line 141, there is no Fig. 3B at all. Do you mean the right panel of Fig. 3? The same issue happens in Fig. 3A at Line 150.

Yes this is an astute observation. We have adjusted these lines to accurately reflect the figure as left and right panel.

Results: Viability of J774A.1 cells and *B. subtilis* LLO replication in the host cell cytoplasm

With IPTG induction, the LLO strain was observed to replicate in the host cell cytoplasm after phagosomal escape indicating active metabolism and viability (Fig. 3, right panel). An increase in the number of bacteria was visualized over time, after co-incubation with a 10:1 MOI. Zoomed regions demonstrate that each bacterium doubled twice during the two time points from 3 bacteria at 1 h to 12 at 2.5 h in this representative instance (Fig. 3, two right panels).

3) The units of Table S1 should be written clearly. How is the “%infected” calculated?

We have adjusted the table to make this more clear. The % infected is more accurately the amount of J774A.1 cells containing EES which is calculated by determining fluorescent EES throughout the z-depth of each cell over the total number of cells imaged.

Methods:

Brightfield and fluorescent images were acquired consecutively, using a 63x oil objective every 1 h beginning at 1 h post co-incubation and continuing until 4 h post incubation Z-stacks were taken at all time points at 0.4 μm steps to confirm *B. subtilis* LLO presence within cytoplasm. *B. subtilis* LLO presence in J774A.1 cells was quantified using Fiji (ImageJ) software and cell counter plugin by counting $>1.5 \mu\text{m}$ rods throughout z-depth. An area of 2090 μm by 1254 μm in each well was imaged and used to perform this quantification.

4) Line 159, the full name of β -gal should be given. The “mannose-inducible system” should be

introduced briefly at the beginning of the section of “EES secretes β -gal with delivery to the nuclei of J774A.1 cells”.

We have introduced the mannose inducible system with supporting literature and provided the full name of β -gal.

Results: Engineered *B. subtilis* LLO secretes β -gal with delivery to the nuclei of J774A.1 cells

The mannose-inducible system was amplified from the genome of the *B. subtilis* strain 168¹⁹ to provide a genetic switch to control protein production. The mannose-inducible system was chosen due to the characterized uptake of this sugar in mammalian cells²⁰.

5) In Fig. 4, the difference between ‘mannose –NLS’ and ‘mannose NLS’ is not obvious in fluorescence microscopy figure. The effect of nuclear localization is not obvious without nucleus staining. With regards to the nuclear SNR, how the authors distinguished the nuclear signal and the cytoplasmic signal?

We have included an example of how a nuclear stain was used to guide region of interest drawing around the nucleus to determine nuclear SNR as Fig. S3. When SV40 NLS was discovered (reference 41), the authors also observed β -gal accumulating in the nucleus without any NLS signal indicating an intrinsic signal even on a bacterial protein.

Results: Fig. S3

Reviewers' comments:

Reviewer #1 (Remarks to the Author):

The authors have greatly improved the clarity of their manuscript and the inclusion of additional controls has addressed my original concerns regarding the effects of the TF-producing strains.

Minor:

Please add the number of times each experiment was performed and whether the data presented are representative or pooled from multiple experiments, preferably to the figure legends.

Reviewer #2 (Remarks to the Author):

The revised manuscript successfully addresses the major concerns I originally had with this article: the claims in the sections of cell marker, cytokine, and chemokine expression have been toned down, and the comparisons are made with respect to the LLO strain while acknowledging that a significant portion of the observed effect is due to the mere presence of the EES. Furthermore, the experiments regarding expression and localization of LacZ and TFs now have proper quantifications and controls. There are still some smaller issues and clarifications I would like to ask the authors to address, after which I would be comfortable recommending publication of the manuscript.

- Fig 1: Removing the membrane stain layer from the zoomed-in images really helps with distinguishing colocalization of LAMP-1 and *B. subtilis*. It would be nice if the same could be done with Movies S1 and S2, or to create additional movies with/without the membrane stain layer.
- Line 145: Briefly mention what method was used to assess host cell viability.
- Line 147: Define MOI
- Can the authors comment on why they observed doubling of the LLO strain only at 10:1 MOI? at 25:1 MOI the increase is much more modest, and at 50:1 there is even a small decrease.
- If > 90% of the host cells contain LLO cells, why are there so many hosts that don't show *B. subtilis* cells in Fig. 2?
- Line 170: "Protein secretion from was used..." from what?
- Is the LLO still under IPTG control during the beta-gal experiments and after?
- Line 186: "and for this reason was for the additional studies". Possible typo, otherwise this phrase does not make sense to me.
- For Fig. 4 and S7, are the error bars calculated from biological replicates? If so, how many? Or are they calculated from the number of nuclei analyzed from images of a single experiment? Or something else?
- Fig 5: the flow cytometry scatter plots show that samples with LLO-KG +mannose have significantly fewer cells than the rest. Can the authors comment on this?
- The cytokine and chemokine section would benefit from a brief description of what the expected effects are for M1 and M2 induction.

Reviewer #3 (Remarks to the Author):

The response letter and revised manuscript satisfied this referee.

Final reviewer's comments:

The authors would like to thank the reviewers for insightful comments and suggestions for improving the manuscript. We have included direct responses (in blue text) in this document and made notes on where we included the updated text and data analysis in the manuscript files (red text).

Reviewer #1 (Remarks to the Author):

The authors have greatly improved the clarity of their manuscript and the inclusion of additional controls has addressed my original concerns regarding the effects of the TF-producing strains.

Minor:

Please add the number of times each experiment was performed and whether the data presented are representative or pooled from multiple experiments, preferably to the figure legends.

We have added additional n values to each figured legend and indicated whether the experiment was representative or pooled as was done in Fig. 5.

Some examples:

Fig. S7

Plotted data is mean \pm SD from $n = 158.4 \pm 63.6$ (SD) random individuals in a representative experiment; ***** $p < 0.000001$.

Fig. 6/S10

Plotted data is mean \pm SD from $n = 3$ biological replicates;

Reviewer #2 (Remarks to the Author):

The revised manuscript successfully addresses the major concerns I originally had with this article: the claims in the sections of cell marker, cytokine, and chemokine expression have been toned down, and the comparisons are made with respect to the LLO strain while acknowledging that a significant portion of the observed effect is due to the mere presence of the EES. Furthermore, the experiments regarding expression and localization of LacZ and TFs now have proper quantifications and controls. There are still some smaller issues and clarifications I would like to ask the authors to address, after which I would be comfortable recommending publication of the manuscript.

- Fig 1: Removing the membrane stain layer from the zoomed-in images really helps with distinguishing colocalization of LAMP-1 and *B. subtilis*. It would be nice if the same could be done with Movies S1 and S2, or to create additional movies with/without the membrane stain layer.

After looking at the video analysis further, we have decided to keep the membrane stain in the videos as the videos serve to highlight the location of *B. subtilis* LLO in relation to the cell membrane in z-depth. The membrane stain helps to provide contextual understanding of where the bacteria are inside the cell and removing it causes ambiguity in z-depth.

- Line 145: Briefly mention what method was used to assess host cell viability.

We have mentioned that both the MTS assay and flow cytometry are using to assess host cell viability.

Host cell viability was assessed by an MTS assay and flow cytometry after delivery of the LLO strain at different multiplicity of infection (MOI) at 2 different time points.

- Line 147: Define MOI

We have defined MOI as multiplicity of infection.

Host cell viability was assessed by an MTS assay and flow cytometry after delivery of the LLO strain at different multiplicity of infection (MOI) at 2 different time points.

- Can the authors comment on why they observed doubling of the LLO strain only at 10:1 MOI? at 25:1 MOI the increase is much more modest, and at 50:1 there is even a small decrease.

We observed that with the lower number of bacteria per cell at the first time point allowed for visualization of the doubling between the two time points. A balance was observed at 25:1 where approximately 11 bacteria were stable in the cells without losing much cell viability so in the whole population the bacteria per cell was relatively stable or increased slightly. However at 50:1, there was a significant loss in host cell viability specifically in cell containing the most bacteria so over the whole population the cells left remaining tended to have less bacteria which led to those cells remaining viable indicating 25:1 was an appropriate balance of many bacteria while maintaining viability of the host.

- If > 90% of the host cells contain LLO cells, why are there so many hosts that don't show *B. subtilis* cells in Fig. 2?

We appreciate this observation and this was the motivation for doing population quantification because of some of the heterogeneity seen throughout any random area of imaging in the population. We used this representative image from confocal because it demonstrated the variety of interactions that occur when the IPTG is added and the *B. subtilis* LLO strain is escaping phagosomal destruction as a non-pathogenic organism. Sometimes the cells escape efficiently and in other it takes more time giving an accurate representation of some of the intricacies observed in the population.

- Line 170: "Protein secretion from was used..." from what?

We have clarified that this was from the *B. subtilis* LLO strain.

Protein secretion from *B. subtilis* LLO was used to further demonstrate bacterial viability within the cytoplasm and as an initial demonstration of protein delivery to the nucleus.

- Is the LLO still under IPTG control during the beta-gal experiments and after?

Yes the *B. subtilis* LLO strain is still being regulated by IPTG throughout the experiments. We have added some additional comments in the text to clarify this question and clarify the sentence mentioned in line 186 below.

- Line 186: "and for this reason was for the additional studies". Possible typo, otherwise this phrase does not make sense to me.

The mannose-inducible system was amplified from the genome of the *B. subtilis* strain 168¹⁹ to provide a genetic switch specifically for controlling protein delivery to nucleus.

At 3 h, the mannose-inducible system was more efficient than the IPTG-inducible system as indicated by the intensity of fluorescent signals from the nuclei (Figure S4). For this reason and providing a second control switch specifically for protein delivery, the mannose inducible system was used for the additional studies.

- For Fig. 4 and S7, are the error bars calculated from biological replicates? If so, how many? Or are they calculated from the number of nuclei analyzed from images of a single experiment? Or something else?

This error bars represent number of individuals quantified in a single experiment. Preliminary experiments were performed for each figure and demonstrated the trends observed in the final experiment by quantifying random individuals in random imaged areas as described in the methods (β -gal antibody staining, ICC confirming EES manufacturing and delivering TFs). We have added the n values to the figure legends as well.

Example:

Fig. S7

Plotted data is mean \pm SD from n = 158.4 \pm 63.6 (SD) random individuals in a representative experiment; *****p<0.000001.

- Fig 5: the flow cytometry scatter plots show that samples with LLO-KG +mannose have significantly fewer cells than the rest. Can the authors comment on this?

The representative dot plot chosen for the 24h LLO-KG +mannose happened to be the one sample (from the biological triplicate) with the least amount of cells analyzed, although still within the range of all other conditions. Samples chosen to represent each condition as a dot plot had the median value of the three.

The 48h LLO-KG +mannose did have a lower number of cells overall. There was no difference in viability when compared to the other conditions which contained EES. This could have been due to cell loss during wash steps.

- The cytokine and chemokine section would benefit from a brief description of what the expected effects are for M1 and M2 induction.

The expected effects for M1 and M2 induction are thoroughly described in the discussion section with references to guide the reader regarding typical cytokines which can broadly

identify polarization. Expected and unexpected results are discussed here as well. With the variety of cytokine and chemokines analyzed along with complexities in classifying these analytes, we do not believe we can accurately guide the reader through this in the results section but made sure to do so in discussion.

Reviewer #3 (Remarks to the Author):

The response letter and revised manuscript satisfied this referee.

Reviewers' comments:

The authors would like to thank the reviewers for insightful comments and suggestions for improving the manuscript. We have included direct responses (in blue text) in this document and made notes on where we included the updated text and data analysis in the manuscript files (red text).

Reviewer #1 (Remarks to the Author):

Madsen and Makela et al report *Bacillus subtilis* strains engineered to secrete transcription factors with the goal of using these strains to control gene expression in mammalian cells. The creation of these transcription factor(TF)-secreting variants and the demonstration that they can deliver transcription factors into the nuclei of macrophages is quite a feat and has exciting potential. The strain development work is interesting, and the experiments are well performed. However, their data for functional readouts of how the TF-secreting strains impact M1/M2 polarization (Fig 5 – mac surface markers and 6 -mac cytokine production) suggest that these strains are not functioning as anticipated. Rather, their data suggest that the EES/LLO chassis (i.e., non-TF secreting) causes a marked innate immune response in macrophages with cytokine responses that dwarf that of polarized M1 and M2 macrophages. This finding means it is difficult to interpret whether the observed changes in macrophage-derived cytokine levels and CD86/CD206 expression between EES and TF-secreting EES actually represent controlled regulation of gene expression or simply changes in the innate response to the EES. As such major revisions are necessary, but overall, with these changes, I believe this paper could make an important contribution to the field.

Thank you for the interest in the work and we will try to clarify why we believe there is innate response to the LLO strain and specific response that are changed as a result of the TF strains. We will address these concerns below through the more detailed comments.

Major:

Central to the author's manuscript is the idea that gene expression can be modulated by EES and as proof of principle the authors attempt to use their TF-secreting EES to polarize macrophages toward M1 or M2. However, the authors find that EES treatment alone causes macrophages to produce various cytokines (e.g., IL-10, IL-12p40, TNF, G-CSF, IL-1b) at levels, well above their positive controls for macrophage polarization. In some cases (e.g., TNF and G-CSF) cytokine levels are multiple orders of magnitude above the control. The author's

discussion of this important finding is relatively cursory, and they should elaborate on what they think is occurring. It seems likely that this is an innate response to EES/LLO infection. Does this occur because the strain is escaping the phagosome? Have the authors compared the cytokine response/immunophenotype (CD86, CD206) of macrophages infected with EES +/- LLO?

The LLO strain causing high expression of the cytokines/chemokines comparatively to the controls and to the marker expression was an important finding. A few explanations of why the LLO strain is causing cytokine/chemokine production to be even magnitudes above the controls include the difference between Toll-like receptor (TLR)2 and TLR4 in determining cytokine response, the impact of live bacteria compared to just single molecule agonists and the patterns of impact on cytokine/chemokines from the positive controls. Live bacteria have been shown to cause a higher stimulation of macrophages compared to just single molecules (reference in manuscript response below) and that combined with both the amount of bacteria and TLR2 also being stimulated with the bacteria which can cause certain cytokines and chemokine responses to be much higher than TLR4 inducers (references in manuscript response below) all contribute to the results displayed in this study.

Discussion: 5th paragraph

Overall, dynamic responses were seen in many of the cytokines and chemokines in response to the LLO strain with the LLO-SK and LLO-KG strains able to modify expression further in comparison to the LLO strain in some key examples. One component of the dynamic response was the elevated expression of cytokines and chemokines in response to the bacteria compared to the positive controls of IFN- γ and LPS (M1+) or IL-4 and IL-13 (M2+). Additionally, the difference in response to the bacteria conditions compared to the positive controls was larger with the cytokines and chemokines than the surface markers. Enhanced signaling protein (cytokines and chemokines) response from macrophages and other professional antigen presenting cells compared to surface marker expression has been observed previously in response to bacterial or signaling molecule stimulus which was further shown in this study⁶³⁻⁶⁵. Other factors that could contribute to the complex response involves *B. subtilis* stimulating Toll-like receptor 2 (TLR2) compared to IFN- γ and LPS stimulating TLR4⁶⁶. Additionally, viable bacteria can cause a more dynamic response than just bacterial products alone⁶⁷.

Previously, *Listeria monocytogenes* was shown to have specific patterns of inducing cytokine/chemokine production based on expression of LLO (references in manuscript response below). In P388D1 cells, a lymphoma monocyte/macrophage line similar to J774A.1, expression of LLO in the WT *L. monocytogenes* strain demonstrated induction of TNF- α , IL-6 and IL- α while the $\Delta hlyA$ strain did not induce these cytokines. Yet, the $\Delta hlyA$ strain did induce expression of IL- β and to the same level as the WT strain. Conversely, in bone marrow derived macrophages, the expression of these cytokines was not dependent upon LLO expression. Ultimately, the phagosome escape does play a role as demonstrated by our study and previous studies which indicates that the LLO strain is the best control for how much the J774A.1 cytokine/chemokine production is being impacted independently of the TFs. Yet, the LLO-SK and LLO-KG strains impacted the expression of the cytokines/chemokines in significant ways as compared to the LLO strain with similar and opposite trends indicating specificity to the TFs for expression differences.

Additionally, CD86/206 expression resulting from the LLO strain impact is less clear in the literature. Accordingly, we performed the flow cytometry again to include the -LLO strain. We observed no significant differences in response to the *B. subtilis* 168 strain (-LLO) compared to

the LLO strain in both surface markers at both time points. The new flow data included an updated version of the staining protocol which improved CD206 staining. We have recently performed additional CD206 optimization and found that CD206 staining was improved when performed without permeabilization (contrary to manufacturers suggestion); this was implemented in the most recent flow cytometry staining and subsequent data.

Results: Fig. 5

Results: Modulation of J77A.1 cell cytokine and chemokine expression using engineered *B. subtilis* LLO strains

The *B. subtilis* strain 168 did not change marker expression in comparison to the LLO strain and a $\Delta hlyA$ mutant (no LLO expression) of *Listeria monocytogenes* has been shown to either not change cytokine and chemokine expression in a comparable macrophage cell line to J77A.1, or change expression of these proteins equally to the wild-type *L. monocytogenes* in bone marrow derived macrophages^{43,44}.

Discussion: 3rd paragraph

CD86 and CD206 expression was impacted by the bacterial conditions and the LLO-SK and LLO-KG were observed to regulate CD206 (Fig. 5) expression as expected based on the known activity of the TFs. At 24 h, CD86 response was more complex as it is expected to be upregulated as a general response to the bacteria. However, by 48 h the LLO-SK strain had increased the expression significantly compared to the LLO strain (Fig. 5, S8). The *B. subtilis* 168 strain also increased expression between 24 and 48 h but not significantly compared to the LLO strain. The surface protein CD206 has been shown to recognize the surface carbohydrates of pathogens and be triggered by proteases produced by *B. subtilis*^{46,47}.

Methods:

After addition of the engineered *B. subtilis* LLO strains and controls, followed by incubation for 24 or 48 h, cells were collected and stained in a 96-well round bottom plate. Treatment addition was staggered so that flow staining and data acquisition could be performed at the same time for 24 and 48 h cells. All staining steps were performed in 100 μ L volume at 4°C in the dark. Samples were first incubated with Zombie NIR viability dye (1:750, Biolegend) for 20 min. Cells were washed once with flow buffer, followed by incubation with TruStain FcX™ PLUS (anti-mouse CD16/32) Antibody (Biolegend, Cat#156603; 0.25 μ g/sample) for 10 min. Alexa Fluor® 647 anti-mouse CD86 Antibody (0.125 μ g/sample; Biolegend; Cat#105020) and FITC anti-mouse CD206 (MMR) antibody (0.1 μ g/sample, Biolegend; Cat#141703) were then added and incubated for 20 min. Cells were washed twice with flow staining buffer and fixed with 4% PFA for 10 min and resuspended in a final volume of 100 μ l for flow cytometry analysis using the Cytex Aurora spectral flow cytometer (Cytex).

The TF-secreting strains reduce or further increase macrophage cytokine production in some cases, however, in general, the changes do not appear to align with the predicted outcome for the EES-SK and EES-KG i.e. polarization toward M1 versus M2. Changes in CD86 and CD206 levels also do not appear to align the cytokine production levels is also suggestive of an alternate explanation. Although I found the author's discussion of their hypothesis relating to EES-SK and EES-KG effects on macrophage polarization to be somewhat unclear and this should be remedied. This is important given that "specificity" is key to how the authors state they

would like EES should operate. Specifically, Line 180 – EES-SK or KG - the authors should provide the rationale for why they chose these TF pairs to engineer into EES in the results i.e. SK because presumably, they hypothesized this would polarize the macs to M1 and KG combination would polarize to M2 and whether or no the data was consistent with that hypothesis. They should also write out the SK and KG transcription factor pairs in full the first time they are used before describing the results of the TF immunofluorescence in Fig S4.

We have expanded the introduction and results to provide clearer rationale and hypothesis of expected results. We included more of the rationale for the chosen transcription factors in the introduction and referenced back to this in results and discussion. We can also include more rationale again in results if necessary. We have also written out the full names in the results section. We have also included the discussion on the difference in marker expression versus cytokine/chemokine response in with the previous comment in the discussion section.

Introduction:

Therefore, the EES was designed to express and deliver TFs in two operons for modulating macrophage phenotype towards pro- or anti-inflammatory states. One operon encodes the TFs signal transducer and activator of transcription 1 (STAT-1) and Krüppel-like factor 6 (KLF6) which induce a general response to an inflammatory state in macrophages, and the second encodes Krüppel-like factor 4 (KLF4) and GATA binding protein 3 (GATA-3) which are both characterized to drive an anti-inflammatory response in macrophages^{33–37}. Both STAT-1 and KLF4 are upstream regulators that impact several mechanisms while KLF6 and GATA-3 are more specific in regulation which lends to a dual approach towards driving the desired cell fates^{33–37}. When expressed from intracellular EES these TFs altered patterns of cell surface markers, cytokine and chemokine expression with some patterns of modulation towards anti- or pro-inflammatory phenotypes which indicates that the EES may be used to direct immune cell function and be used to elucidate mechanisms of macrophage response to intracellular bacteria.

Results: Modulation of J77A.1 cell cytokine and chemokine expression using engineered *B. subtilis* LLO strains

B-gal delivery by the LLO-*lacZ* strains demonstrated the ability of an intracellular EES to functionally deliver protein with transcriptional regulation Accordingly, the next step is to deliver protein to alter host cell fate and TFs were chosen. The LLO strain was engineered to secrete STAT-1 and KLF6 (LLO-SK; pro-inflammatory) or KLF4 and GATA-3 (LLO-KG; anti-inflammatory) with the same approach as β -gal to impact macrophage function (Fig. S5). Introduction of these TFs into the genome of the *B. subtilis* LLO minimally impacted growth rates, did not change the impact on J77A.1 cell viability compared to the LLO strain and did not impact the ability of these new strains to escape destruction of phagosomes and persist in macrophages (percent cells containing bacteria and distribution of the fluorescence intensity relating to number of bacteria per cell) (Fig. S6). Immunofluorescence quantification confirmed production and delivery of STAT-1 and KLF6 or KLF4 and GATA-3 in cells containing LLO-SK or LLO-KG, respectively after 3 h of TF delivery (Fig. S7). Quantification of nuclear SNR identified increases in fluorescence in all four TFs after delivery from the LLO-SK and LLO-KG strains with and without addition of D-mannose. D-mannose significantly increased delivery of

all four TFs. Additionally, the appropriate positive controls produced some increase in nuclear SNR but significantly less than the LLO-SK and LLO-KG strains over the 3 h.

Discussion: 2nd paragraph

During immune responses, STAT-1 impacts pro-inflammatory response as a strong modulator that is directly upregulated by the presence of pro-inflammatory IFN- γ and cytokines which was also shown in 3 h in this study (Fig. S7) while KLF6 has been shown to be upregulated in M1 polarized macrophages^{33,34,37}. The activity of KLF4 is directly stimulated by the signal cascade from the cytokine IL-4 and while this stimulation did not produce a significant immunofluorescent result in 3 h (Fig. S7), it is known to promote M2 polarization and GATA-3 is highly upregulated in M2 macrophages^{35,36}. The LLO-SK and LLO-KG strains produced and delivered more of these TFs to the nuclei of J774A.1 cells than using the known signal cascade inducers to cause the host cells to produce these TFs. Accordingly, this demonstrated the potential of *B. subtilis* LLO to impact the host cell function within the 3 h timeframe of interaction between the LLO strain and host cells.

Discussion paragraphs 3 and 4 also expand on the discussion of the expected versus unexpected patterns more

Discussion: 5th paragraph

Overall, dynamic responses were seen in many of the cytokines and chemokines in response to the LLO strain with the LLO-SK and LLO-KG strains able to modify expression further in comparison to the LLO strain in some key examples. One component of the dynamic response was the elevated expression of cytokines and chemokines in response to the bacteria compared to the positive controls of IFN- γ and LPS (M1+) or IL-4 and IL-13 (M2+). Additionally, the difference in response to the bacteria conditions compared to the positive controls was larger with the cytokines and chemokines than the surface markers. Enhanced signaling protein (cytokines and chemokines) response from macrophages and other professional antigen presenting cells compared to surface marker expression has been observed previously in response to bacterial or signaling molecule stimulus which was further shown in this study^{63–65}. Other factors that could contribute to the complex response involves *B. subtilis* stimulating Toll-like receptor 2 (TLR2) compared to IFN- γ and LPS stimulating TLR4⁶⁶. Additionally, viable bacteria can cause a more dynamic response than just bacterial products alone⁶⁷. Altogether, even with the dynamic responses from the J774A.1 cells in response to complex signals, the LLO-SK and LLO-KG strains were able to change trends in macrophage surface marker, cytokine and chemokine expression in comparison to the LLO strain. This observation is further supported by not observing significant differences in J774A.1 viability, percentage of cells containing fluorescently labeled bacteria and distribution of fluorescence intensity (relating to number of bacteria per cell) when comparing the LLO, LLO-SK and LLO-KG strains (Fig. S6).

Whether these macrophage responses represent plasticity or a host response also merits some additional investigation. In the experiments shown in Fig 5 and 6, were EES-SK and EES-KG as infectious as the parental strain? Did macrophage survival change in response to the different EES strains? Did both strains reach the same titers as the parental? Related, the authors should report whether there was any metabolic burden (e.g. perform growth curves) for the introduced circuits and TFF production for EES-SK and EES-KG. Answers to these questions would help to resolve whether the authors are observing controlled changes of macrophage

polarization as they predict versus simply changes in the innate immune response to EES/LLO infection.

We have performed growth curves, further viability tests on J774A.1 by flow cytometry and characterization of amount of J774A.1 containing the engineered LLO strains and the fluorescence difference throughout the population of J774A.1 cells indicating number of bacteria in cells. Below we describe the results that we found that further support hypothesis by controlling for these other variables. This was an important suggestion that helped to support the hypothesis and ideas further.

Results: Fig. S6

Results: Engineered *B. subtilis* LLO transcription factor delivery and modulation of J77A.1 cell marker expression

Introduction of these TFs into the genome of the *B. subtilis* LLO minimally impacted growth rates, did not change the impact on J774A.1 cell viability compared to the LLO strain and did not impact the ability of these new strains to escape destruction of phagosomes and persist in macrophages (percent cells containing bacteria and distribution of the fluorescence intensity relating to number of bacteria per cell) (Fig. S6).

Discussion: 5th paragraph

Altogether, even with the dynamic responses from the J774A.1 cells in response to complex signals, the LLO-SK and LLO-KG strains were able to change trends in macrophage surface marker, cytokine and chemokine expression in comparison to the LLO strain. This observation is further supported by not observing significant differences in J774A.1 viability, percentage of cells containing fluorescently labeled bacteria and distribution of fluorescence intensity (relating to number of bacteria per cell) when comparing the LLO, LLO-SK and LLO-KG strains (Fig. S6).

Language throughout this manuscript relating to the ability of the strain to modulate macrophage gene expression and plasticity in a specific and predictive way such as Line 252 "In this study macrophage plasticity was modulated..." should be toned down or supported with further experiments as discussed above.

We agree with this point. We used the added studies and literature to provide more clarity on the study and rephrased many of these areas to indicate impact on what we were predicting but the complexity that was shown. We focused more on the trends that were seen which did follow some patterns predicted and even those that might have been unexpected could be explained or alluded to based on the TFs. Some examples shown below:

Results: Engineered *B. subtilis* LLO transcription factor delivery and modulation of J77A.1 cell marker expression

To determine the impact of engineered *B. subtilis* LLO strain TF delivery on J774A.1 modulation, surface marker expression was examined in cells containing various strains (168 strain, LLO strain, LLO-SK and LLO-KG strains with and without mannose) and compared to M0 and M1+/M2+ polarized J774.1 cells at 24 or 48 h post-incubation. As determined by flow cytometry, engineered *B. subtilis* LLO strains delivering TFs exhibited different patterns of J774A.1 surface marker expression when compared to the LLO strain (Fig. 5, S8).

Results: Modulation of J774A.1 cell cytokine and chemokine expression using engineered *B. subtilis* LLO strains

LLO-SK and LLO-KG were shown to alter J774A.1 cytokine and chemokine expression in comparison to LLO strain and positive controls (Fig. 6, S10). Addition of LLO-SK and LLO-KG strains to J774A.1 cells led to differential expression of cytokines compared to the cytokine profile observed for the LLO strain, as shown by levels of interleukin (IL)-10, IL-12p40 and tumor necrosis factor alpha (TNF- α) (Fig. 6A-D, S10). LLO-SK downregulated granulocyte colony stimulating factor (G-CSF) relative to the LLO strain and LLO-KG (Fig. 6D-E). Although cytokine production was generally higher at 24 h compared to 48 h post bacterial exposure for most cytokines and chemokines, significant differences were observed at both time points from LLO-SK and LLO-KG in comparison to LLO strain and positive controls.

Discussion: 3rd paragraph

Even with these complexities, clear trends were observed that suggest the LLO-SK and LLO-KG impacted surface marker expression in comparison to the LLO strain.

Discussion paragraph 4 also talks in detail about the patterns of what was expected and seen and what was unexpected and seen

Discussion: 5th paragraph

Altogether, even with the dynamic responses from the J774A.1 cells in response to complex signals, the LLO-SK and LLO-KG strains were able to change trends in macrophage surface marker, cytokine and chemokine expression in comparison to the LLO strain. This observation is further supported by not observing significant differences in J774A.1 viability, percentage of cells containing fluorescently labeled bacteria and distribution of fluorescence intensity (relating to number of bacteria per cell) when comparing the LLO, LLO-SK and LLO-KG strains (Fig. S6).

Throughout the manuscript the phrase “modulation of gene expression changes” or variants thereof is used to refer to the effect of EES delivered transcription factors. However, at no point is gene expression measured. Rather readouts are performed at the protein level. The authors should either measure transcript levels of their target genes or re-phrase accordingly.

While TFs do directly impact the expression of genes, we agree that we should be more specific and more appropriately discuss the impact on marker expression and cytokine/chemokine production instead of using the gene expression phrasing because we are not measuring transcript levels. We believe RNA-seq would be ideal for understanding the complex array of transcriptomic changes caused by the EES but we believe these resources are better allocated towards primary macrophages and as a result is outside the scope of the study. We have altered the title along with results and conclusions to account for this. Some examples:

Title:

Engineered endosymbionts that alter mammalian cell surface marker, cytokine and chemokine expression

Introduction:

When expressed from intracellular EES these TFs altered patterns of cell surface markers, cytokine and chemokine expression with some patterns of modulation towards anti- or pro-inflammatory phenotypes which indicates that the EES may be used to direct immune cell function and be used to elucidate mechanisms of macrophage response to intracellular bacteria.

Results: Modulation of J77A.1 cell cytokine and chemokine expression using engineered *B. subtilis* LLO strains

LLO-SK and LLO-KG were shown to alter J774A.1 cytokine and chemokine expression in comparison to LLO strain and positive controls (Fig. 6, S10). Addition of LLO-SK and LLO-KG strains to J774A.1 cells led to differential expression of cytokines compared to the cytokine profile observed for the LLO strain, as shown by levels of interleukin (IL)-10, IL-12p40 and tumor necrosis factor alpha (TNF- α) (Fig. 6A-D, S10).

Discussion: 5th paragraph

Altogether, even with the dynamic responses from the J774A.1 cells in response to complex signals, the LLO-SK and LLO-KG strains were able to change trends in macrophage surface marker, cytokine and chemokine expression in comparison to the LLO strain. This observation is further supported by not observing significant differences in J774A.1 viability, percentage of cells containing fluorescently labeled bacteria and distribution of fluorescence intensity (relating to number of bacteria per cell) when comparing the LLO, LLO-SK and LLO-KG strains (Fig. S6).

Minor:

In general, the Results are quite well written, however, the authors should add some description of each experimental setup. While it is acknowledged that the experimental details are present in the methods, an abbreviated description at the beginning of each Results subsection would aid the reader in understanding what was done without having to flick back and forth between the methods and results. Moreover, describing the rationale within the Results section would help the reader follow the author's line of reasoning for the experiments being performed.

We agree and we have incorporated brief overviews of both rationale and experimental details into the results section for further clarity. Some examples:

Results: Engineered *B. subtilis* LLO secretes β -gal with delivery to the nuclei of J774A.1 cells

Protein secretion from the LLO strain was tested to further elucidate viability within the cytoplasm and initial demonstration of protein delivery to the nucleus. *B. subtilis* LLO was engineered to produce and secrete β -galactosidase (β -gal) (LLO-*lacZ*) through the twin-arginine translocation (Tat) pathway by synthesizing the PhoD signal peptide⁴⁰ and the amino acids from 126-132 of the simian virus (SV) 40 nuclear localization signal (NLS)⁴¹.

Results: Engineered *B. subtilis* LLO transcription factor delivery and modulation of J77A.1 cell marker expression

β -gal delivery by the LLO-*lacZ* strains demonstrated the ability of an intracellular EES to functionally deliver protein with transcriptional regulation. Accordingly, the next step is to deliver protein to alter host cell fate and TFs were chosen. The LLO strain was engineered to secrete STAT-1 and KLF6 (LLO-SK; pro-inflammatory) or KLF4 and GATA-3 (LLO-KG; anti-inflammatory) with the same approach as β -gal to impact macrophage function (Fig. S5).

Line 124: EES term. In the intro, the authors use “EES” broadly to refer to engineered endosymbionts but elsewhere in the manuscript, starting in the Results, it is also used to refer to the engineered *B. subtilis* LLO strain. The authors should distinguish between the two for clarity. I would suggest reserving EES as a general term a new name used for the strain/platform developed in this manuscript.

We agree that this would help with clarity. We have kept the EES term for the broader sense and we have changed the specific strain names to *B. subtilis* LLO (LLO strain), *B. subtilis* LLO *Stat-1Klf6* (LLO-SK) and *B. subtilis* LLO *Klf4Gata-3* (LLO-KG). We also refer to these strains as the engineered *B. subtilis* LLO strains or TF expressing strains when appropriate. This is demonstrated by many of the examples above.

Lie 154 Can the authors discuss whether the anti420 subtilisin antibody stains viable *B. subtilis* exclusively? If it does not, I think a gentamicin protection assay (or similar) is warranted to demonstrate that the intracellular bacteria are indeed viable as the authors claim.

The anti-subtilisin antibody is just specific to the protein so it does not select for viable cells necessarily even though *B. subtilis* would have needed to be viable to make the protein. However, we believe the live cell imaging data in Fig 3 validates the viability within the cytoplasm. We were able to observe *B. subtilis* LLO replicating in real time within the same macrophage over time with many examples of this occurring. We believe this provides a clear demonstration of intracellular viability that gentamicin protection assays suggest and can only be certain of if the antibiotic is applied for long enough and high enough concentration to not allow for extracellular bacteria to grow. Additionally, the protein delivery of β -gal demonstrates this viability further and the supernatant control indicates that only viable intracellular bacteria contribute to protein localization to the nucleus which is demonstrated again with the TFs (Fig. 4, S6).

Line 167 “high concentration of gentamicin” – did this effectively kill the intracellular bacteria? A schematic of the synthetic operons introduced into EES would help the reader better understand these circuits and their regulation. If the figure limit is an issue, perhaps this could accompany figure 1.

Yes, the high concentration of gentamicin (25 μ g/mL) effectively killed the intracellular bacteria and allowed the J774A.1 cells to recover without rods being present at 24 hours (Fig. 4, S3).

Results: Engineered *B. subtilis* LLO secretes β -gal with delivery to the nuclei of J774A.1 cells

Further, the addition of a high concentration of gentamicin (25 μ g/mL) after 3 h of induced β -gal production rescued the J774A.1 cells from overgrowth of the LLO strain, allowing for another 21 h of trafficking of protein to the nucleus by eliminating intracellular bacteria (Fig. 4, lower image panels; Fig. S3).

The schematic has been included in the supplemental figures as Fig. S5.

Typographical:

Line 405: (Lonza, xx).

We have addressed changed this to include country of origin for mycoplasma kit used.

Methods: Delivery protocol for *B. subtilis* LLO and engineered strains

Cells were tested negative for mycoplasma using the MycoAlert PLUS Mycoplasma Detection Kit (Lonza, USA).

Reviewer #2 (Remarks to the Author):

In this study, the authors attempt to control macrophage polarization using engineered bacteria. First, they engineer *B. subtilis* such that, upon being endocytosed by macrophages, can escape the phagosome and survive in the cytoplasm of the macrophages. Then, they evaluate whether these *B. subtilis* can express proteins that can be imported into the nucleus of the macrophage. Finally, they attempt to influence macrophage polarization by engineering *B. subtilis* to express cell fate-determining transcription factors.

While the proposed idea is intriguing, I find that the majority of the data does not support the conclusions the authors want to make. In general, it seems that most of the effects shown are driven by the mere presence of *B. subtilis* instead of TF expression or addition of the inducer mannose. Moreover, when gene- and mannose-dependent effects are observed, it is sometimes in the opposite direction that would be expected (e.g. Fig 6B and C). While the authors acknowledge some of these irregularities in the discussion section, the rest of the paper is mostly written as if these irregularities did not exist. For example, in line 205 the authors claim that "EES-SK and EES-KG were shown to modulate mammalian gene expression as determined by cytokine profiling (Fig 6)", but don't acknowledge in the same paragraph that most of these cytokines are released in the presence of EES that does not express any TFs. A few other examples:

Thank you for the interest in the work and we clarified our thoughts on the results observed and tried to provide thorough explanation throughout the paper with more writing about why the differences that we see can be explained. We have provided more analysis and experiments to control for extraneous variables and accounted for the concerns in flow cytometry by improving the staining method for CD206 (will talk more in the area specific to that comment). For cytokines and chemokines, all conditions were compared to the LLO strain. The TFs bearing strains did alter expression in comparison to that strain and it is expected that the LLO strain alone would increase many of the cytokines and chemokines which was reported in previous literature when studying *Listeria monocytogenes*. While the macrophage response is complex and is impacted by several factors, we fortunately did see trends from the TF strains.

Results: Engineered *B. subtilis* LLO transcription factor delivery and modulation of J77A.1 cell marker expression

Fig. 5, S6

Introduction of these TFs into the genome of the *B. subtilis* LLO minimally impacted growth rates, did not change the impact on J77A.1 cell viability compared to the LLO strain and did not impact the ability of these new strains to escape destruction of phagosomes and persist in macrophages (percent cells containing bacteria and distribution of the fluorescence intensity relating to number of bacteria per cell) (Fig. S6). Immunofluorescence quantification confirmed

production and delivery of STAT-1 and KLF6 or KLF4 and GATA-3 in cells containing LLO-SK or LLO-KG, respectively after 3 h of TF delivery (Fig. S7). Quantification of nuclear SNR identified increases in fluorescence in all four TFs after delivery from the LLO-SK and LLO-KG strains with and without addition of D-mannose. D-mannose significantly increased delivery of all four TFs. Additionally, the appropriate positive controls produced some increase in nuclear SNR but significantly less than the LLO-SK and LLO-KG strains over the 3 h.

To determine the impact of engineered *B. subtilis* LLO strain TF delivery on J774A.1 modulation, surface marker expression was examined in cells containing various strains (168 strain, LLO strain, LLO-SK and LLO-KG strains with and without mannose) and compared to M0 and M1+/M2+ polarized J774.1 cells at 24 or 48 h post-incubation. As determined by flow cytometry, engineered *B. subtilis* LLO strains delivering TFs exhibited different patterns of J774A.1 surface marker expression when compared to the LLO strain (Fig. 5, S8). There was no difference in surface marker expression between *B. subtilis* strain 168 and LLO strain at any time point. At 24 h post-incubation, a significant decrease in CD206 expression was observed in J774A.1 cells containing LLO-SK, with and without the addition of mannose ($p = 0.0048$; not shown on plot) compared to the LLO strain; the same trend was observed at 48 h (Fig. 5A, C). Conversely, CD206 MFI was increased by LLO-KG with and without mannose ($p = 0.0599$; not shown on graph) at 24 h comparable to the positive control (M2+) however the increase was not sustained at 48 h. CD86 expression demonstrated a similar impact with all bacterial conditions at 24 h in comparison to resting cells with no significant differences in comparison to M1+ (Fig. 5B, D). At 48 h, the LLO-SK strain significantly increased CD86 MFI in comparison to the LLO strain with or without mannose added. The *B. subtilis* 168 strain showed a significant difference in CD86 MFI in comparison to untreated control but not in comparison to the LLO strain. Differences in CD206 and CD86 expression between 24 and 48 h were analyzed to elucidate impact from each treatment over time (Fig. S8). CD206 expression was further increased in M2+ J774A.1 cells whereas the LLO-KG with and without mannose were not able to sustain CD206 expression and decreased. CD86 expression was increased at 48 h in the *B. subtilis* strain 168 and LLO-SK with and without mannose.

Results: Modulation of J77A.1 cell cytokine and chemokine expression using engineered *B. subtilis* LLO strains

Functional readouts for macrophage polarization and resulting fate change include shifts in cytokine and chemokine expression^{7,11,42} so profiling the impacts on these proteins provides further insight on the ability of LLO-SK and LLO-KG to differentially impact macrophage function compared to the LLO strain. The *B. subtilis* strain 168 did not change marker expression in comparison to the LLO strain and a $\Delta hlyA$ mutant (no LLO expression) of *Listeria monocytogenes* has been shown to either not change cytokine and chemokine expression in a comparable macrophage cell line to J774A.1, or change expression of these proteins equally to the wild-type *L. monocytogenes* in bone marrow derived macrophages^{43,44}. Therefore, the 168 strain was not used in this characterization. LLO-SK and LLO-KG were shown to alter J774A.1 cytokine and chemokine expression in comparison to LLO strain and positive controls (Fig. 6, S10). Addition of LLO-SK and LLO-KG strains to J774A.1 cells led to differential expression of cytokines compared to the cytokine profile observed for the LLO strain, as shown by levels of interleukin (IL)-10, IL-12p40 and tumor necrosis factor alpha (TNF- α) (Fig. 6A-D, S10). LLO-SK downregulated granulocyte colony stimulating factor (G-CSF) relative to the LLO strain and LLO-KG (Fig. 6D-E). Although cytokine production was generally higher at 24 h compared to 48

h post bacterial exposure for most cytokines and chemokines, significant differences were observed at both time points from LLO-SK and LLO-KG in comparison to LLO strain and positive controls.

Discussion:

In this study macrophage plasticity was modulated using TFs expressed from the EES. Cell surface markers are commonly used to differentiate M1 and M2 polarization^{7,11}. CD86 and CD206 expression was impacted by the bacterial conditions and the LLO-SK and LLO-KG were observed to regulate CD206 (Fig. 5) expression as expected based on the known activity of the TFs. At 24 h, CD86 response was more complex as it is expected to be upregulated as a general response to the bacteria. However, by 48 h the LLO-SK strain had increased the expression significantly compared to the LLO strain (Fig. 5, S8). The *B. subtilis* 168 strain also increased expression between 24 and 48 h but not significantly compared to the LLO strain. The surface protein CD206 has been shown to recognize the surface carbohydrates of pathogens and be triggered by proteases produced by *B. subtilis*^{46,47}. This would explain the increase in CD206 expression caused by the *B. subtilis* 168 strain and LLO strain in comparison to the untreated controls. Additionally, while CD86 is regulated directly by inflammatory responses to the engineered *B. subtilis* LLO, the plasticity of the macrophages could cause CD86 expression to change over time¹¹. It has been suggested that products of *B. subtilis* such as subblancin or exopolysaccharide (EPS), or the treatment of macrophages with *B. subtilis* spores can result in either M1 or M2 activation⁴⁸. Even with these complexities, clear trends were observed that suggest the LLO-SK and LLO-KG impacted surface marker expression in comparison to the LLO strain.

Although changes in cytokine and chemokine expression vary along the spectrum of macrophage polarization, there are well documented cytokines which are used to broadly identify M1 or M2 macrophages^{7,11,49}. These identifying cytokines and chemokines are important when studying disease as they provide information on the function and characteristics of the macrophage population^{42,50}. The M1 and M2 polarization classifications represent a trend in immune responses but complex and mixed signals such as bacterial stimulation and infection can cause these classifications to be incomplete or oversimplified^{49,51}. Complexities in macrophage activation, phenotype and plasticity were encountered in this study. In some instances, the LLO-SK and LLO-KG strains impacted the host cell as predicted in comparison to the LLO strain alone (Fig. 6, S10). The upregulation of IL-10 by LLO-KG and down regulation by LLO-SK was an expected result along with G-CSF being downregulated by LLO-SK^{35,37,52,53}. The LLO strain increasing IL-10 and G-CSF are explainable results as well as IL-10 and G-CSF have been shown to be produced in response to bacteria⁵³⁻⁵⁵. However, in some cases there were results that were not anticipated, and possibly pleiotropic effects of each TF need to be considered. The change in IL-12p40 levels is an example of an unexpected result based on known signaling cascades. IL-12p40 is known to be produced during inflammatory phenotypes by nuclear factor kappa light chain enhancer of activated B cells (NF-κB)⁵⁶. Therefore, LLO-KG resulting in an increase in production of IL-12p40 is unexpected; however, it is possible GATA-3 could play a role in IL-12p40 production. The IL-12p40 promoter has a characterized GATA binding site but GATA-3 has not been characterized in terms of impact on this promoter⁵⁶. The LLO-SK strain showed some downregulation at 48 h which could be due to STAT-1 driving alternative cytokine production such as IL-27p28^{56,57}. TNF-α downregulation by LLO-KG is another expected result³⁵ but downregulation by LLO-SK in comparison to the LLO strain

indicates further complexity and potential pleiotropic impact from STAT-1 because of TNF- α being regulated by NF- κ B⁵⁸⁻⁶⁰. Furthermore, there may have been metabolic impact on cytokine production due to the addition of D-mannose, which impairs glucose metabolism and is known to suppress the succinate-driven HIF1 α activation of IL-1 β leading to downregulation of IL-1 β ^{61,62}. Accordingly, results indicated that D-mannose had a significant impact on IL-1 β and other cytokines and chemokines which is important consideration for future EES studies that rely on sugar-inducible systems. Metabolic reprogramming is most likely playing a significant role in response to the chemical inducer of the engineered *B. subtilis* LLO TF operons and possibly the LLO strain alone⁶².

Overall, dynamic responses were seen in many of the cytokines and chemokines in response to the LLO strain with the LLO-SK and LLO-KG strains able to modify expression further in comparison to the LLO strain in some key examples. One component of the dynamic response was the elevated expression of cytokines and chemokines in response to the bacteria compared to the positive controls of IFN- γ and LPS (M1+) or IL-4 and IL-13 (M2+). Additionally, the difference in response to the bacteria conditions compared to the positive controls was larger with the cytokines and chemokines than the surface markers. Enhanced signaling protein (cytokines and chemokines) response from macrophages and other professional antigen presenting cells compared to surface marker expression has been observed previously in response to bacterial or signaling molecule stimulus which was further shown in this study⁶³⁻⁶⁵. Other factors that could contribute to the complex response involves *B. subtilis* stimulating Toll-like receptor 2 (TLR2) compared to IFN- γ and LPS stimulating TLR4⁶⁶. Additionally, viable bacteria can cause a more dynamic response than just bacterial products alone⁶⁷. Altogether, even with the dynamic responses from the J774A.1 cells in response to complex signals, the LLO-SK and LLO-KG strains were able to change trends in macrophage surface marker, cytokine and chemokine expression in comparison to the LLO strain. This observation is further supported by not observing significant differences in J774A.1 viability, percentage of cells containing fluorescently labeled bacteria and distribution of fluorescence intensity (relating to number of bacteria per cell) when comparing the LLO, LLO-SK and LLO-KG strains (Fig. S6). Future studies should focus on characterizing the impact of these strains in primary macrophages such as bone marrow derived macrophages to clearly understand application potential. In addition, more TF pairings informed by thorough studies that elucidate the complexity of macrophage response should be constructed and tested to optimize the response from the macrophages for the desired application.

- Fig 2: It is hard to see colocalization of red and purple spots (or the lack thereof). This is less of a problem in the SI movies but it is still not great. At the very least, I recommend using less similar colors.

We have improved Fig. 2 to help clarify the confocal imaging. We were informed by experts in the imaging core that pseudo coloring Cy5 to magenta is becoming a new standard when other orange to red wavelengths are being used. We decided to enhance Fig. 2 by withdrawing the membrane stain in some panels to give better viewing of the LAMP-1 and *B. subtilis* staining. We also slightly adjusted the z-depth projection of *B. subtilis* to reveal better visualization of rods in relation to phagosomal pockets.

Results: *B. subtilis* LLO escape from phagosomes of J774A.1 cells

Fig. 2

- Table S1: IPTG does not seem to make a significant difference in EES/J774A.1 and % infected, therefore I would not conclude that expression of LLO is what is driving EES numbers or replication.

We have adjusted the table to make this more clear. The %infected is more accurately the amount of J774A.1 cells containing EES which is calculated by determining fluorescent EES throughout the z-depth of each cell over the total number of cells imaged. The -IPTG condition is the only condition where bacteria per J774A.1 decreased between 1 h and 2 h post bacterial addition due to destruction in the phagosome. Fig. 2 (4 h after bacterial addition) shows minimal rods in -IPTG condition and more punctate signal along with the SI movies. This result was also shown during the *B. subtilis* LLO developmental studies.

Results: *B. subtilis* LLO escape from phagosomes of J774A.1 cells

Confocal microscopy confirmed the escape of *B. subtilis* LLO from phagosomes after uptake into J774A.1 cells¹². To further elucidate the mechanism of escape, *B. subtilis* LLO (magenta) localization was compared to LAMP-1³⁸ positive structures (phagosomes, red) in J774A.1 cells (green) with and without IPTG induction of LLO expression (+IPTG and -IPTG). The LAMP-1 protein is crucial for phagosomal assembly and therefore will reveal when the LLO strain is contained within the phagosomes and when the phagosomes have been disrupted³⁹. Without IPTG induction, few of the LLO strain were observed and many regions of punctate signal within LAMP-1 positive regions were observed (Fig. 2, zoom-solid line; Movie S2). In contrast, when LLO expression was induced many of the LLO strain were intact and present throughout the mammalian cells (Movie S2). Z-stack data analysis identified EES throughout the cytoplasm of J774A.1 cells and not associated with LAMP-1 positive structures (Movie S1). Accordingly, the expression of LLO when induced by IPTG allows *B. subtilis* LLO to access the cytoplasm of the host cells.

- Figure S4: expression of KLF6, KLF4, and GATA-3 are visible in the negative controls, and thus it is hard to conclude that the EES or addition of manose have any influence in expression of these regulators. Quantification of fluorescence levels may partially alleviate this issue.

We have quantified the fluorescence to help address this concern. These TFs are native to the host cells so some expression is expected and even serves as a control for the antibodies. Additionally, only in the LLO-SK and LLO-KG conditions with and without mannose are the bacteria also fluorescent indicating expression of the proteins. The mannose system is known to allow transcription even without the presence of mannose which is why the -mannose condition also demonstrates signal which follows Fig. 4.

Results: Engineered *B. subtilis* LLO transcription factor delivery and modulation of J774A.1 cell marker expression

Immunofluorescence quantification confirmed production and delivery of STAT-1 and KLF6 or KLF4 and GATA-3 in cells containing LLO-SK or LLO-KG, respectively after 3 h of TF delivery (Fig. S7). Quantification of nuclear SNR identified increases in fluorescence in all four TFs after delivery from the LLO-SK and LLO-KG strains with and without addition of D-mannose. D-

mannose significantly increased delivery of all four TFs. Additionally, the appropriate positive controls produced some increase in nuclear SNR but significantly less than the LLO-SK and LLO-KG strains over the 3 h.

Discussion: 2nd paragraph

During immune responses, STAT-1 impacts pro-inflammatory response as a strong modulator that is directly upregulated by the presence of pro-inflammatory IFN- γ and cytokines which was also shown in 3 h in this study (Fig. S7) while KLF6 has been shown to be upregulated in M1 polarized macrophages^{33,34,37}. The activity of KLF4 is directly stimulated by the signal cascade from the cytokine IL-4 and while this stimulation did not produce a significant immunofluorescent result in 3 h (Fig. S7), it is known to promote M2 polarization and GATA-3 is highly upregulated in M2 macrophages^{35,36}. The LLO-SK and LLO-KG strains produced and delivered more of these TFs to the nuclei of J774A.1 cells than using the known signal cascade inducers to cause the host cells to produce these TFs. Accordingly, this demonstrated the potential of *B. subtilis* LLO to impact the host cell function within the 3 h timeframe of interaction between the LLO strain and host cells.

- Figure S5: In panel A, are the thresholds chosen to represent some type of transition to the M1 or M2 states? Why do the negative controls start with high levels of CD86? Why does the M1+ positive control also have high CD206? It is hard to assess whether the samples under evaluation are doing anything if the controls are not behaving as expected.

We repeated the flow cytometry with an improved staining procedure to resolve the concern as mentioned in the first comment. We tested CD206 with/without permeabilization (manufacturer suggested to permeabilize cells) and we found that CD206 was much improved without permeabilization. The gating/thresholds in the previous data and new data were chosen based on FMO/single stained controls. We observed with this new staining procedure that the positive controls performed as expected and we also saw the TF expressing strains showing trends similar to those expected (Fig. 5).

Results: Engineered *B. subtilis* LLO transcription factor delivery and modulation of J774.1 cell marker expression

To determine the impact of engineered *B. subtilis* LLO strain TF delivery on J774A.1 modulation, surface marker expression was examined in cells containing various strains (168 strain, LLO strain, LLO-SK and LLO-KG strains with and without mannose) and compared to M0 and M1+/M2+ polarized J774.1 cells at 24 or 48 h post-incubation. As determined by flow cytometry, engineered *B. subtilis* LLO strains delivering TFs exhibited different patterns of J774A.1 surface marker expression when compared to the LLO strain (Fig. 5, S8). There was no difference in surface marker expression between *B. subtilis* strain 168 and LLO strain at any time point. At 24 h post-incubation, a significant decrease in CD206 expression was observed in J774A.1 cells containing LLO-SK, with and without the addition of mannose ($p = 0.0048$; not shown on plot) compared to the LLO strain; the same trend was observed at 48 h (Fig. 5A, C). Conversely, CD206 MFI was increased by LLO-KG with and without mannose ($p = 0.0599$; not shown on graph) at 24 h comparable to the positive control (M2+) however the increase was not sustained at 48 h. CD86 expression demonstrated a similar impact with all bacterial conditions at 24 h in comparison to resting cells with no significant differences in comparison to M1+ (Fig. 5B, D). At 48 h, the LLO-SK strain significantly increased CD86 MFI in comparison to the LLO

strain with or without mannose added. The *B. subtilis* 168 strain showed a significant difference in CD86 MFI in comparison to untreated control but not in comparison to the LLO strain. Differences in CD206 and CD86 expression between 24 and 48 h were analyzed to elucidate impact from each treatment over time (Fig. S8). CD206 expression was further increased in M2+ J774A.1 cells whereas the LLO-KG with and without mannose were not able to sustain CD206 expression and decreased. CD86 expression was increased at 48 h in the *B. subtilis* strain 168 and LLO-SK with and without mannose.

Discussion: 3rd paragraph

In this study macrophage plasticity was modulated using TFs expressed from the EES. Cell surface markers are commonly used to differentiate M1 and M2 polarization^{7,11}. CD86 and CD206 expression was impacted by the bacterial conditions and the LLO-SK and LLO-KG were observed to regulate CD206 (Fig. 5) expression as expected based on the known activity of the TFs. At 24 h, CD86 response was more complex as it is expected to be upregulated as a general response to the bacteria. However, by 48 h the LLO-SK strain had increased the expression significantly compared to the LLO strain (Fig. 5, S8). The *B. subtilis* 168 strain also increased expression between 24 and 48 h but not significantly compared to the LLO strain. The surface protein CD206 has been shown to recognize the surface carbohydrates of pathogens and be triggered by proteases produced by *B. subtilis*^{46,47}. This would explain the increase in CD206 expression caused by the *B. subtilis* 168 strain and LLO strain in comparison to the untreated controls. Additionally, while CD86 is regulated directly by inflammatory responses to the engineered *B. subtilis* LLO, the plasticity of the macrophages could cause CD86 expression to change over time¹¹. It has been suggested that products of *B. subtilis* such as sublancin or exopolysaccharide (EPS), or the treatment of macrophages with *B. subtilis* spores can result in either M1 or M2 activation⁴⁸. Even with these complexities, clear trends were observed that suggest the LLO-SK and LLO-KG impacted surface marker expression in comparison to the LLO strain.

Methods:

After addition of the engineered *B. subtilis* LLO strains and controls, followed by incubation for 24 or 48 h, cells were collected and stained in a 96-well round bottom plate. Treatment addition was staggered so that flow staining and data acquisition could be performed at the same time for 24 and 48 h cells. All staining steps were performed in 100 µL volume at 4°C in the dark. Samples were first incubated with Zombie NIR viability dye (1:750, Biolegend) for 20 min. Cells were washed once with flow buffer, followed by incubation with TruStain FcX™ PLUS (anti-mouse CD16/32) Antibody (Biolegend, Cat#156603; 0.25 µg/sample) for 10 min. Alexa Fluor® 647 anti-mouse CD86 Antibody (0.125 µg/sample; Biolegend; Cat#105020) and FITC anti-mouse CD206 (MMR) antibody (0.1 µg/sample, Biolegend; Cat#141703) were then added and incubated for 20 min. Cells were washed twice with flow staining buffer and fixed with 4% PFA for 10 min and resuspended in a final volume of 100 µl for flow cytometry analysis using the Cytex Aurora spectral flow cytometer (Cytex).

- Figure 6: Why do the M1+ and M2+ positive controls not show any signal?

We have included some panels in Fig. 6 with broken axis to clarify that the controls do causes changes and even magnitudes higher than background but just not to the extent of the LLO strain and TF expressing strains. Fig. S10 also shows more clarity on the impact from the

positive controls. We have also noted that the bacteria producing a substantial response in comparison to the positive controls is an explainable result that has been seen previously.

Discussion: 5th paragraph

Overall, dynamic responses were seen in many of the cytokines and chemokines in response to the LLO strain with the LLO-SK and LLO-KG strains able to modify expression further in comparison to the LLO strain in some key examples. One component of the dynamic response was the elevated expression of cytokines and chemokines in response to the bacteria compared to the positive controls of IFN- γ and LPS (M1+) or IL-4 and IL-13 (M2+). Additionally, the difference in response to the bacteria conditions compared to the positive controls was larger with the cytokines and chemokines than the surface markers. Enhanced signaling protein (cytokines and chemokines) response from macrophages and other professional antigen presenting cells compared to surface marker expression has been observed previously in response to bacterial or signaling molecule stimulus which was further shown in this study⁶³⁻⁶⁵. Other factors that could contribute to the complex response involves *B. subtilis* stimulating Toll-like receptor 2 (TLR2) compared to IFN- γ and LPS stimulating TLR4⁶⁶. Additionally, viable bacteria can cause a more dynamic response than just bacterial products alone⁶⁷.

In summary, it is hard for me to recommend publication of this paper in its current form. I suggest the authors to substantially revise their manuscript and be more forthcoming about the data irregularities.

We hope that our revisions, added analyses and experiments have provided more clarity on our observations and conclusions. We have made sure to indicate that we do see some expected results which is promising for the approach but a better understanding of macrophage response to bacteria and the TFs can improve studies that seek to design a targeted impact approach. We hope the EES can be a mechanism to understanding macrophage response.

One last point about line 263: *B. subtilis* 168 absolutely does sporulate, and it is frequently used as a model of sporulation. See, for example, <https://www.ncbi.nlm.nih.gov/pmc/articles/PMC178487/>

We have updated this area to acknowledge that the LLO strain and other engineered *B. subtilis* strains could be producing spores if the strains are under enough stress to do so (<https://www.ncbi.nlm.nih.gov/pmc/articles/PMC2819312/#R1>) in these conditions (this was an important point for clarification). We were still able to see trends in marker expression even with these other factors that could be involved.

Discussion: 3rd paragraph

It has been suggested that products of *B. subtilis* such as sublinicin or exopolysaccharide (EPS), or the treatment of macrophages with *B. subtilis* spores can result in either M1 or M2 activation⁴⁸. Even with these complexities, clear trends were observed that suggest the LLO-SK and LLO-KG impacted surface marker expression in comparison to the LLO strain.

Reviewer #3 (Remarks to the Author):

In this manuscript, *Bacillus subtilis* was developed as the chassis organism for engineered

endosymbionts (EES) that escape phagosome destruction, reside in the cytoplasm of mammalian cells, and secrete proteins that are transported to the nucleus to control host cell gene expression. Two sets of transcription factors Stat-1-Klf6 and Klf4-Gata-3 were recombined into the genome of the EES and expressed from regulated promoters. Controlled expression of the mammalian proteins from the EES in the cytoplasm of J774A.1 cells directed gene expression and modulated the host cell fates toward anti- or pro- inflammatory phenotypes by each of the distinct transcription factor pairs. Expressing mammalian transcription factors from engineered intracellular *B. subtilis* as engineered endosymbionts provides a new tool for directing cell fate for therapeutic and research purposes. For example, the EES shows the possibility to be applied in treating damaging inflammatory conditions such as arthritis in the future. Overall, this work has good novelty. However, the writing is not easy to understand and needs further modification, such as strengthening the description of the details and the further interpretation of the data.

Thank you for the interest in our work. We have extended our discussion of the data and provided more details. We have also done further analysis and experiments that provide more clarity and resolution. We have also expanded details in results section to hopefully increase the clarity and alleviate ambiguity from introduction through results to discussion. Some examples:

Introduction:

Therefore, the EES was designed to express and deliver TFs in two operons for modulating macrophage phenotype towards pro- or anti-inflammatory states. One operon encodes the TFs signal transducer and activator of transcription 1 (STAT-1) and Krüppel-like factor 6 (KLF6) which induce a general response to an inflammatory state in macrophages, and the second encodes Krüppel-like factor 4 (KLF4) and GATA binding protein 3 (GATA-3) which are both characterized to drive an anti-inflammatory response in macrophages³³⁻³⁷. Both STAT-1 and KLF4 are upstream regulators that impact several mechanisms while KLF6 and GATA-3 are more specific in regulation which lends to a dual approach towards driving the desired cell fates³³⁻³⁷. When expressed from intracellular EES these TFs altered patterns of cell surface markers, cytokine and chemokine expression with some patterns of modulation towards anti- or pro-inflammatory phenotypes which indicates that the EES may be used to direct immune cell function and be used to elucidate mechanisms of macrophage response to intracellular bacteria.

Results: Engineered *B. subtilis* LLO secretes β -gal with delivery to the nuclei of J774A.1 cells

Protein secretion from the LLO strain was tested to further elucidate viability within the cytoplasm and initial demonstration of protein delivery to the nucleus. *B. subtilis* LLO was engineered to produce and secrete β -galactosidase (β -gal) (LLO-*lacZ*) through the twin-arginine translocation (Tat) pathway by synthesizing the PhoD signal peptide⁴⁰ and the amino acids from 126-132 of the simian virus (SV) 40 nuclear localization signal (NLS)⁴¹.

Results: Engineered *B. subtilis* LLO transcription factor delivery and modulation of J77A.1 cell marker expression

β -gal delivery by the LLO-*lacZ* strains demonstrated the ability of an intracellular EES to functionally deliver protein with transcriptional regulation Accordingly, the next step is to deliver protein to alter host cell fate and TFs were chosen. The LLO strain was engineered to secrete STAT-1 and KLF6 (LLO-SK; pro-inflammatory) or KLF4 and GATA-3 (LLO-KG; anti-inflammatory) with the same approach as β -gal to impact macrophage function (Fig. S5).

Introduction of these TFs into the genome of the *B. subtilis* LLO minimally impacted growth rates, did not change the impact on J774A.1 cell viability compared to the LLO strain and did not impact the ability of these new strains to escape destruction of phagosomes and persist in macrophages (percent cells containing bacteria and distribution of the fluorescence intensity relating to number of bacteria per cell) (Fig. S6). Immunofluorescence quantification confirmed production and delivery of STAT-1 and KLF6 or KLF4 and GATA-3 in cells containing LLO-SK or LLO-KG, respectively after 3 h of TF delivery (Fig. S7). Quantification of nuclear SNR identified increases in fluorescence in all four TFs after delivery from the LLO-SK and LLO-KG strains with and without addition of D-mannose. D-mannose significantly increased delivery of all four TFs. Additionally, the appropriate positive controls produced some increase in nuclear SNR but significantly less than the LLO-SK and LLO-KG strains over the 3 h.

To determine the impact of engineered *B. subtilis* LLO strain TF delivery on J774A.1 modulation, surface marker expression was examined in cells containing various strains (168 strain, LLO strain, LLO-SK and LLO-KG strains with and without mannose) and compared to M0 and M1+/M2+ polarized J774.1 cells at 24 or 48 h post-incubation.

Discussion:

During immune responses, STAT-1 impacts pro-inflammatory response as a strong modulator that is directly upregulated by the presence of pro-inflammatory IFN- γ and cytokines which was also shown in 3 h in this study (Fig. S7) while KLF6 has been shown to be upregulated in M1 polarized macrophages^{33,34,37}. The activity of KLF4 is directly stimulated by the signal cascade from the cytokine IL-4 and while this stimulation did not produce a significant immunofluorescent result in 3 h (Fig. S7), it is known to promote M2 polarization and GATA-3 is highly upregulated in M2 macrophages^{35,36}. The LLO-SK and LLO-KG strains produced and delivered more of these TFs to the nuclei of J774A.1 cells than using the known signal cascade inducers to cause the host cells to produce these TFs. Accordingly, this demonstrated the potential of *B. subtilis* LLO to impact the host cell function within the 3 h timeframe of interaction between the LLO strain and host cells.

In this study macrophage plasticity was modulated using TFs expressed from the EES. Cell surface markers are commonly used to differentiate M1 and M2 polarization^{7,11}. CD86 and CD206 expression was impacted by the bacterial conditions and the LLO-SK and LLO-KG were observed to regulate CD206 (Fig. 5) expression as expected based on the known activity of the TFs. At 24 h, CD86 response was more complex as it is expected to be upregulated as a general response to the bacteria. However, by 48 h the LLO-SK strain had increased the expression significantly compared to the LLO strain (Fig. 5, S8). The *B. subtilis* 168 strain also increased expression between 24 and 48 h but not significantly compared to the LLO strain. The surface protein CD206 has been shown to recognize the surface carbohydrates of pathogens and be triggered by proteases produced by *B. subtilis*^{46,47}. This would explain the increase in CD206 expression caused by the *B. subtilis* 168 strain and LLO strain in comparison to the untreated controls. Additionally, while CD86 is regulated directly by inflammatory responses to the engineered *B. subtilis* LLO, the plasticity of the macrophages could cause CD86 expression to change over time¹¹. It has been suggested that products of *B. subtilis* such as subblancin or exopolysaccharide (EPS), or the treatment of macrophages with *B. subtilis* spores can result in either M1 or M2 activation⁴⁸. Even with these complexities, clear trends were observed that suggest the LLO-SK and LLO-KG impacted surface marker expression in comparison to the LLO strain.

Although changes in cytokine and chemokine expression vary along the spectrum of macrophage polarization, there are well documented cytokines which are used to broadly identify M1 or M2 macrophages^{7,11,49}. These identifying cytokines and chemokines are important when studying disease as they provide information on the function and characteristics of the macrophage population^{42,50}. The M1 and M2 polarization classifications represent a trend in immune responses but complex and mixed signals such as bacterial stimulation and infection can cause these classifications to be incomplete or oversimplified^{49,51}. Complexities in macrophage activation, phenotype and plasticity were encountered in this study. In some instances, the LLO-*SK* and LLO-*KG* strains impacted the host cell as predicted in comparison to the LLO strain alone (Fig. 6, S10). The upregulation of IL-10 by LLO-*KG* and down regulation by LLO-*SK* was an expected result along with G-CSF being downregulated by LLO-*SK*^{35,37,52,53}. The LLO strain increasing IL-10 and G-CSF are explainable results as well as IL-10 and G-CSF have been shown to be produced in response to bacteria^{53–55}. However, in some cases there were results that were not anticipated, and possibly pleiotropic effects of each TF need to be considered. The change in IL-12p40 levels is an example of an unexpected result based on known signaling cascades. IL-12p40 is known to be produced during inflammatory phenotypes by nuclear factor kappa light chain enhancer of activated B cells (NF- κ B)⁵⁶. Therefore, LLO-*KG* resulting in an increase in production of IL-12p40 is unexpected; however, it is possible GATA-3 could play a role in IL-12p40 production. The IL-12p40 promoter has a characterized GATA binding site but GATA-3 has not been characterized in terms of impact on this promoter⁵⁶. The LLO-*SK* strain showed some downregulation at 48 h which could be due to STAT-1 driving alternative cytokine production such as IL-27p28^{56,57}. TNF- α downregulation by LLO-*KG* is another expected result³⁵ but downregulation by LLO-*SK* in comparison to the LLO strain indicates further complexity and potential pleiotropic impact from STAT-1 because of TNF- α being regulated by NF- κ B^{58–60}. Furthermore, there may have been metabolic impact on cytokine production due to the addition of D-mannose, which impairs glucose metabolism and is known to suppress the succinate-driven HIF1 α activation of IL-1 β leading to downregulation of IL-1 β ^{61,62}. Accordingly, results indicated that D-mannose had a significant impact on IL-1 β and other cytokines and chemokines which is important consideration for future EES studies that rely on sugar-inducible systems. Metabolic reprogramming is most likely playing a significant role in response to the chemical inducer of the engineered *B. subtilis* LLO TF operons and possibly the LLO strain alone⁶².

Overall, dynamic responses were seen in many of the cytokines and chemokines in response to the LLO strain with the LLO-*SK* and LLO-*KG* strains able to modify expression further in comparison to the LLO strain in some key examples. One component of the dynamic response was the elevated expression of cytokines and chemokines in response to the bacteria compared to the positive controls of IFN- γ and LPS (M1+) or IL-4 and IL-13 (M2+). Additionally, the difference in response to the bacteria conditions compared to the positive controls was larger with the cytokines and chemokines than the surface markers. Enhanced signaling protein (cytokines and chemokines) response from macrophages and other professional antigen presenting cells compared to surface marker expression has been observed previously in response to bacterial or signaling molecule stimulus which was further shown in this study^{63–65}. Other factors that could contribute to the complex response involves *B. subtilis* stimulating Toll-like receptor 2 (TLR2) compared to IFN- γ and LPS stimulating TLR4⁶⁶. Additionally, viable bacteria can cause a more dynamic response than just bacterial products alone⁶⁷. Altogether, even with the dynamic responses from the J774A.1 cells in response to complex signals, the

LLO-SK and LLO-KG strains were able to change trends in macrophage surface marker, cytokine and chemokine expression in comparison to the LLO strain. This observation is further supported by not observing significant differences in J774A.1 viability, percentage of cells containing fluorescently labeled bacteria and distribution of fluorescence intensity (relating to number of bacteria per cell) when comparing the LLO, LLO-SK and LLO-KG strains (Fig. S6). Future studies should focus on characterizing the impact of these strains in primary macrophages such as bone marrow derived macrophages to clearly understand application potential. In addition, more TF pairings informed by thorough studies that elucidate the complexity of macrophage response should be constructed and tested to optimize the response from the macrophages for the desired application.

Major points:

1) How to control the secretion of target proteins by EES? By fusion of TF genes with the Sec and Tat secretion signals or original substrates? The construction strategy of the exogenous protein-delivery strain should be elaborated in the result section.

We have included an explanation in the results section as well as Fig. S5 as a schematic of the operon construction. The PhoD secretion peptide is fused to the protein of interest then is cleaved by SPase I when the fully folded protein is translocated across the membrane leaving just the protein of interest secreted. The same construction strategy was used for both β -gal and the TFs except the intrinsic NLS to the TFs was used instead of a viral NLS as for β gal.

Results: Engineered *B. subtilis* LLO secretes β -gal with delivery to the nuclei of J774A.1 cells

Protein secretion from the LLO strain was tested to further elucidate viability within the cytoplasm and initial demonstration of protein delivery to the nucleus. *B. subtilis* LLO was engineered to produce and secrete β -galactosidase (β -gal) (LLO-*lacZ*) through the twin-arginine translocation (Tat) pathway by synthesizing the PhoD signal peptide⁴⁰ and the amino acids from 126-132 of the simian virus (SV) 40 nuclear localization signal (NLS)⁴¹.

Results: Engineered *B. subtilis* LLO transcription factor delivery and modulation of J774A.1 cell marker expression

β -gal delivery by the LLO-*lacZ* strains demonstrated the ability of an intracellular EES to functionally deliver protein with transcriptional regulation. Accordingly, the next step is to deliver protein to alter host cell fate and TFs were chosen. The LLO strain was engineered to secrete STAT-1 and KLF6 (LLO-SK; pro-inflammatory) or KLF4 and GATA-3 (LLO-KG; anti-inflammatory) with the same approach as β -gal to impact macrophage function (Fig. S5).

2) Since the title is "Engineered endosymbionts that direct mammalian cell gene expression", whether the transcriptional profile (RNA-seq) of the EES-TFs treated J774A.1 cells should be determined to compare with the controls (resting and polarized macrophages) to demonstrate the EES-TFs treatments indeed direct cell fate switch.

We agree that RNA-seq would be ideal for understanding the complex array of transcriptomic changes caused by the EES but we believe these resources are better allocated towards primary macrophages and as a result is outside the scope of the study. We agree with the implication to change the emphasis of the paper to be more specific in regards to impacting marker expression and cytokine/chemokine production. We believe that measuring these

protein outputs is consistent with understanding the functional change of the J774A.1 cells after treatment with the LLO strain and TF expressing strains. We have adjusted the title and discussion throughout the manuscript to be more accurate with what was measured. Some examples of changes:

Title:

Engineered endosymbionts that alter mammalian cell surface marker, cytokine and chemokine expression

Introduction:

When expressed from intracellular EES these TFs altered patterns of cell surface markers, cytokine and chemokine expression with some patterns of modulation towards anti- or pro-inflammatory phenotypes which indicates that the EES may be used to direct immune cell function and be used to elucidate mechanisms of macrophage response to intracellular bacteria.

Results: Modulation of J774A.1 cell cytokine and chemokine expression using engineered *B. subtilis* LLO strains

LLO-*SK* and LLO-*KG* were shown to alter J774A.1 cytokine and chemokine expression in comparison to LLO strain and positive controls (Fig. 6, S10). Addition of LLO-*SK* and LLO-*KG* strains to J774A.1 cells led to differential expression of cytokines compared to the cytokine profile observed for the LLO strain, as shown by levels of interleukin (IL)-10, IL-12p40 and tumor necrosis factor alpha (TNF- α) (Fig. 6A-D, S10).

Discussion: 3rd paragraph

In this study macrophage plasticity was modulated using TFs expressed from the EES. Cell surface markers are commonly used to differentiate M1 and M2 polarization^{7,11}.

Discussion: 4th paragraph

Although changes in cytokine and chemokine expression vary along the spectrum of macrophage polarization, there are well documented cytokines which are used to broadly identify M1 or M2 macrophages^{7,11,49}. These identifying cytokines and chemokines are important when studying disease as they provide information on the function and characteristics of the macrophage population^{42,50}.

Discussion: 5th paragraph

Altogether, even with the dynamic responses from the J774A.1 cells in response to complex signals, the LLO-*SK* and LLO-*KG* strains were able to change trends in macrophage surface marker, cytokine and chemokine expression in comparison to the LLO strain. This observation is further supported by not observing significant differences in J774A.1 viability, percentage of cells containing fluorescently labeled bacteria and distribution of fluorescence intensity (relating to number of bacteria per cell) when comparing the LLO, LLO-*SK* and LLO-*KG* strains (Fig. S6).

Minor points:

1) Line 132, the "Movie S2" in the parentheses should be Movie S1.

This is an important find and we have corrected this error.

2) Line 141, there is no Fig. 3B at all. Do you mean the right panel of Fig. 3? The same issue happens in Fig. 3A at Line 150.

Yes this is an astute observation. We have adjusted these lines to accurately reflect the figure as left and right panel.

Results: Viability of J774A.1 cells and *B. subtilis* LLO replication in the host cell cytoplasm

With IPTG induction, the LLO strain was observed to replicate in the host cell cytoplasm after phagosomal escape indicating active metabolism and viability (Fig. 3, right panel). An increase in the number of bacteria was visualized over time, after co-incubation with a 10:1 MOI. Zoomed regions demonstrate that each bacterium doubled twice during the two time points from 3 bacteria at 1 h to 12 at 2.5 h in this representative instance (Fig.3, left panel).

3) The units of Table S1 should be written clearly. How is the “%infected” calculated?

We have adjusted the table to make this more clear. The %infected is more accurately the amount of J774A.1 cells containing EES which is calculated by determining fluorescent EES throughout the z-depth of each cell over the total number of cells imaged.

Methods:

Brightfield and fluorescent images were acquired consecutively, using a 63x oil objective every 1 h beginning at 1 h post co-incubation and continuing until 4 h post incubation Z-stacks were taken at all time points at 0.4 μm steps to confirm *B. subtilis* LLO presence within cytoplasm. *B. subtilis* LLO presence in J774A.1 cells was quantified using Fiji (ImageJ) software and cell counter plugin by counting $>1.5 \mu\text{m}$ rods throughout z-depth. An area of 2090 μm by 1254 μm in each well was imaged and used to perform this quantification.

4) Line 159, the full name of β -gal should be given. The “mannose-inducible system” should be introduced briefly at the beginning of the section of “EES secretes β -gal with delivery to the nuclei of J774A.1 cells”.

We have introduced the mannose inducible system with supporting literature and provided the full name of β -gal.

Results: Engineered *B. subtilis* LLO secretes β -gal with delivery to the nuclei of J774A.1 cells

The mannose-inducible system was amplified from the *B. subtilis* strain 168 genome¹⁹ to provide a genetic switch specific to protein production. The mannose-inducible system was chosen due to the characterized uptake of this sugar in mammalian cells²⁰.

5) In Fig. 4, the difference between ‘mannose –NLS’ and ‘mannose NLS’ is not obvious in fluorescence microscopy figure. The effect of nuclear localization is not obvious without nucleus staining. With regards to the nuclear SNR, how the authors distinguished the nuclear signal and the cytoplasmic signal?

We have included an example of how a nuclear stain was used to guide region of interest drawing around the nucleus to determine nuclear SNR as Fig. S3. When SV40 NLS was discovered (reference 41), the authors also observed β -gal accumulating in the nucleus without any NLS signal indicating an intrinsic signal even on a bacterial protein.

Results: Fig. S3

Reviewer #2 (Remarks to the Author):

I do not have any additional concerns. I recommend publication of the article.